# GENOT: Entropic (Gromov) Wasserstein Flow Matching with Applications to Single-Cell Genomics

**Dominik Klein**[*][†]
Helmholtz Munich
dominik.klein@helmholtz-munich.de

**Théo Uscidda**[*]
CREST-ENSAE
theo.uscidda@ensae.fr

**Fabian Theis**
Helmholtz Munich
fabian.theis@helmholtz-munich.de

**Marco Cuturi**
Apple
cuturi@apple.com

## Abstract

Single-cell genomics has significantly advanced our understanding of cellular behavior, catalyzing innovations in treatments and precision medicine. However, single-cell sequencing technologies are inherently destructive and can only measure a limited array of data modalities simultaneously. This limitation underscores the need for new methods capable of realigning cells. Optimal transport (OT) has emerged as a potent solution, but traditional discrete solvers are hampered by scalability, privacy, and out-of-sample estimation issues. These challenges have spurred the development of neural network-based solvers, known as neural OT solvers, that parameterize OT maps. Yet, these models often lack the flexibility needed for broader life science applications. To address these deficiencies, our approach learns stochastic maps (i.e. transport plans), allows for any cost function, relaxes mass conservation constraints and integrates quadratic solvers to tackle the complex challenges posed by the (Fused) Gromov-Wasserstein problem. Utilizing flow matching as a backbone, our method offers a flexible and effective framework. We demonstrate its versatility and robustness through applications in cell development studies, cellular drug response modeling, and cross-modality cell translation, illustrating significant potential for enhancing therapeutic strategies.

## 1 Introduction

**Discrete Optimal Transport in Single-Cell Genomics.** Due to the destructive nature of single-cell sequencing technologies, it is fundamental to realign distributions of sequenced cells. Discrete optimal transport (OT) has proven useful in this task. For example, it is standard practice to leverage the framework of OT to reconstruct cellular trajectories across developmental time points [72] and to study the response of cells to external perturbations like drugs or gene knockouts [18]. In these settings, both distributions, i.e. earlier and later time point and before perturbation and after perturbation, respectively, live in the same space, a setting which we refer to as *linear* OT. In contrast, most applications of OT in single-cell genomics require maps across (partially) incomparable spaces, the setting which we refer to as *quadratic* OT. For example, analysis of spatial transcriptomics datasets requires the quadratic formulation of OT as the coordinate systems of two different slices are warped and rotated, thus not living in the same coordinate space [60]. Similarly, adding spatial information as a prior has been shown to improve the performance of modeling the trajectory of cells in the discrete setting [39]. Analogously, adding information from lineage tracing readouts [89]

---

[*]Equal contribution
[†]Work done during internship at Apple

38th Conference on Neural Information Processing Systems (NeurIPS 2024).

recovers the evolution of cells more faithfully [42]. Another prevalent task in single-cell genomics which is commonly approached with quadratic OT solvers is the alignment of cells across modalities [17]. Most single-cell technologies capture only one modality, which is insufficient to obtain a comprehensive representation of the cellular state. As single-cell datasets grow larger [65], traditional OT solvers become less applicable to tasks in single-cell genomics due to their high computational complexity. Thanks to recent advancements in low-rank OT [69, 70, 71], the aforementioned OT-based algorithms are more accessible to single-cell biologists [39]. Yet, the non-parametric nature of discrete OT solvers entails issues with respect to data privacy and prevents their application to large scale single-cell atlases capturing millions of cells [65]. Moreover, out-of-samples estimation is limited to very specific scenarios [63].

**Current Limitations of Neural Optimal Transport in Single-Cell Genomics.** To overcome these limitations, neural OT solvers have been leveraged to study cellular perturbations [7, 8, 86] and to model cellular trajectories [21, 83]. Yet, these methods estimate Monge maps, i.e., deterministic maps, contradicting the assumption that cells evolve stochastically [19]. Stochastic formulations are also favorable as they can produce a conditional distribution that can be used to quantify uncertainty [25]. In the discrete setting, stochastic maps can be obtained from entropy-regularized OT (EOT) [13]. Recently, a number of works have addressed learning EOT plans [29, 41, 76, 84, 85] in the neural setting. Yet, all of these methods are limited in the choice of the cost function, with most of them being restricted to the squared Euclidean cost. Single-cell genomics data is known to be non-Euclidean [57] and thus requires a flexible choice of the cost function in OT applications as demonstrated in the discrete OT setting in Demetci et al. [17], Huguet et al. [35], Klein et al. [39]. The third requirement for applying OT to single-cell genomics is the option to lift the mass conservation constraint, allowing for *unbalanced* optimal transport [11, 24, 74]. Unbalanced OT is crucial to model cellular growth and death as well as for automatically discarding outliers, which are prevalent in highly noisy measurements in single-cell genomics. Eyring et al. [20], Lübeck et al. [50], Yang and Uhler [94] proposed ways to incorporate unbalancedness into deterministic linear OT maps, while unbalanced formulations for entropic neural OT have barely been explored. The most severe shortcoming of the existing plethora of neural OT estimators is their limitation to the linear OT scenario, i.e., to learning maps within spaces. Yet, most applications in single-cell genomics require the Gromov-Wasserstein (GW) [54] or the fused Gromov-Wasserstein (FGW) [88] formulation. To the best of our knowledge, the only neural formulation for GW proposed thus far learns deterministic, balanced maps for the inner product costs, using a min-max-min optimization procedure, severely limiting its applications [58]. Hence, we arrive at four necessities that need to be fulfilled *and* have to be flexibly combinable to make neural OT generally applicable to problems in single-cell genomics:

- **N1**: modeling the evolution of cells stochastically as opposed to deterministically,
- **N2**: flexibly choosing cost functions due to non-Euclidean geometry of single-cell genomics data,
- **N3**: allowing for unbalanced OT to model cellular growth and death or accounting for outliers,
- **N4**: mapping across *completely* or *partially* incomparable spaces.

**Contributions.** Hence, we propose GENOT (Generative Entropic Neural Optimal Transport), a powerful and flexible neural OT framework that satisfies all of the above requirements:

- GENOT is the first method that parameterizes linear and quadratic EOT couplings for *any* cost by modeling their conditional distributions, using flow matching [45] as a backbone.
- We extend GENOT to the unbalanced setting, resulting in U-GENOT, to flexibly allow for mass variations in both the linear and the quadratic setting.
- We extend (U-)GENOT to address the fused problem and this way propose, to the best of our knowledge, the first neural OT solver for the FGW problem.
- We showcase GENOT's ability to handle common challenges in single-cell biology: we (i) quantify lineage branching events in the developing mouse pancreas, (ii) predict cellular responses to drug perturbations, along with a well-calibrated uncertainty estimation, and (iii) translate ATAC-seq to RNA-seq with GW and introduce a novel method to perform this translation task with FGW.

## 2 Background

**Notations.** Let $\mathcal{X} \subset \mathbb{R}^p$, $\mathcal{Y} \subset \mathbb{R}^q$ compact sets, referred to as the source and the target domain, respectively. In general, $p \neq q$. The set of positive (resp. probability) measures on $\mathcal{X}$ is denoted by $\mathcal{M}^+(\mathcal{X})$ (resp. $\mathcal{M}_1^+(\mathcal{X})$). For $\pi \in \mathcal{M}^+(\mathcal{X} \times \mathcal{Y})$, $\pi_1 := p_1 \sharp \pi$ and $\pi_2 := p_2 \sharp \pi$ denote its marginals.

Then, for $\mu \in \mathcal{M}^+(\mathcal{X}), \nu \in \mathcal{M}^+(\mathcal{Y})$, $\Pi(\mu, \nu) = \{\pi : \pi_1 = \mu, \pi_2 = \nu\}$. $\frac{d\mu}{d\nu}$ denotes the relative density of $\mu$ w.r.t. $\nu$, s.t. $\mu = \frac{d\mu}{d\nu} \cdot \nu$. For $\rho, \gamma \in \mathcal{M}^+(\mathcal{X})$, $\mathrm{KL}(\rho|\gamma) = \int_{\mathcal{X}} \log(\frac{d\rho}{d\gamma}) \, d\rho - \int_{\mathcal{X}} d\gamma + \int_{\mathcal{X}} d\rho$.

**Linear Entropic OT.** Let $c : \mathcal{X} \times \mathcal{Y} \to \mathbb{R}$ be a continuous cost function, $\mu \in \mathcal{M}_1^+(\mathcal{X}), \nu \in \mathcal{M}_1^+(\mathcal{Y})$ and $\varepsilon \geq 0$. The linear entropy-regularized OT problem reads

$$\min_{\pi \in \Pi(\mu, \nu)} \int_{\mathcal{X} \times \mathcal{Y}} c \, d\pi + \varepsilon \mathrm{KL}(\pi | \mu \otimes \nu). \tag{LEOT}$$

A solution $\pi_\varepsilon^\star$ of (LEOT) always exists and is unique when $\varepsilon > 0$. (LEOT) is also known as the static Schrödinger bridge (SB) problem [52]. With $\varepsilon = 0$, we recover the Kantorovich [38] problem. For discrete $\mu$ and $\nu$, we can solve (LEOT) with the Sinkhorn algorithm [13], whose complexity for $n$ points is $\mathcal{O}(n^2)$ in time, and $\mathcal{O}(n)$ or $\mathcal{O}(n^2)$ in memory, depending on $c$.

**Quadratic Entropic OT.** As opposed to considering an *inter-domain* cost defined on $\mathcal{X} \times \mathcal{Y}$, quadratic entropic OT is concerned with seeking couplings that minimize the distortion of the geometries induced by *intra-domain* cost functions $c_\mathcal{X} : \mathcal{X} \times \mathcal{X} \to \mathbb{R}$ and $c_\mathcal{Y} : \mathcal{Y} \times \mathcal{Y} \to \mathbb{R}$:

$$\min_{\pi \in \Pi(\mu, \nu)} \int_{(\mathcal{X} \times \mathcal{Y})^2} D_{c_\mathcal{X}, c_\mathcal{Y}} \, d(\pi \otimes \pi) + \varepsilon \, \mathrm{KL}(\pi \| \mu \otimes \nu), \tag{QEOT}$$

where $D_{c_\mathcal{X}, c_\mathcal{Y}}(\mathbf{x}, \mathbf{y}, \mathbf{x}', \mathbf{y}') := |c_X(\mathbf{x}, \mathbf{x}') - c_Y(\mathbf{y}, \mathbf{y}')|^2$ quantifies the pointwise cost distortion. A solution $\pi_\varepsilon^\star$ to Prob.(QEOT) always exists (see B.1). With $\varepsilon = 0$, we recover the Gromov-Wasserstein [54] problem, which is the standard OT formulation for transporting measures supported on *incomparable* spaces. In addition to statistical benefits [95], using $\varepsilon > 0$ also offers computational benefits, since for discrete $\mu$ and $\nu$, we can solve (QEOT) with a mirror descent scheme iterating the Sinkhorn algorithm [62]. For measures on $n$ points, its time complexity is $\mathcal{O}(n^2)$ or $\mathcal{O}(n^3)$, depending on $c_\mathcal{X}$ and $c_\mathcal{Y}$ [70, Alg. 1&2], while its memory complexity is always $\mathcal{O}(n^2)$.

**Unbalanced Extensions.** The EOT formulations presented above can only handle measures with the same total mass. Unbalanced optimal transport (UOT) [11, 44] lifts this constraint by penalizing the deviation of $p_1 \sharp \pi$ to $\mu$ and $p_2 \sharp \pi$ to $\nu$ with a divergence. Using the KL and introducing weightings $\lambda_1, \lambda_2 > 0$ the unbalanced extension of (LEOT) reads

$$\min_{\pi \in \mathcal{M}^+(\mathcal{X} \times \mathcal{Y})} \int_{\mathcal{X} \times \mathcal{Y}} c \, d\pi + \varepsilon \mathrm{KL}(\pi | \mu \otimes \nu) + \lambda_1 \mathrm{KL}(\pi_1 | \mu) + \lambda_2 \mathrm{KL}(\pi_2 | \nu). \tag{ULEOT}$$

This problem can be solved efficiently in the discrete setting using a variant of the Sinkhorn algorithm [24, 80]. Analogously, quadratic OT also admits an unbalanced generalization, which reads

$$\min_{\pi \in \mathcal{M}^+(\mathcal{X} \times \mathcal{Y})} \int_{(\mathcal{X} \times \mathcal{Y})^2} D_{c_\mathcal{X}, c_\mathcal{Y}} \, d(\pi \otimes \pi) + \varepsilon \mathrm{KL}^\otimes(\pi | \mu \otimes \nu) + \lambda_1 \mathrm{KL}^\otimes(\pi_1 | \mu) + \lambda_2 \mathrm{KL}^\otimes(\pi_2 | \nu), \tag{UQEOT}$$

where $\mathrm{KL}^\otimes(\rho|\gamma) = \mathrm{KL}(\rho \otimes \rho | \gamma \otimes \gamma)$. A solution $\pi_{\varepsilon, \tau}^\star$ to Prob.(UQEOT) always exists (see B.1). In the discrete setting, Prob. (UQEOT) can be solved using an extension of Peyré et al. [62]'s scheme introduced by Séjourné et al. [81]. Each solver has the same time and memory complexity as its balanced counterpart. For both (ULEOT) and (UQEOT), instead of selecting $\lambda_i$, we introduce $\tau_i = \frac{\lambda_i}{\lambda_i + \varepsilon}$ s.t. we recover the marginal constraint for $\tau_i = 1$, when $\lambda_i \to +\infty$. We write $\tau = (\tau_1, \tau_2)$.

**Flow Matching (FM).** Given a prior $\rho_0 \in \mathcal{M}_1^+(\mathbb{R}^d)$ and a time-dependent vector field $(v_t)_{t \in [0,1]}$, one can define a probability path $p_t$ starting from $\rho_0$ using the flow $\phi_t$ solving

$$\frac{d}{dt} \phi_t(\mathbf{z}) = v_t(\phi_t(\mathbf{z})), \quad \phi_0(\mathbf{z}) = \mathbf{z}, \tag{1}$$

by setting $p_t = \phi_t \sharp \rho_0$. We then say that $v_t$ generates the path $p_t$. Continuous Normalizing Flows [9] (CNFs) model $v_{t,\theta}$ with a neural network , which is trained to match a terminal condition $p_1 = \rho_1 \in \mathcal{M}_1^+(\mathbb{R}^d)$. FM [45] is a simulation-free technique to train CNFs by constructing individual paths between samples, and minimizing

$$\mathbb{E}_{t, Z_0 \sim \rho_0, Z_1 \sim \rho_1}[\|v_{t,\theta}([Z_0, Z_1]_t) - (Z_1 - Z_0)\|_2^2], \tag{2}$$

where $[Z_0, Z_1]_t := (1-t)Z_0 + tZ_1$ with $t \sim \mathcal{U}([0,1])$. If this loss is 0, the flow maps $\rho_0$ to $\rho_1$, i.e., $\phi_1 \sharp \rho_0 = \rho_1$. This property is often referred to as FM preserving the marginal distribution.

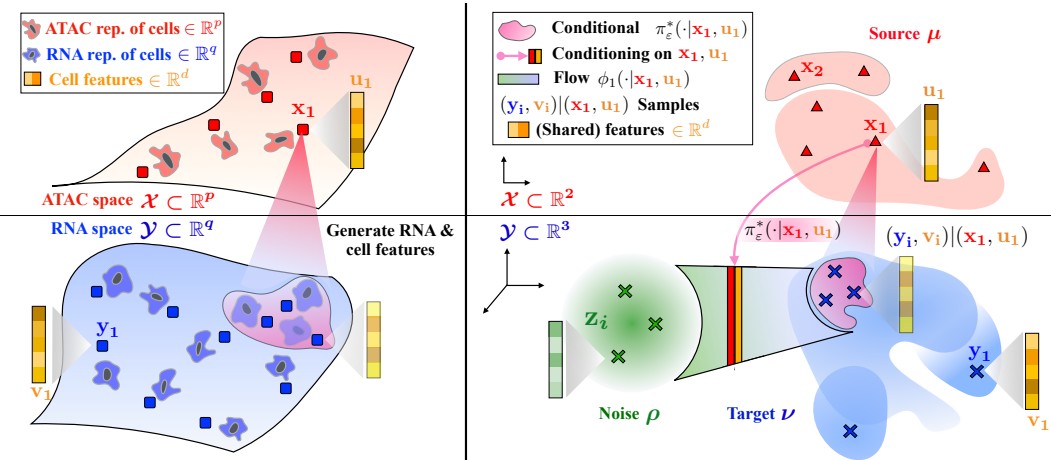

Figure 1: **Left: What do we do?** One task we consider is generating RNA cell profiles from ATAC measurements and an additional cell feature. This is explained in § 5.2, and demonstrated in Fig. 4. As the cells live on manifolds in two (partially) incomparable spaces , we rely on the Fused Gromov-Wasserstein (FGW) formulation, as described in § 3.3. Here, the incomparable structural information is contained in the ATAC and the RNA measurements, while the comparable information are the cell features. **Right: How do we do it?** For each $(\mathbf{x}, \mathbf{u})$ in the support of the source $\boldsymbol{\mu}$, we learn a flow $\phi_1(\cdot|\mathbf{x}, \mathbf{u})$ from the noise $\boldsymbol{\rho}$ to the conditional $\pi_\varepsilon^\star(\cdot|\mathbf{x}, \mathbf{u})$, whose support lies in that of the target $\boldsymbol{\nu}$. The flow is multi-modal: It allows sampling structural informations $\mathbf{y}$, as well as features $\mathbf{v}$ simultaneously. We highlight this procedure for a specific pair $(\mathbf{x}_1, \mathbf{u}_1)$, with $\boldsymbol{p} = 2$ and $\boldsymbol{q} = 3$.

## 3 Generative Entropic Neural OT

We introduce GENOT, a method to learn EOT couplings (thus satisyfing **N1**) with any cost (**N2**) by learning their conditional distributions. In § 3.1, we focus on balanced OT, and show that GENOT approximates linear or quadratic EOT couplings (**N4**), solutions to (LEOT) or (QEOT) respectively. Then, in § 3.2, we extend GENOT to the unbalanced setting by loosening the conservation of mass constraint (**N3**). We define U-GENOT, which approximates solutions to (ULEOT) and (UQEOT). In § 3.3, we highlight that GENOT also adresses a fused problem, combining (LEOT) and (QEOT) (**N4**). Finally, we demonstrate in § 3.4 the GENOT algorithm, that allows to flexibly adapt to different OT formulations, which is key for easy usability for single-cell genomics analysts.

### 3.1 Learning Entropic Couplings with GENOT

Let $\mu \in \mathcal{M}_1^+(\mathcal{X}), \nu \in \mathcal{M}_1^+(\mathcal{Y})$ and $\pi_\varepsilon^\star$ be an EOT coupling between $\mu$ and $\nu$, which can be a solution of problem (LEOT) or (QEOT). By the measure disintegration theorem, we get

$$\mathrm{d}\pi_\varepsilon^\star(\mathbf{x}, \mathbf{y}) = \mathrm{d}\mu(\mathbf{x})\,\mathrm{d}\pi_\varepsilon^\star(\mathbf{y}|\mathbf{x})\,.$$

Knowing $\mu$, we can hence fully describe $\pi_\varepsilon^\star$ via the conditional distributions $(\pi_\varepsilon^\star(\cdot|\mathbf{x}))_{\mathbf{x}\in\mathcal{X}}$. The latter are of great practical interest, as they provide a way to transport a point $\mathbf{x}$ sampled from $\mu$ to the target domain $\mathcal{Y}$; either *stochastically* by sampling $\mathbf{y}_1, ..., \mathbf{y}_n$ from $\pi_\varepsilon^\star(\cdot|\mathbf{x})$, or *deterministically* by averaging over conditional samples: $T_\varepsilon(\mathbf{x}) := \mathbb{E}_{Y\sim\pi_\varepsilon^\star(\cdot|\mathbf{x})}[Y]$. Moreover, we can assess the uncertainty of these predictions using any statistic of $\pi_\varepsilon^\star(\cdot|\mathbf{x})$, which is crucial in single-cell genomics.

**Learning the Conditional Distributions.** Let $\rho = \mathcal{N}(0, I_q)$ the standard Gaussian on the target space $\mathbb{R}^q \supset \mathcal{Y}$. From the noise outsourcing lemma [37], there exists a collection of conditional generators $\{T^\star(\cdot|\mathbf{x})\}_{\mathbf{x}\in\mathcal{X}}$ s.t. $T^\star(\cdot|\mathbf{x}) : \mathbb{R}^q \to \mathbb{R}^q$ and for each $\mathbf{x}$ in the support of $\mu$, $\pi_\varepsilon^\star(\cdot|\mathbf{x}) = T^\star(\cdot|\mathbf{x})\sharp\rho$. This means that if $Z \sim \rho$, then $Y = T^\star(Z|\mathbf{x}) \sim \pi_\varepsilon^\star(\cdot|\mathbf{x})$. Here, we seek to learn a neural collection $\{T_\theta(\cdot|\mathbf{x})\}_{\mathbf{x}\in\mathcal{X}}$ of such conditional generators, fitting the constraint $T_\theta(\cdot|\mathbf{x})\sharp\rho = \pi_\varepsilon^\star(\cdot|\mathbf{x})$, for any $\mathbf{x}$ in the support of $\mu$. We employ the FM framework (see § 2) and parameterize each $T_\theta(\cdot|\mathbf{x})$ implicitly, as the flow induced by a neural vector field $v_{t,\theta}(\cdot|\mathbf{x}) : \mathbb{R}^q \to \mathbb{R}^q$. Namely, $T_\theta(\cdot|\mathbf{x}) = \phi_1(\cdot|\mathbf{x}) : \mathbb{R}^q \to \mathbb{R}^q$ where $\phi_t(\cdot|\mathbf{x})$ solves

$$\frac{\mathrm{d}}{\mathrm{d}t}\phi_t(\mathbf{z}|\mathbf{x}) = v_{t,\theta}(\phi_t(\mathbf{z}|\mathbf{x})|\mathbf{x}), \quad \phi_0(\mathbf{z}|\mathbf{x}) = \mathbf{z}. \tag{3}$$

We stress that while $\mathbf{x} \in \mathcal{X} \subset \mathbb{R}^d$, the flow $\phi_1(\cdot|\mathbf{x})$ from $\rho$ to $\pi_\varepsilon^\star(\cdot|\mathbf{x})$ is defined on the target space $\mathbb{R}^q \supset \mathcal{Y}$. Hence, we can map samples *within* the same space when $p = q$, but also *across* incomparable spaces when $p \neq q$. In particular, this allows us to consider the quadratic OT problem (QEOT). Thus, for each $\mathbf{x}$, we optimize $v_{t,\theta}(\cdot|\mathbf{x})$ by minimizing the FM loss (2) to map $\rho$ to $\pi_\varepsilon^\star(\cdot|\mathbf{x})$, i.e.

$$\mathbb{E}_{t, Z \sim \rho, Y \sim \pi_\varepsilon^\star(\cdot|\mathbf{x})}[\|v_{t,\theta}([Z, Y]_t|\mathbf{x}) - (Y - Z)\|_2^2], \tag{4}$$

where $[Z, Y]_t = (1 - t)Z + tY$ interpolates between noise and conditional vectors. Averaging for all $\mathbf{x}$ in the support of $\mu$ and using Fubini's Theorem, we derive the GENOT loss:

$$\mathcal{L}_{\text{GENOT}}(\theta) := \mathbb{E}_{t, Z \sim \rho, (X, Y) \sim \pi_\varepsilon^\star}[\|v_{t,\theta}([Y, Z]_t|X) - (Y - Z)\|_2^2]. \tag{5}$$

In practice, we optimize a *sample-based* GENOT loss by estimating $\pi_\varepsilon^n = \sum_{i,j} \mathbf{P}_\varepsilon^{ij} \delta_{(X_i, Y_j)}$, from $X_1, \ldots, X_n \sim \mu$ and $Y_1, \ldots, Y_n \sim \nu$, with a *discrete* EOT solver, see Alg. 1.

**GENOT Captures the Signal from the Mini-Batch Couplings.** Standard FM only preserves straight couplings [47], while bridge matching (its stochastic counterpart) only preserves the linear EOT coupling for $c(\mathbf{x}, \mathbf{y}) = \|\mathbf{x} - \mathbf{y}\|_2^2$ [5]. In contrast, GENOT is a conditional FM model, learning the conditional distributions of the coupling independently of each other: For each $\mathbf{x}$, we leverage FM to learn a conditional flow $\phi_1(\cdot|\mathbf{x})$ that maps $\rho$ to $\pi_\varepsilon^\star(\cdot|\mathbf{x})$. Therefore, as FM preserves the marginal distributions (see 2), GENOT preserves all couplings. Thus, it always captures the signal carried out by the mini-batch couplings, in both linear and quadratic settings, and for any cost. We highlight this property in Fig. 5, comparing GENOT to other OT-based FM approaches [64, 84, 85].

**GENOT Handles Any Cost.** We can use GENOT to approximate linear or quadratic EOT couplings. In both cases, we make no assumptions on the cost functions: We only need to evaluate them on samples to estimate $\pi_\varepsilon^\star$ using a discrete solver, see GENOT Alg. 1, line 5. In particular, we can use implicitly defined costs, whose evaluation requires a non-differentiable sub-routine. For instance, we can use the geodesic distance on the data manifold, which can be approximated from the shortest path distance on the $k$-nn graph induced by the Euclidean distance [12, 77]. This approach has demonstrated its effectiveness in a range of single-cell genomics tasks relying on discrete OT [17, 35, 39]. We use it with GENOT for both linear (see § 5.1) and quadratic OT (see § 5.2).

**GENOT Approximates Conditional Densities.** For each $\mathbf{x}$, we build a CNF $p_t(\cdot|\mathbf{x}) = \phi_t(\cdot|\mathbf{x})\sharp\rho$ between $p_0 = \rho$ and $p_1 = \pi_\varepsilon^\star(\cdot|\mathbf{x})$. From the instantaneous change of variables formula [9], we can then approximate the conditional density $\pi_\varepsilon^\star(\mathbf{y}|\mathbf{x})$ at an arbitrary point $\mathbf{y} \in \mathcal{Y}$, proving useful to evaluate the likelihood of one-to-one matches between cells, see Fig.23.

## 3.2 U-GENOT: Extension to the Unbalanced Setting

**Re-Balancing the UOT Problems.** In its standard form, GENOT respects marginal constraints, so it cannot directly handle unbalanced formulations (ULEOT) or (UQEOT). We show that unbalanced EOT problems can be *re-balanced*. Eyring et al. [20], Lübeck et al. [50], Yang and Uhler [93] introduced previously these ideas in the Monge map estimation setting, namely, in a static and deterministic setup. Our method stems from the fact that, for both linear and quadratic OT, the unbalanced EOT coupling $\pi_{\varepsilon,\tau}^\star$ between $\mu \in \mathcal{M}^+(\mathcal{X})$, $\nu \in \mathcal{M}^+(\mathcal{Y})$ solves a balanced EOT problem between its marginals, which are re-weighted versions $\tilde{\mu}$ and $\tilde{\nu}$ of $\mu$ and $\nu$, that have the same mass.

**Proposition 3.1** (Re-Balancing the unbalanced problems.). *Let $\pi_{\varepsilon,\tau}^\star$ be an unbalanced EOT coupling, solution of* (ULEOT) *or* (UQEOT) *between $\mu \in \mathcal{M}^+(\mathcal{X})$ and $\nu \in \mathcal{M}^+(\mathcal{Y})$. We note $\tilde{\mu} = p_1\sharp\pi_{\varepsilon,\tau}^\star$ and $\tilde{\nu} = p_2\sharp\pi_{\varepsilon,\tau}^\star$ its marginals. Then, in both cases, $\tilde{\mu}$ (resp. $\tilde{\nu}$) has a density w.r.t. $\mu$ (resp. $\nu$). That is, there exist two re-weighting functions, one on each space, $\eta : \mathcal{X} \rightarrow \mathbb{R}^+$ and $\xi : \mathcal{Y} \rightarrow \mathbb{R}^+$ s.t. $\tilde{\mu} = \eta \cdot \mu$ and $\tilde{\nu} = \xi \cdot \nu$. Furthermore, $\tilde{\mu}$ and $\tilde{\nu}$ have the same total mass and*

1. *(Linear) $\pi_{\varepsilon,\tau}^\star$ solves the balanced problem* (LEOT) *between $\tilde{\mu}$ and $\tilde{\nu}$ with the same $\varepsilon$.*

2. *(Quadratic) Provided that $c_\mathcal{X}$, $c_\mathcal{Y}$ (or $-c_\mathcal{X}$, $-c_\mathcal{Y}$) are CPD kernels (see Def. B.2), $\pi_{\varepsilon,\tau}^\star$ solves the balanced problem* (QEOT) *between $\tilde{\mu}$ and $\tilde{\nu}$ with $\varepsilon' = m(\pi_{\varepsilon,\tau}^\star)\varepsilon$, where $m(\pi_{\varepsilon,\tau}^\star) = \pi_{\varepsilon,\tau}^\star(\mathcal{X} \times \mathcal{Y})$.*

**Learning the Coupling and the Re-Weightings Simultaneously.** Thanks to Prop. 3.1, we aim to [i] learn a balanced EOT coupling between $\tilde{\mu}$ and $\tilde{\nu}$ along with [ii] the re-weighting functions $\eta, \xi$. Learning them is desirable since they model the creation and destruction of mass. We do both

simultaneously by adapting the GENOT procedure. Formally, we seek to optimize:

$$\mathcal{L}_{\text{U-GENOT}}(\theta) = \mathbb{E}_{t, Z \sim \rho, (X,Y) \sim \pi^{\star}_{\varepsilon,\tau}}[\|v_{t,\theta}([Z,Y]_t|X) - (Y - Z)\|_2^2] \ [\text{i}]$$

$$+ \mathbb{E}_{X \sim \mu}[(\eta - \eta_\theta)(X)^2] + \mathbb{E}_{Y \sim \nu}[(\xi - \xi_\theta)(Y)^2], \qquad [\text{ii}]$$

where $\eta_\theta, \xi_\theta$ are (non-negative) neural re-weighting functions. Crucially, similar to (balanced) GENOT, we only need to estimate a discrete unbalanced EOT coupling $\pi^n_{\varepsilon,\tau}$ using samples $X_1, \ldots, X_n \sim \mu$ and $Y_1, \ldots, Y_n \sim \nu$ to compute the two components, [i] and [ii], of the U-GENOT loss. We build upon theoretical insights on the linear OT case and extend them to the quadratic OT case in practice.

**Proposition 3.2** (Pointwise estimation of re-weighting functions.). *Let $\pi^n_{\varepsilon,\tau} = \sum_{i,j} \mathbf{P}^{i,j}_{\varepsilon,\tau} \delta_{(X_i, Y_j)}$, solution to* (ULEOT) *between empirical counterparts of $\mu$ and $\nu$. Let $\mathbf{a} = \mathbf{P}_{\varepsilon,\tau} \mathbf{1}_n$ and $\mathbf{b} = \mathbf{P}^{\top}_{\varepsilon,\tau} \mathbf{1}_n$ its marginals weights. Then, almost surely, $n\, a_i \to \eta(X_i)$ and $n\, b_i \to \xi(Y_i)$.*

Using Prop. 3.1, $\hat{\pi}^n_{\varepsilon,\tau}$ is a balanced EOT coupling between its marginals, which are empirical approximations of $\tilde{\mu}$ and $\tilde{\nu}$. Hence, we estimate the first term [i] of the loss as we do in the balanced case by sampling from the discrete coupling. Furthermore, Prop.3.2 highlights that the estimation of $\hat{\pi}^n_{\varepsilon,\tau}$ also provides a consistent estimate of the re-weighting function evaluations at each $X_i$ and $Y_i$. This enables estimating the second term [ii]. Therefore, switching from GENOT to U-GENOT simply involves using an unbalanced solver instead of a balanced one, and regressing the neural re-weighting functions on the marginal weights of the estimated discrete coupling. We detail our procedure in Alg. 1, showing the additional steps w.r.t. GENOT in teal.

### 3.3 Combining Linear and Quadratic OT

We show in § 3.1 and § 3.2 how to use GENOT to solve OT problems within the same space or across incomparable spaces. On the other hand, numerous real-world problems pose the challenge of the source and target domains being only *partially* incomparable [88]. Therefore, suppose that the source and target space can be decomposed as $\mathcal{X} = \Omega \times \tilde{\mathcal{X}}$ and $\mathcal{Y} = \Omega \times \tilde{\mathcal{Y}}$, respectively. Intuitively, a sample $(\mathbf{u}, \mathbf{x}) \in \Omega \times \tilde{\mathcal{X}}$ can be interpreted as a structural information $\mathbf{x}$ equipped with a feature $\mathbf{u}$. Assume we are given a cost $c : \Omega \times \Omega \to \mathbb{R}$ to compare features, along with the intra-domain costs $c_{\tilde{\mathcal{X}}}, c_{\tilde{\mathcal{Y}}}$. The entropic Fused Gromov-Wasserstein (FGW) problem reads

$$\min_{\pi \in \Pi(\mu,\nu)} \int_{((\Omega \times \tilde{\mathcal{X}}) \times (\Omega \times \tilde{\mathcal{Y}}))^2} D^c_{c_{\tilde{\mathcal{X}}}, c_{\tilde{\mathcal{Y}}}} \, \mathrm{d}(\pi \otimes \pi) + \varepsilon \mathrm{KL}(\pi | \mu \otimes \nu), \qquad \text{EFGW}$$

where $D^c_{c_{\tilde{\mathcal{X}}}, c_{\tilde{\mathcal{Y}}}}((\mathbf{u}, \mathbf{x}), (\mathbf{v}, \mathbf{y}), \mathbf{x}', \mathbf{y}') := (1-\alpha)\, c(\mathbf{u}, \mathbf{v}) + \alpha\, |c_{\tilde{\mathcal{X}}}(\mathbf{x}, \mathbf{x}') - c_{\tilde{\mathcal{Y}}}(\mathbf{y}, \mathbf{y}')|^2$ and $\alpha \in [0, 1]$. This loss combines the pointwise structural distortion and the feature information. We refer to the additional cost on the feature information as the *fused term*. The weighting $\alpha$ allows us to interpolate between purely linear OT on the feature ($\alpha = 0$), and purely quadratic OT ($\alpha = 1$) on the structural information. Problem (EFGW) also admits an unbalanced extension, derived similarly as (UQEOT) with the quadratic $\mathrm{KL}^{\otimes}$ [82]. An (un)balanced fused EOT coupling always exists (see B.1), it minimizes distortion along the structural information and displacement cost along the features.

**(U-)GENOT for the Fused Setting.** Whether in the balanced or unbalanced setting, we can use our method to learn a specific coupling as soon as it can be estimated from samples. We stress that the discrete solvers introduced by Peyré et al. [62] and Séjourné et al. [81] we use for (QEOT) and (UQEOT), respectively, are still applicable in the fused setting. As a result, we can approximate solutions of (EFGW) and its unbalanced counterpart with (U-)GENOT. To illustrate the learning outcome, take a solution $\pi^{\star}_\alpha$ of (EFGW). Learning $\pi^{\star}_\alpha$ with our method amounts to training vector fields that are conditioned on pairs of modalities from the source domain $v_{t,\theta}(\cdot, |\mathbf{u}, \mathbf{x})$, to sample pairs of modalities from the target domain via the flow: $Z \sim \rho$, $\phi_1(Z|\mathbf{u}, \mathbf{x}) = (V, Y) \sim \pi^{\star}_\alpha(\cdot | \mathbf{u}, \mathbf{x})$. The sampled modalities $(V, Y)$ (i) minimize transport cost quantified by $c$ along the feature (ii) while minimizing the distortion along the structural information quantified by $c_{\tilde{\mathcal{X}}}$ and $c_{\tilde{\mathcal{Y}}}$. In § 5.2, we use fused couplings to enhance the translation between different modalities of cells.

**Why Does Adding a Fused Term Provide Benefits?** Generally, unlike the linear EOT problem (LEOT), the solution $\pi^{\star}_\varepsilon$ to the quadratic EOT problem (QEOT) is not unique. This is where adding a fused term helps, as it introduces more constraints. Intuitively, the fused term on additional features $\mathbf{u}, \mathbf{v}$ allows us to 'select' an optimal GW coupling: we trade some GW optimality

by choosing a coupling that not only minimizes the distortion of the structural information $\mathbf{x}, \mathbf{y}$, but also reduces the displacement cost, quantified by $c$, along the features $\mathbf{u}, \mathbf{v}$. Empirically, this significantly mitigates the issue of non-uniqueness in (pure) GW couplings. In § 5.2, we demonstrate that it enhances the stability of our procedure, particularly for single-cell data, as illustrated in Fig. 4 and Fig. 18. On a related note, but independently of the addition of the fused term, we investigate in App. E.2 other strategies to mitigate the non-uniqueness of the GW coupling. Notably, we show that using the geodesic distance on the data manifold as costs, instead of the squared Euclidean distance, helps address this issue on single-cell data. As an alternative, we also introduce an initialization scheme to bias the discrete GW solver to a specific solution based on the neural coupling obtained in the previous iteration (App. E.2). Yet, this approach renders GENOT non-simulation free.

## 3.4 One algorithm to flexibly switch between neural OT problems

**Algorithm 1** U-GENOT. Skip teal steps for GENOT.

1: **Require parameters:** Batch size $n$; entropic regularization $\varepsilon$; unbalancedness parameter $\tau$; *linear, quadratic* or *fused* discrete OT solver $\text{Solver}_{\varepsilon, \tau}$.
2: **Require network:** Time-dependent conditional velocity field $v_{t,\theta}(\cdot|\cdot) : \mathbb{R}^q \times \mathbb{R}^p \to \mathbb{R}^q$; re-weighting functions $\eta_\theta : \mathbb{R}^p \to \mathbb{R}, \xi_\theta : \mathbb{R}^q \to \mathbb{R}$.
3: **for** $t = 1, \ldots, T_{\text{iter}}$ **do**
4:     Sample $X_1, \ldots, X_n \sim \mu$ and $Y_1, \ldots, Y_n \sim \nu$.
5:     $\mathbf{P}_{\varepsilon, \tau} \leftarrow \text{Solver}_{\varepsilon, \tau}(\{X_i\}_{i=1}^n, \{Y_i\}_{i=1}^n) \in \mathbb{R}_+^{n \times n}$.
6:     $\mathbf{a} \leftarrow \mathbf{P}_{\varepsilon, \tau} 1_n$ and $\mathbf{b} \leftarrow \mathbf{P}_{\varepsilon, \tau}^\top 1_n$.
7:     Sample $(i_1, j_1), \ldots, (i_n, j_n) \sim \mathbf{P}_{\varepsilon, \tau}$.
8:     Sample $Z_1, \ldots, Z_n \sim \rho$ and $t_1, \ldots, t_n \sim \mathcal{U}([0, 1])$.
9:     $\mathcal{L}(\theta) \leftarrow \sum_k \|v_{t,\theta}([Z_k, Y_{j_k}]_t | X_{i_k}) - (Y_{j_k} - Z_k)\|_2^2$
10:       $+ \sum_k (\eta_\theta(X_k) - n\mathbf{a}_k)^2 + (\xi_\theta(Y_k) - n\mathbf{b}_k)^2$.
11:     $\theta \leftarrow \text{Update}(\theta, \frac{1}{n} \nabla \mathcal{L}(\theta))$.
12: **end for**

Single-cell genomics data is inherently noisy, due to biases incurred by sequencing protocols and the high sparsity of the measurements [34]. Thus, it is indispensable to flexibly switch between configurations of the problem setup. For example, it is often not clear whether the mass conservation constraint should be actually loosened (for example whether to allow for modeling cell death), or whether incorporating prior information for trajectory inference via the quadratic term (e.g. with spatial information [39] or lineage barcoding [42]) is beneficial. Similarly, trying different costs is crucial in single-cell genomics ([17, 35, 39]). The GENOT formulation offers this flexibility by introducing only minor changes into the algorithm. We detail our procedure in Alg. 1, showing the additional steps to switch from GENOT to U-GENOT in teal. Limitations are discussed in § A.

## 4 Related work and Discussions

**Static Neural EOT.** While GENOT is the first model to consider the quadratic (and fused) EOT setting, various methods have been proposed in the linear EOT scenario. The first class of methods solves (LEOT)'s dual. While some of them [27, 73] do not allow direct sampling from to $\pi_\varepsilon^\star$, Daniels et al. [15] and Mokrov et al. [55] model $\pi_\varepsilon^\star(\cdot|\mathbf{x})$. However, these methods might be costly and unstable, as they rely on Langevin sampling during training or inference.

**Dynamic Neural EOT.** The second class of linear EOT solvers builds upon the link between (LEOT) and the SB problem [10, 16, 29, 87]. Although they operate primarily in the balanced setting, Gazdieva et al. [26] recently considered the unbalanced one. As simulations are costly in this setting, recent works consider simulation-free training via bridge matching [41, 46, 61, 76, 84, 85, 96]. While Tong et al. [84, 85] use mini-batch OT, our method is fundamentally different, as we do not build upon the link between EOT and SB. We only use flow matching as a powerful black box to learn a flow from the noise $\rho$ to each conditional $\pi_\varepsilon^\star(\cdot|\mathbf{x})$. Therefore, our approach allows for more flexibility. First (i), the abovementioned methods assume that $\mathcal{X}, \mathcal{Y} \subset \mathbb{R}^d$, since they learn a velocity field (or a drift) by directly bridging $\mathbf{x} \in \mathcal{X}, \mathbf{y} \in \mathcal{Y}$. Then, they map $\mu$ to $\nu$ with the induced (stochastic) flow. On the other hand, conditionally to each $\mathbf{x} \in \mathcal{X} \subset \mathbb{R}^d$, we learn a velocity field $v_{t,\theta}(\cdot|\mathbf{x}) : \mathbb{R}^q \to \mathbb{R}^q$ in the *target space*, by building paths between noise $\mathbf{z} \in \mathbb{R}^q$ and $\mathbf{y} \in \mathcal{Y} \subset \mathbb{R}^q$. Then, we map $\rho$ to each $\pi_\varepsilon^\star(\cdot|\mathbf{x})$ with the flows $\phi_1(\cdot|\mathbf{x})$, and recover $\nu = \phi_1(\cdot|\cdot)\sharp(\rho \otimes \mu)$. Second (ii), they can only approximate the SB for $c(\mathbf{x}, \mathbf{y}) = \|\mathbf{x} - \mathbf{y}\|_2^2$. This results from [5, Prop. 2]. In contrast, our method handles any cost, as shown in Fig 5. Third (iii), as they learn a single flow directly transporting $\mu$ to $\nu$, they do not approximate the conditional densities $\pi_\varepsilon^\star(\mathbf{y}|\mathbf{x})$. Similarly to Tong et al. [85], Pooladian et al. [64] couple samples from $\mu$ and $\nu$, but they only model deterministic maps and assume $\mu = \mathcal{N}(0, I_d)$. Finally, Bortoli et al. [5] recently proposed an augmented bridge matching

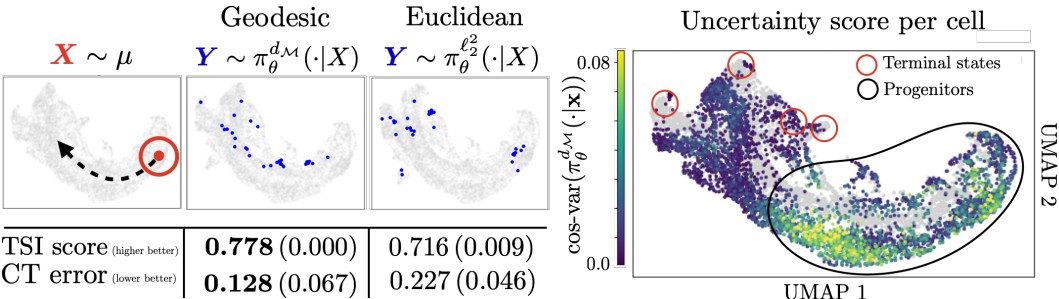

Figure 2: **Left:** Source cell from the early time points (top left) and samples of the conditional distributions of the EOT coupling learned with GENOT for the geodesic cost $d_{\mathcal{M}}$ (middle) and the $\ell_2^2$ cost (right) projected onto a UMAP [53], along with biological assessment of the learnt dynamics (TSI score, CT error § C.2, Fig. 9). **Right:** UMAP colored according to the uncertainty score $\text{cos-var}(\pi_\theta^{d_{\mathcal{M}}}(\cdot|X))$ of each source cell $\mathbf{x}$. Target cells are colored in gray.

procedure that preserves couplings. However, it still requires $\mathcal{X}, \mathcal{Y} \subset \mathbb{R}^d$ and does not approximate conditional densities, limiting its applicability in single-cell genomics tasks.

**Mini-batches and Biases** Quantifying non-asymptotically the bias resulting from minimizing a sample-based GENOT loss, and *not* its population value, is challenging. The OT curse of dimensionality [91] has been discussed in generative models relying on mini-batch couplings [22, 27, 84, 85]. Yet, our goal is *not* to model *non-regularized* OT, such as a deterministic Monge map, or a Benamou-Brenier vector field. We explicitly target the *entropy-regularized* OT coupling. Thus, using $\varepsilon \gg 0$ helps to mitigate the curse of dimensionality because of two qualitative factors:

(i) **Statistical.** For both linear and quadratic OT, all statistical recovery rates that relate to entropic costs [28, 95], maps [63, 66], or couplings [90], have a far more favorable regime, with a parametric rate in $\varepsilon > 0$ that dodges the curse of dimensionality.

(ii) **Computational.** While the benefits of employing a large enough $\varepsilon$ in Sinkhorn's algorithm are widely known, Rioux et al. [67] have recently shown that as $\varepsilon$ increases, the quadratic OT problem (QEOT) becomes convex, making discrete solvers faster and more reliable.

To demonstrate this aspect, we empirically study the influence of the batch size on GENOT, using a recent benchmark [30], with known true linear EOT coupling $\pi_\varepsilon^\star$, see Fig. 6.

## 5 Experiments

While there is no evidence that cells exactly evolve according to an entropic OT plan, leveraging OT is an established way to realign cells. We also evaluate the performance on simulated data in settings which closed-form solutions of the EOT coupling are available for. Metrics and datasets are discussed in App. C and App. D, respectively. Further experimental details or results are reported in App. E. Setups for baselines are listed in App. F. Implementation details can be found in App. G. We denote by **GENOT-L** the GENOT model for solving the linear problem (LEOT), **GENOT-Q** for the quadratic (QEOT) and **GENOT-F** for the fused (EFGW) one. When considering the **unbalanced** counterparts, we add the **prefix U**. Moreover, when using the **conditional mean** of a GENOT model, we add the **suffix CM**. We denote our learned neural couplings by $\pi_\theta$.

### 5.1 GENOT-L for modeling cell trajectories

**GENOT-L on simulated data** We show that GENOT accurately computes EOT couplings using a recent benchmark [30]. Tab. 1 shows that GENOT-L outperforms all baselines considered. These results are of particular significance since most of the baselines are tailored towards the specific task of solving (i) the linear EOT problem (LEOT), (ii) in the balanced setting (iii) with the $\ell_2^2$ cost. In contrast, the contributions of GENOT go *beyond* this setting, see **N2**, **N3**, and **N4**. Going beyond the balanced problem (**N3**), we show that U-GENOT learns meaningful unbalanced EOT couplings between Gaussians [36] (11), which we visualize in Fig. 10.

**GENOT-L learns the evolution of cells.** Trajectory inference is a prominent task in single-cell biology. It is key to understand the development of cells from less mature to more mature cell

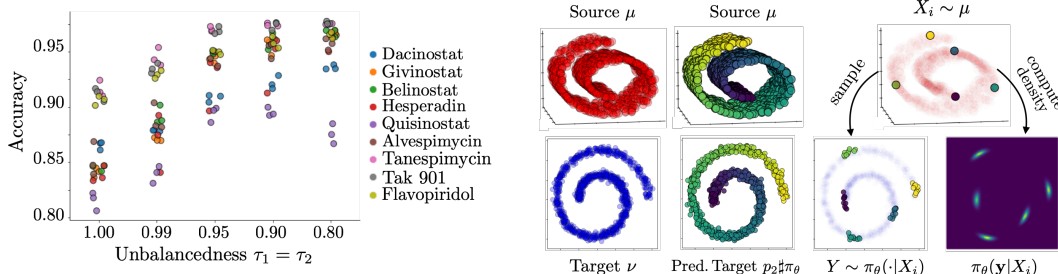

Figure 3: **Left**: Accuracy of cellular response predictions of U-GENOT-L for cancer drugs with varying unbalancedness parameter $\tau = \tau_1 = \tau_2$. *Smaller* $\tau$ implies *more* unbalancedness (3 runs per $\tau$). **Right:** Mapping a Swiss roll in $\mathbb{R}^3$ ($\mu$) to a spiral in $\mathbb{R}^2$ ($\nu$) with GENOT-Q. Center: Color code tracks where samples from $\mu$ (top) are mapped to (bottom). Right column: samples from $\mu$ (top) and the corresponding conditionals, along with conditional density estimates. The learned QEOT coupling minimizes the distortion: Points close in support of $\mu$ generate points close in support of $\nu$.

states. In the following, we consider a dataset capturing gene expression of the developing mouse pancreas at embryonic days 14.5 (source) and 15.5 (target) [2]. Using the geodesic distance on the data manifold $d_\mathcal{M}$ as cost (§ 3.1) improves the learnt vector field compared to using the $\ell_2^2$ cost (TSI score in Fig.2, § C.2). The CT error measures to what extent cells follow the manifold (§ C.2, Fig. 9), which is visually confirmed by samples of the conditional distribution (Fig.2). These results support the need for a flexible choice of the cost function (**N2**). As cells are known to evolve stochastically, we leverage GENOT's capability to generate samples from the conditional distribution to model non-deterministic trajectories (**N1**). We follow Gayoso et al. [25] for assessing the uncertainty of cell trajectories by computing $\text{cos-var}(\pi_\theta(\cdot|\mathbf{x})) = \text{Var}_{Y \sim \pi_\theta(\cdot|\mathbf{x})}[\text{cos-sim}(Y, \mathbb{E}_{Y \sim \pi_\theta(\cdot|\mathbf{x})}[Y])]$ (§ C.1). We expect high uncertainty in cell types in early developmental stages and low variance in mature cell types. Indeed, Fig. 2 and Fig. 8 show that the computed uncertainty is biologically meaningful.

**U-GENOT-L predicts single-cell responses to perturbations.** Neural OT has been successfully applied to model cellular responses to perturbations using deterministic maps [7]. GENOT has the comparative advantage to model conditional distributions, allowing for uncertainty quantification. We consider single cell RNA-seq data measuring the response of cells to 163 cancer drugs [78]. Each drug has been applied to a population of cells that can be partitioned into 3 different cell types. The source (resp. target) distribution consists of cells before (resp. after) drug application. While there is no ground truth in the matching between unperturbed and perturbed cells due to the destructive nature of sequencing technologies (hence the need for realignment), we know which unperturbed subset of cells is supposed to be mapped to which perturbed subset of cells. We use this to define an accuracy metric (see App. C.2), while we assess the uncertainty (**N1**) of the prediction using cos-var. Fig. 12 shows that for 117 out of 163 drugs, the model is perfectly calibrated (see App. C.1), while it yields a negative correlation between error and uncertainty only for one drug. We can improve the generation of cells by accounting for class imbalances between different cell types using U-GENOT. These imbalances might occur due to biases in the experimental setup or due to cell death [78]. Indeed, Fig. 3 shows that allowing for mass variation increases accuracy for nine different cancer drugs, selected as they are known to have a strong effect. Fig. 13 and 14 confirm the results visually (**N3**).

### 5.2 GENOT for Quadratic EOT Problems

As outlined in § 1, quadratic OT problems are of particular interest in single-cell genomics (**N4**). Here, we focus on the task of aligning different cellular modalities, which is necessary as most sequencing technologies are limited to measuring one modality [34]. Yet, there exist few protocols measuring two modalities simultaneously, thus allowing us to assess the alignment as the ground truth matching is known. We leverage this to assess GENOT-Q's performance on the task of translating cells from ATAC-seq to RNA-seq, and also propose a novel way to improve the alignment using GENOT-F. Before, we demonstrate the task of mapping across incomparable spaces on simulated data.

**GENOT-Q and U-GENOT-Q on simulated data.** To visualize the problem of quadratic OT, we transport a Swiss role in $\mathbb{R}^3$ to a spiral in $\mathbb{R}^2$. Fig. 3 shows that GENOT-Q successfully mimics an isometric alignment. Here, we set $\varepsilon = 0.01$ and investigate its influence in more detail in Fig. 15. To quantitatively assess the performance of GENOT-Q and U-GENOT-Q, we compute (unbalanced)

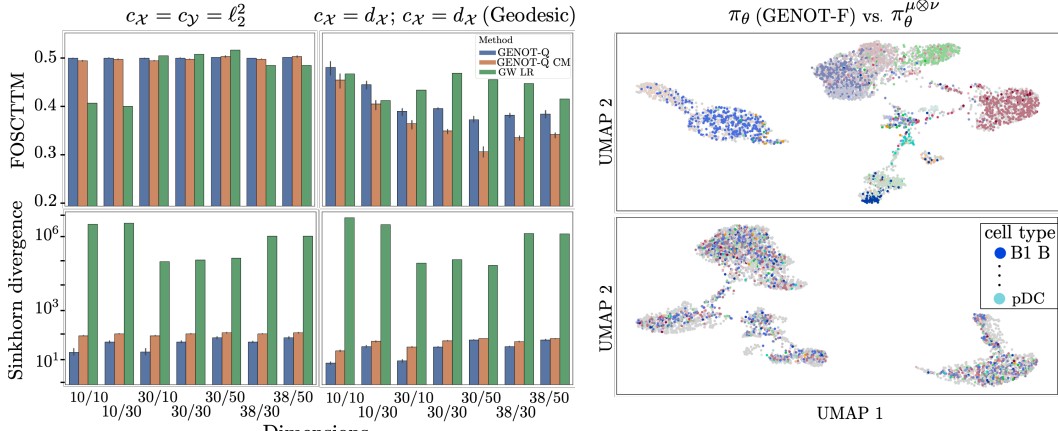

Figure 4: **Left**: Benchmark of GENOT-Q models against discrete GW (GW-LR, App. F) on translating cells between ATAC space of dim. $d_1$ and RNA space of dim. $d_2$, with performance measured with FOSCTTM score (App. C.2) and Sinkhorn divergence between target and predicted target distribution. (left) intra-domain costs $c_{\mathcal{X}} = c_{\mathcal{Y}} = \ell_2^2$, (right) geodesic distances $c_{\mathcal{X}} = d_{\mathcal{X}}$ and $c_{\mathcal{Y}} = d_{\mathcal{Y}}$. We show mean and std across 3 runs. **Right:** Top: UMAP of transported cells with GENOT-F (colored by cell type) and cells in the target distribution (gray). Cells of the same cell type generate cells which cluster together in RNA space. Bottom: UMAP of transported cells with a GENOT model trained on batch-wise independent couplings, thus not using OT, generating cells which are randomly mixed.

entropic OT plans between Gaussian distributions, which the closed form EOT plan is known for [43]. Fig.19 and Fig.20 show that GENOT-Q is able to meaningfully learn (un-)balanced entropic EOT plans for lower dimensions, but its performance decreases with increasing dimensionality.

**GENOT-Q translates modalities of single cells.** We use GENOT-Q to translate ATAC measurements (source) to gene expression space (target) on a bone marrow dataset [51]. As both modalities were measured in the same cell, the true match of each cell is known for this specific dataset. We compare GENOT-Q with discrete OT extended out-of-sample with linear regression (GW-LR, see F.2). We assess the performance using (i) the FOSCTTM ("Fractions of Samples Closer to the True Match", see C.2) that measures the optimality of the learned coupling, and (ii) the Sinkhorn divergence [23] between the predicted target and the target to assess distributional fitting. As in § 5.1, we leverage GENOT's flexibility and use as intra-domains costs the geodesic distances on the source and target domain, namely $c_{\mathcal{X}} = d_{\mathcal{X}}$, $c_{\mathcal{Y}} = d_{\mathcal{Y}}$ (**N2**). We also estimate the EOT coupling for the $\ell_2^2$ cost for comparison. Results are shown in Fig. 3. Regarding the FOSCTTM score, we see that (i) using geodesic costs is crucial in high dimensions and (ii) GW-LR is competitive in low dimensions but not for higher ones. Regarding distributional fitting, GENOT models outperform by orders of magnitude.

**GENOT-F improves modality translation.** To enhance the performances attained by purely quadratic OT-based models, we introduce a novel method for translating between ATAC and RNA. We extend the idea of translating between cell modalitiies proposed by Demetci et al. [17] to the fused setting: We approximate RNA data from ATAC measurements using gene activity [79], and we further process the data using a conditional VAE [48] to reduce batch effects. This way, we construct a joint space $\Omega$. Following the notations in § 3.3, RNA (source $\mathbf{x} \in \tilde{\mathcal{X}}$) and ATAC (target $\mathbf{y} \in \tilde{\mathcal{Y}}$) carry the structural information, while features $\mathbf{u} \in \Omega$ and $\mathbf{v} \in \Omega$ are obtained from the VAE embedding. Fig. 21 shows that the additional linear penalty on the feature term helps to obtain a better alignment compared to GENOT-Q. Fig. 4 visualizes the learned fused coupling compared to GENOT trained with batch-wise independent couplings. When aligning multiple cell modalities, the proportions of cell types in the two modalities often differ (**N3**). We simulate this setting by removing cells belonging to certain cell types Tab. 4 shows that U-GENOT-F preserves high accuracy while learning meaningful rescaling functions.

**Conclusion.** Motivated by applications in single-cell genomics, we introduce GENOT, a versatile neural OT framework to learn cost-efficient stochastic maps, within the same space or across incomparable spaces. GENOT can be used for learning EOT couplings with any cost function, and allows for loosening mass constraints if desired.

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

# Appendix

## A   Limitations

While our study opens up the use of neural optimal transport to a wide range of tasks in single-cell genomics, there are limitations of our work:

- The solution to the entropic (Fused) Gromov-Wasserstein problem is not necessarily unique. As GENOT-Q and GENOT-F leverage discrete solvers in each iteration, the orientation of the solution across iterations is not maintained. To prevent learning a mixture of solutions, we propose to use a fused term whenever possible, make use of different cost functions if helpful, or using GENOT with our proposed initialisation scheme (§ E.2). While the latter can always be applied, its major drawback is the costly training as a neural ODE has to be solved at every iteration.

- When learning unbalanced EOT plans, choosing a value for the hyperparameters $(\tau_a, \tau_b)$ is not evident. As discussed in Séjourné et al. [75], the choice of these parameters is mostly empirical.

- While GENOT is motivated by the extensive use of optimal transport in single-cell genomics, there is no evidence that cells evolve exactly according to an entropic OT plan.

- Evaluations of learnt transport plans for tasks in single-cell genomics experiments are task-specific, and do not necessarily reflect to what extent the learnt coupling is an entropic OT coupling. In effect, we rely on prior biological knowledge (CT error, Fig. 2) or on common downstream trajectory inference methods in single-cell genomics to evaluate the learnt velocity field (TSI score, Fig. 2) when computing EOT plans for trajectory inference. Similarly, we can only assess learnt EOT plan on a cluster level, and not a single data point level when studying cellular perturbations (3). For the task of modality translation (Fig. 4), we know the ground-truth coupling. Yet, the ground-truth coupling is deterministic and one-to-one, and hence is not a valid solution of an entropy-regularized Gromov-Wasserstein OT plan with $\varepsilon > 0$.

## B   Propositions and Proofs

### B.1   Existence of EOT couplings

In this section, we point out that a solution to the entropic fused GW problem (EFGW), and its unbalanced counterpart, always exists. Note that for $\alpha = 0$, we recover the existence for the pure entropic GW (EFGW) and its unbalanced counterpart (UQEOT) problems. The arguments are adapted from [88, Prop. 2]

**Balanced setting:** We first recall the fused Gromov-Wasserstein problem below.

$$\min_{\pi \in \Pi(\mu,\nu)} \int_{((\Omega \times \tilde{\mathcal{X}}) \times (\Omega \times \tilde{\mathcal{Y}}))^2} Q^c_{c_{\tilde{\mathcal{X}}}, c_{\tilde{\mathcal{Y}}}} \mathrm{d}(\pi \otimes \pi) + \varepsilon \mathrm{KL}(\pi | \mu \otimes \nu) \tag{6}$$

where $Q^c_{c_{\tilde{\mathcal{X}}}, c_{\tilde{\mathcal{Y}}}} ((\mathbf{u}, \mathbf{x}), (\mathbf{v}, \mathbf{y}), \mathbf{x}', \mathbf{y}') := (1 - \alpha)c(\mathbf{u}, \mathbf{v}) + \alpha |c_{\tilde{\mathcal{X}}}(\mathbf{x}, \mathbf{x}') - c_{\tilde{\mathcal{Y}}}(\mathbf{y}, \mathbf{y}')|^2$. We first need to show that a solution $\pi^\star_\varepsilon$ to this problem always exists. We assume that the intra-domain $c_{\mathcal{X}}, c_{\mathcal{Y}}$ costs, and the inter-domain cost $c$ are l.s.c, an assumption that matches all our experimental settings. From [88, Lemma 3], this implies that the functional:

$$\pi \in \Pi(\mu,\nu) \mapsto \int_{((\Omega \times \tilde{\mathcal{X}}) \times (\Omega \times \tilde{\mathcal{Y}}))^2} Q^c_{c_{\tilde{\mathcal{X}}}, c_{\tilde{\mathcal{Y}}}} \mathrm{d}(\pi \otimes \pi) \tag{7}$$

is weakly l.s.c, i.e. l.s.c for the convergence in law. Moreover, from [23, Prop. 8], $\pi \mapsto \mathrm{KL}(\pi | \mu \otimes \nu)$ is also weakly l.s.c. Therefore, the objective function of the fused Gromov-Wasserstein problem:

$$\mathcal{F} : \pi \in \Pi(\mu,\nu) \mapsto \int_{((\Omega \times \tilde{\mathcal{X}}) \times (\Omega \times \tilde{\mathcal{Y}}))^2} Q^c_{c_{\tilde{\mathcal{X}}}, c_{\tilde{\mathcal{Y}}}} \mathrm{d}(\pi \otimes \pi) + \varepsilon \mathrm{KL}(\pi | \mu \otimes \nu) \tag{8}$$

is weakly l.s.c, as a sum of weakly l.s.c functionals. Then, since we assume that $\mathcal{X}$ and $\mathcal{Y}$ are compact, they are polish spaces, so from [68, Theorem 1.7], $\Pi(\mu, \nu)$ is weakly compact. Therefore, from Weierstrass Theorem, $\mathcal{F}$ attains its minimum on $\Pi(\mu, \nu)$, i.e. it exists $\pi^\star_\varepsilon$ solution to this problem.

**Unbalanced setting:** The proof is very similar. We first recall the unbalanced fused Gromov-Wasserstein problem.

$$\min_{\pi \in \mathcal{M}^+(\mathcal{X} \times \mathcal{Y})} \int_{((\Omega \times \tilde{\mathcal{X}}) \times (\Omega \times \tilde{\mathcal{Y}}))^2} Q^c_{c_{\tilde{\mathcal{X}}}, c_{\tilde{\mathcal{Y}}}} d(\pi \otimes \pi) + \varepsilon \mathrm{KL}(\pi|\mu \otimes \nu) + \lambda_1 \mathrm{KL}^{\otimes}(\pi_1|\mu) + \lambda_2 \mathrm{KL}^{\otimes}(\pi_2|\mu) \tag{9}$$

where $\mathrm{KL}^{\otimes}(\alpha|\beta) = \mathrm{KL}(\alpha \otimes \alpha|\beta \otimes \beta)$. From [81, Prop. 8], $\mathrm{KL}^{\otimes}$ is weakly l.s.c, so is $\pi \mapsto \mathrm{KL}^{\otimes}(\pi_1|\mu)$ and $\pi \mapsto \mathrm{KL}^{\otimes}(\pi_2|\nu)$. Therefore, the objective function of the unbalanced fused Gromov-Wasserstein problem is also weakly l.s.c. as a sum of weakly l.s.c. functions. Moreover, since $\mathcal{X}$ and $\mathcal{Y}$ are assumed to be compact, $\mathcal{X} \times \mathcal{Y}$ is also compact and the Banach-Alaoglu provides the compactness of $\mathcal{M}^+(\mathcal{X} \times \mathcal{Y})$. Then, from Weierstrass Theorem, similarly to the balanced setting, we get the existence of a solution $\pi^{\star}_{\varepsilon, \tau}$ to this problem.

## B.2  Proofs of § 3.2

**Proposition B.1** (Re-Balancing the unbalanced problems.). *Let $\pi^{\star}_{\varepsilon, \tau}$ be an unbalanced EOT coupling, solution of* (ULEOT) *or* (UQEOT) *between $\mu \in \mathcal{M}^+(\mathcal{X})$ and $\nu \in \mathcal{M}^+(\mathcal{Y})$. We note $\tilde{\mu} = p_1 \sharp \pi^{\star}_{\varepsilon, \tau}$ and $\tilde{\nu} = p_2 \sharp \pi^{\star}_{\varepsilon, \tau}$ its marginals. Then, in both cases, $\tilde{\mu}$ (resp. $\tilde{\nu}$) has a density w.r.t. $\mu$ (resp. $\nu$). That is, there exist two re-weighting functions, one on each space, $\eta : \mathcal{X} \to \mathbb{R}^+$ and $\xi : \mathcal{Y} \to \mathbb{R}^+$ s.t. $\tilde{\mu} = \eta \cdot \mu$ and $\tilde{\nu} = \xi \cdot \nu$. Furthermore, $\tilde{\mu}$ and $\tilde{\nu}$ have the same total mass and*

1. *(Linear) $\pi^{\star}_{\varepsilon, \tau}$ solves the balanced problem* (LEOT) *between $\tilde{\mu}$ and $\tilde{\nu}$ with the same $\varepsilon$.*

2. *(Quadratic) Provided that $c_{\mathcal{X}}, c_{\mathcal{Y}}$ (or $-c_{\mathcal{X}}, -c_{\mathcal{Y}}$) are CPD kernels (see Def. B.2), $\pi^{\star}_{\varepsilon, \tau}$ solves the balanced problem* (QEOT) *between $\tilde{\mu}$ and $\tilde{\nu}$ with $\varepsilon' = m(\pi^{\star}_{\varepsilon, \tau}) \varepsilon$, where $m(\pi^{\star}_{\varepsilon, \tau}) = \pi^{\star}_{\varepsilon, \tau}(\mathcal{X} \times \mathcal{Y})$.*

We first remind the definition of a conditionally positive kernel, which is involved in point 2. of Prop. 3.1.

**Definition B.2.** A kernel $k : \mathbb{R}^d \times \mathbb{R}^d \to \mathbb{R}$ is conditionally positive if it is symmetric and for any $\mathbf{x}_1, ..., \mathbf{x}_n \in \mathbb{R}^d$ and $\mathbf{a} \in \mathbb{R}^n$ s.t. $\mathbf{a}^{\top} \mathbf{1}_n = 0$, one has

$$\sum_{i,j=1}^n a_i a_j \, k(\mathbf{x}_i, \mathbf{x}_j) \geq 0$$

Conditionally positive kernels include all positive kernels, such $k(\mathbf{x}, \mathbf{y}) = \langle \mathbf{x}, \mathbf{y} \rangle$ or $k_{\gamma}(\mathbf{x}, \mathbf{y}) = \exp(-\frac{1}{\gamma} \|\mathbf{x} - \mathbf{y}\|_2^2)$, but also the negative squared Euclidean distance $k(\mathbf{x}, \mathbf{y}) = -\|\mathbf{x} - \mathbf{y}\|_2^2$.

*Proof of 3.1.* **Step 1: Re-weightings.** We first show that for $\pi^{\star}_{\varepsilon, \tau}$ solution of (LEOT) or (QEOT), it exists $\eta, \xi : \mathbb{R}^d \to \mathbb{R}^+$ s.t. $\tilde{\mu} = p_1 \sharp \pi^{\star}_{\varepsilon, \tau} = \eta \cdot \mu$ and $\tilde{\nu} = p_1 \sharp \pi^{\star}_{\varepsilon, \tau} = \xi \cdot \nu$.

We start with the linear case and consider $\pi^{\star}_{\varepsilon, \tau}$ solution of (LEOT). The result follows from duality. Indeed, from [80, Prop. 2], one has the existence of the so-called entropic potentials $f^{\star} \in \mathcal{C}(\mathcal{X})$ and $g^{\star} \in C(\mathcal{Y})$ s.t.

$$\frac{d\pi^{\star}_{\varepsilon, \tau}}{d(\mu \otimes \nu)}(\mathbf{x}, \mathbf{y}) = \exp\left(\frac{f^{\star}(\mathbf{x}) + g^{\star}(\mathbf{y}) - c(\mathbf{x}, \mathbf{y})}{\varepsilon}\right) \tag{10}$$

Therefore, $\tilde{\mu} = \eta \cdot \mu$ and $\tilde{\nu} = \xi \cdot \nu$ where $\eta : \mathcal{X} \to \mathbb{R}^+$ and $\xi : \mathcal{Y} \to \mathbb{R}^+$ are defined by:

$$\eta(\mathbf{x}) = \int_{\mathcal{Y}} \exp\left(\frac{f^{\star}(\mathbf{x}) + g^{\star}(\mathbf{y}) - c(\mathbf{x}, \mathbf{y})}{\varepsilon}\right) d\nu(\mathbf{y})$$

$$\text{and} \quad \xi(\mathbf{y}) = \int_{\mathcal{X}} \exp\left(\frac{f^{\star}(\mathbf{x}) + g^{\star}(\mathbf{y}) - c(\mathbf{x}, \mathbf{y})}{\varepsilon}\right) d\mu(\mathbf{x}). \tag{11}$$

We now handle the quadratic case and consider $\pi^{\star}_{\varepsilon, \tau}$ solution of (UQEOT). We remind that the (UQEOT) problem between $\mu$ and $\nu$ reads:

$$\min_{\pi \in \mathcal{M}^+(\mathcal{X} \times \mathcal{Y})} \int_{(\mathcal{X} \times \mathcal{Y})^2} |c_{\mathcal{X}}(\mathbf{x}, \mathbf{x}') - c_{\mathcal{Y}}(\mathbf{y}, \mathbf{y}')|^2 \, d\pi(\mathbf{x}, \mathbf{y}) \, d\pi(\mathbf{x}', \mathbf{y}')$$

$$+ \varepsilon \mathrm{KL}^{\otimes}(\pi|\mu \otimes \nu) + \lambda_1 \mathrm{KL}^{\otimes}(\pi_1|\mu) + \lambda_2 \mathrm{KL}^{\otimes}(\pi_2|\nu),$$

For $\pi \in \mathcal{M}^+(\mathcal{X} \times \mathcal{Y})$, the quadratic relative entropy $\mathrm{KL}^\otimes(\pi|\mu \otimes \mu) = \mathrm{KL}(\pi \otimes \pi|(\mu \otimes \nu) \otimes (\mu \otimes \nu))$ is finite i.f.f. $\pi \otimes \pi$ has a density w.r.t. $(\mu \otimes \nu) \otimes (\mu \otimes \nu)$, which implies that $\pi$ has a density w.r.t. $\mu \otimes \nu$. Therefore, one can reformulate (UQEOT) as:

$$
\min_{h \in L_1^+(\mathcal{X} \times \mathcal{Y})} \int_{(\mathcal{X} \times \mathcal{Y})^2} (c_\mathcal{X}(\mathbf{x}, \mathbf{x}) - c_\mathcal{Y}(\mathbf{y}, \mathbf{y}))^2 h(\mathbf{x}, \mathbf{y}) h(\mathbf{x}', \mathbf{y}') \, \mathrm{d}\mu(\mathbf{x}) \, \mathrm{d}\mu(\mathbf{x}') \, \mathrm{d}\nu(\mathbf{y}) \, \mathrm{d}\nu(\mathbf{y}')
$$
$$
+ \varepsilon \mathrm{KL}(h|\mu \otimes \nu) + \tau_1 \mathrm{KL}(h_1|\mu) + \tau_2 \mathrm{KL}(h_2|\nu)
\tag{12}
$$

where we extend the KL divergence for densities: $\mathrm{KL}(r|\gamma) = \int (r \log(r) + r - 1) \, \mathrm{d}\gamma$, and define the marginal densities $h_1 : \mathbf{x} \in \mathcal{X} \mapsto \int_\mathcal{Y} h(\mathbf{x}, \mathbf{y}) \, \mathrm{d}\nu(\mathbf{y})$ and $h_2 : \mathbf{y} \in \mathcal{Y} \mapsto \int_\mathcal{X} h(\mathbf{x}, \mathbf{y}) \, \mathrm{d}\nu(\mathbf{y})$. As a result, it exists $h^\star \in L_1^+(\mathcal{X} \times \mathcal{Y})$ s.t. $\pi^\star_{\varepsilon,\tau} = h^\star \cdot \mu \otimes \nu$. It follows that $\tilde{\mu} = \eta \cdot \mu$ and $\tilde{\nu} = \xi \cdot \nu$ with $\eta = h_1^\star$ and $\xi = h_2^\star$.

*Remark* B.3. Note that in both cases, since $\mathrm{d}\tilde{\mu}(\mathbf{x}) = \eta(\mathbf{x}) \, \mathrm{d}\mu(\mathbf{x})$ and $\mathrm{d}\tilde{\nu}(\mathbf{y}) = \xi(\mathbf{y}) \, \mathrm{d}\nu(\mathbf{y})$, the equality of mass of $\tilde{\mu}$ and $\tilde{\nu}$ yields $\mathbb{E}_{X \sim \mu}[\eta(X)] = \mathbb{E}_{Y \sim \nu}[\xi(Y)]$.

**Step 2: Optimality in the balanced problem for the linear case.** We now prove **point 1**, stating that if $\pi^\star_{\varepsilon,\tau}$ solves problem ULEOT between $\mu$ and $\nu$, then it solves problem (LEOT) between $\tilde{\mu}$ and $\tilde{\nu}$ for the same entropic regularization strength $\varepsilon$. It also follows from duality. We remind that that thanks to [80, Prop. 2]

$$
\frac{\mathrm{d}\pi^\star_{\varepsilon,\tau}}{\mathrm{d}(\mu \otimes \nu)}(\mathbf{x}, \mathbf{y}) = \exp\left( \frac{f^\star(\mathbf{x}) + g^\star(\mathbf{y}) - c(\mathbf{x}, \mathbf{y})}{\varepsilon} \right)
\tag{13}
$$

with $f^\star \in \mathcal{C}(\mathcal{X})$ and $g^\star \in C(\mathcal{Y})$. Moreover, by [59, Theorem 4.2], such a decomposition is equivalent to the optimality in problem (LEOT). Therefore, $\pi^\star_{\varepsilon,\tau}$ is solves problem LEOT between its marginals $\tilde{\mu}$ and $\tilde{\nu}$, i.e.

$$
\pi^\star_{\varepsilon,\tau} = \arg\min_{\pi \in \Pi(\tilde{\mu},\tilde{\nu})} \int_{\mathcal{X} \times \mathcal{Y}} c(\mathbf{x}, \mathbf{y}) \, \mathrm{d}\pi(\mathbf{x}, \mathbf{y}) + \varepsilon \mathrm{KL}(\pi|\mu \otimes \nu).
\tag{14}
$$

**Step 3: Optimality in the balanced problem for the quadratic case.** We now prove **point 2**, stating that, provided that the costs $c_\mathcal{X}$ and $c_\mathcal{Y}$ are conditionally positive (or conditionally negative), if $\pi^\star_{\varepsilon,\tau}$ solves problem (UQEOT) between $\mu$ and $\nu$, then it actually solves problem (QEOT) between $\tilde{\mu}$ and $\tilde{\nu}$ for the entropic regularization strength $\varepsilon' = m(\pi^\star_{\varepsilon,\tau}) \varepsilon$. We first define the functional:

$$
F : (\gamma, \pi) \in \mathcal{M}^+(\mathcal{X} \times \mathcal{Y})^2 \mapsto \int_{(\mathcal{X} \times \mathcal{Y})^2} |c_\mathcal{X}(\mathbf{x}, \mathbf{x}') - c_\mathcal{Y}(\mathbf{y}, \mathbf{y}')|^2 \, \mathrm{d}\pi(\mathbf{x}, \mathbf{y}) \, \mathrm{d}\gamma(\mathbf{x}', \mathbf{y}')
$$
$$
+ \varepsilon \mathrm{KL}(\pi \otimes \gamma|(\mu \otimes \nu)^2) + \lambda_1 \mathrm{KL}(\pi_1 \otimes \gamma_1|\mu \times \mu) + \lambda_2 \mathrm{KL}^\otimes(\pi_2 \otimes \gamma_2|\nu \times \nu),
\tag{15}
$$

s.t. $\pi^\star_{\varepsilon,\tau} \in \arg\min_{\pi \in \mathcal{M}(\mathcal{X} \times \mathcal{Y})} F(\pi, \pi)$. Then, if we define the linearized cost

$$
c^\star_{\varepsilon,\tau} : (\mathbf{x}, \mathbf{y}) \in \mathcal{X} \times \mathcal{Y} \mapsto \int_{\mathcal{X} \times \mathcal{Y}} |c_\mathcal{X}(\mathbf{x}, \mathbf{x}') - c_\mathcal{Y}(\mathbf{y}, \mathbf{y}')|^2 \, \mathrm{d}\pi^\star_{\varepsilon,\tau}(\mathbf{x}', \mathbf{y}'),
\tag{16}
$$

we get from [80, Proposition 9], that $\pi^\star_{\varepsilon,\tau}$ solves:

$$
\pi^\star_{\varepsilon,\tau} \in \arg\min_{\pi \in \mathcal{M}(\times \mathcal{Y})} \int_{\mathcal{X} \times \mathcal{Y}} c^\star_{\varepsilon,\tau}(\mathbf{x}, \mathbf{y}) \, \mathrm{d}\pi(\mathbf{x}, \mathbf{y}) + \varepsilon m(\pi^\star_{\varepsilon,\tau}) \mathrm{KL}(\pi|\mu \otimes \nu)
$$
$$
+ \lambda_1 m(\pi^\star_{\varepsilon,\tau}) \mathrm{KL}(\pi_1|\mu) + \lambda_2 m(\pi^\star_{\varepsilon,\tau}) \mathrm{KL}(\pi_2|\nu).
\tag{17}
$$

Therefore, $\pi^\star_{\varepsilon,\tau}$ solves problem ULEOT between $\mu$ and $\nu$ for a new cost $c^\star_{\varepsilon,\tau}$, and the regularization strength $\varepsilon' = \varepsilon m(\pi^\star_{\varepsilon,\tau})$. We seek to apply point 1, to get that $\pi^\star_{\varepsilon,\tau}$ solves problem LEOT between $\tilde{\mu} = p_1 \sharp \pi^\star_{\varepsilon,\tau}$ and $\tilde{\nu} = p_2 \sharp \pi^\star_{\varepsilon,\tau}$ for the same entropic regularization strength $\varepsilon'$. To that end, we first verify that $c^\star_{\varepsilon,\tau}$ is continuous. Since $c_\mathcal{X}$ and $c_\mathcal{Y}$ are continuous on $\mathcal{X} \times \mathcal{X}$ and $\mathcal{Y} \times \mathcal{Y}$, the function:

$$
(\mathbf{x}, \mathbf{y}, \mathbf{x}', \mathbf{y}') \in (\mathcal{X} \times \mathcal{Y})^2 \mapsto |c_\mathcal{X}(\mathbf{x}, \mathbf{x}') - c_\mathcal{Y}(\mathbf{y}, \mathbf{y}')|^2
$$

is continuous. Therefore, it is bounded since $(\mathcal{X} \times \mathcal{Y})^2$ is compact as a product of compact sets. Then, since $\pi^\star_{\varepsilon,\tau}$ has finite mass, Lebesgue's dominated convergence yields the continuity of $c^\star_{\varepsilon,\tau}$. We then to apply point 1 and get:

$$
\pi^\star_{\varepsilon,\tau} \in \arg\min_{\pi \in \Pi(\tilde{\mu},\tilde{\nu})} \int_{\mathcal{X} \times \mathcal{Y}} c^\star_{\varepsilon,\tau}(\mathbf{x}, \mathbf{y}) \, \mathrm{d}\pi(\mathbf{x}, \mathbf{y}) + \varepsilon' \mathrm{KL}(\pi|\mu \otimes \nu)
\tag{18}
$$

Since the costs are conditionally positive (or conditionally negative) kernels, (18) finally yields the desired result by applying [81, Theorem 3]:

$$\pi_{\varepsilon,\tau}^{\star} \in \arg\min_{\pi\in\Pi(\tilde{\mu},\tilde{\nu})} \int_{\mathcal{X}\times\mathcal{Y}} |c_{\mathcal{X}}(\mathbf{x},\mathbf{x}') - c_{\mathcal{Y}}(\mathbf{y},\mathbf{y}')|\, d\pi(\mathbf{x}',\mathbf{y}')\, d\pi(\mathbf{x},\mathbf{y}) + \varepsilon'\mathrm{KL}(\pi|\mu\otimes\nu) \qquad (19)$$

*Remark* B.4. In various experimental settings, $\mu$ and $\nu$ have mass 1, and we impose one of the two hard marginal constraints, for instance, on $\mu$, by setting $\tau_1 = 1$. Then $\tilde{\nu}$ has also mass 1 and $m(\pi_{\varepsilon,\tau}^{\star}) = 1$, so $\varepsilon' = m(\pi_{\varepsilon,\tau}^{\star}) = \varepsilon$ and we keep the same regularization strength $\varepsilon$ by re-balancing (UQEOT).

$\square$

**Proposition B.5** (Pointwise estimation of re-weighting functions.). *Let* $\pi_{\varepsilon,\tau}^{n} = \sum_{i,j} \mathbf{P}_{\varepsilon,\tau}^{i,j}\delta_{(X_i,Y_j)}$, *solution to* (ULEOT) *between empirical counterparts of* $\mu$ *and* $\nu$. *Let* $\mathbf{a} = \mathbf{P}_{\varepsilon,\tau}\mathbf{1}_n$ *and* $\mathbf{b} = \mathbf{P}_{\varepsilon,\tau}^{\top}\mathbf{1}_n$ *its marginals weights. Then, almost surely,* $n\, a_i \to \eta(X_i)$ *and* $n\, b_i \to \xi(Y_i)$.

*Proof.* As we saw in the proof of Prop.3.1, using Séjourné et al. [80, Prop. 2], one has the existence of $f^\star \in C(\mathcal{X})$ and $g^\star \in C(\mathcal{Y})$ s.t.

$$\frac{d\pi_{\varepsilon,\tau}^\star}{d(\mu\otimes\nu)}(\mathbf{x},\mathbf{y}) = \exp\left(\frac{f^\star(\mathbf{x}) + g^\star(\mathbf{y}) - c(\mathbf{x},\mathbf{y})}{\varepsilon}\right) := h(\mathbf{x},\mathbf{y})$$

and the relative densities $\eta$ and $\xi$ s.t. $\tilde{\mu} = p_1\sharp\pi_{\varepsilon,\tau}^\star = \eta\cdot\mu$ and $\tilde{\nu} = p_2\sharp\pi_{\varepsilon,\tau}^\star = \xi\cdot\nu$ are given by

$$\eta : \mathbf{x} \mapsto \int_{\mathcal{Y}} h(\mathbf{x},\mathbf{y})\, d\nu(\mathbf{y}) \quad \text{and} \quad \xi : \mathbf{y} \mapsto \int_{\mathcal{X}} h(\mathbf{x},\mathbf{y})\, d\mu(\mathbf{x}) \qquad (20)$$

Now, let consider $\hat{\pi}_{\varepsilon,\tau} = \sum_{i,j} \mathbf{P}_{\varepsilon,\tau}^{i,j}\delta_{(X_i,Y_j)}$ the solution of problem LEOT between $\hat{\mu}_n = \frac{1}{n}\sum_{i=1}^{n}\delta_{X_i}$ and $\hat{\nu}_n = \frac{1}{n}\sum_{i=1}^{n}\delta_{Y_i}$. Similarly, using Séjourné et al. [80, Prop. 2], one has the existence of $f_n^\star \in C(\mathcal{X})$ and $g_n^\star \in C(\mathcal{Y})$ s.t.

$$\frac{d\hat{\pi}_{\varepsilon,\tau}}{d(\hat{\mu}_n\otimes\hat{\nu}_n)}(\mathbf{x},\mathbf{y}) = \exp\left(\frac{f_n^\star(\mathbf{x}) + g_n^\star(\mathbf{y}) - c(\mathbf{x},\mathbf{y})}{\varepsilon}\right) := h_n(\mathbf{x},\mathbf{y}) \qquad (21)$$

and the relative densities $\eta_n$ and $\xi_n$ s.t. $\tilde{\mu}_n = p_1\sharp\hat{\pi}_{\varepsilon,\tau}^n = \eta_n\cdot\hat{\mu}_n$ and $\tilde{\nu}_n = p_2\sharp\hat{\pi}_{\varepsilon,\tau}^n = \xi_n\cdot\hat{\nu}_n$ are given by

$$\eta_n : \mathbf{x} \mapsto \frac{1}{n}\sum_{j=1}^{n} h_n(\mathbf{x},\mathbf{y}_j) \quad \text{and} \quad \xi_n : \mathbf{y} \mapsto \frac{1}{n}\sum_{j=1}^{n} h_n(\mathbf{x}_i,\mathbf{y}) \qquad (22)$$

From Eq. (21), we get that $\mathbf{P}_{\varepsilon,\tau}^{ij} = \frac{1}{n^2}h(\mathbf{x}_i,\mathbf{y}_j)$, so reminding that $\mathbf{a} = \mathbf{P}_{\varepsilon,\tau}\mathbf{1}_n$ and $\mathbf{b} = \mathbf{P}_{\varepsilon,\tau}^{\top}\mathbf{1}_n$, one has $n\, a_i = \eta_n(X_i)$ and $n\, b_i = \xi_n(Y_i)$. We now show that almost surely, $\eta_n \to \eta$ and $\xi_n \to \xi$ pointwise, which implies the desired result that, almost surely, $n\, a_i \to \eta(X_i)$ and $n\, b_i \to \xi(Y_i)$.

Almost surely, $\hat{\mu}_n \rightharpoonup \mu$ and $\hat{\nu}_n \rightharpoonup \nu$, so using [74, Proposition 10], $f_n^\star \to f^\star$ and $g_n^\star \to g^\star$ in sup-norm. Since $f_n^\star \to f^\star$ on $\mathcal{X}$ and $g_n^\star \to g^\star$ on $\mathcal{Y}$ in sup-norm, we can show that $h_n \to h$ in sup-norm on $\mathcal{X}\times\mathcal{Y}$. Indeed, for $(\mathbf{x},\mathbf{y}) \in (\mathcal{X}\times\mathcal{Y})$, one has:

$$\begin{aligned}
&|h_n(\mathbf{x},\mathbf{y}) - h(\mathbf{x},\mathbf{y})| \\
&= \left|\exp\left(\frac{f_n^\star(\mathbf{x}) + g_n^\star(\mathbf{y}) - c(\mathbf{x},\mathbf{y})}{\varepsilon}\right) - \exp\left(\frac{f^\star(\mathbf{x}) + g^\star(\mathbf{y}) - c(\mathbf{x},\mathbf{y})}{\varepsilon}\right)\right| \\
&= \exp\left(\frac{c(\mathbf{x},\mathbf{y})}{\varepsilon}\right)\left|\exp\left(\frac{f_n^\star(\mathbf{x}) + g_n^\star(\mathbf{y})}{\varepsilon}\right) - \exp\left(\frac{f^\star(\mathbf{x}) + g^\star(\mathbf{y})}{\varepsilon}\right)\right| \\
&\leq M_{c,\varepsilon}\left|\exp\left(\frac{f_n^\star(\mathbf{x}) + g_n^\star(\mathbf{y})}{\varepsilon}\right) - \exp\left(\frac{f^\star(\mathbf{x}) + g^\star(\mathbf{y})}{\varepsilon}\right)\right|
\end{aligned} \qquad (23)$$

with $M_{c,\varepsilon} = \sup_{(\mathbf{x},\mathbf{y})\in\mathcal{X}\times\mathcal{Y}} \exp\left(\frac{c(\mathbf{x},\mathbf{y})}{\varepsilon}\right) < +\infty$, since $(\mathbf{x},\mathbf{y}) \mapsto \exp\left(\frac{c(\mathbf{x},\mathbf{y})}{\varepsilon}\right)$ is continuous on the compact $\mathcal{X}\times\mathcal{Y}$, as $c$ is continuous. Afterwards,

$$
\begin{aligned}
&\left|\exp\left(\frac{f_n^\star(\mathbf{x}) + g_n^\star(\mathbf{y})}{\varepsilon}\right) - \exp\left(\frac{f^\star(\mathbf{x}) + g^\star(\mathbf{y})}{\varepsilon}\right)\right| \\
&\leq \left|\exp\left(\frac{f_n^\star(\mathbf{x})}{\varepsilon}\right)\exp\left(\frac{g_n^\star(\mathbf{x})}{\varepsilon}\right) - \exp\left(\frac{f_n^\star(\mathbf{x})}{\varepsilon}\right)\exp\left(\frac{g^\star(\mathbf{x})}{\varepsilon}\right)\right| \\
&+ \left|\exp\left(\frac{f_n^\star(\mathbf{x})}{\varepsilon}\right)\exp\left(\frac{g^\star(\mathbf{x})}{\varepsilon}\right) - \exp\left(\frac{f^\star(\mathbf{x})}{\varepsilon}\right)\exp\left(\frac{g^\star(\mathbf{x})}{\varepsilon}\right)\right|
\end{aligned}
\tag{24}
$$

For the first term, one has:

$$
\begin{aligned}
&\left|\exp\left(\frac{f_n^\star(\mathbf{x})}{\varepsilon}\right)\exp\left(\frac{g_n^\star(\mathbf{x})}{\varepsilon}\right) - \exp\left(\frac{f_n^\star(\mathbf{x})}{\varepsilon}\right)\exp\left(\frac{g^\star(\mathbf{x})}{\varepsilon}\right)\right| \\
&= \exp\left(\frac{f_n^\star(\mathbf{x})}{\varepsilon}\right)\left|\exp\left(\frac{g_n^\star(\mathbf{x})}{\varepsilon}\right) - \exp\left(\frac{g^\star(\mathbf{x})}{\varepsilon}\right)\right| \\
&\leq \exp\left(\frac{\|f_n^\star\|_\infty}{\varepsilon}\right)\left|\exp\left(\frac{g_n^\star(\mathbf{x})}{\varepsilon}\right) - \exp\left(\frac{g^\star(\mathbf{x})}{\varepsilon}\right)\right|
\end{aligned}
\tag{25}
$$

First, we can bound uniformly $\exp(\|f_n^\star\|_\infty/\varepsilon)$ since $f_n$ converges in sup-norm, so $(\|f_n^\star\|_\infty)_{n\geq 0}$ is bounded. Then, since $g_n^\star$ convergences in sup-norm, it is uniformly bounded, and since $g^\star$ is continuous on the compact $\mathcal{X}$, it is bounded. Therefore, we can find a compact $K \subset \mathbb{R}$ s.t. $g^\star(\mathcal{X}) \subset K$ and for each $n$, $g_n^\star(\mathcal{X}) \subset K$. Then, applying the mean value theorem to the $C_1$ function $\mathbf{x} \mapsto \exp(\mathbf{x}/\varepsilon)$ on $K$, we can bound:

$$
\left|\exp\left(\frac{g_n^\star(\mathbf{x})}{\varepsilon}\right) - \exp\left(\frac{g^\star(\mathbf{x})}{\varepsilon}\right)\right| \leq \sup_{\mathbf{z}\in K} \tfrac{1}{\varepsilon}\exp(\tfrac{1}{\varepsilon}\mathbf{z})\,|g_n^\star(\mathbf{x}) - g^\star(\mathbf{x})|
\tag{26}
$$

This yields the existence of a constant $M_1 > 0$ s.t.

$$
\left|\exp\left(\frac{f_n^\star(\mathbf{x})}{\varepsilon}\right)\exp\left(\frac{g_n^\star(\mathbf{x})}{\varepsilon}\right) - \exp\left(\frac{f_n^\star(\mathbf{x})}{\varepsilon}\right)\exp\left(\frac{g^\star(\mathbf{x})}{\varepsilon}\right)\right| \leq M_1\|g_n^\star - g^\star\|_\infty
\tag{27}
$$

Using the same strategy, we get the existence of a constant $M_2 > 0$ s.t.

$$
\left|\exp\left(\frac{f_n^\star(\mathbf{x})}{\varepsilon}\right)\exp\left(\frac{g^\star(\mathbf{x})}{\varepsilon}\right) - \exp\left(\frac{f^\star(\mathbf{x})}{\varepsilon}\right)\exp\left(\frac{g^\star(\mathbf{x})}{\varepsilon}\right)\right| \leq M_2\|f_n^\star - f^\star\|_\infty
\tag{28}
$$

Combining (27) and (28) with (23), we get that:

$$
|h_n(\mathbf{x}) - h(\mathbf{x})| \leq M_{c,\varepsilon}\left(M_1\|g_n^\star - g^\star\|_\infty + M_2\|f_n^\star - f^\star\|_\infty\right)
\tag{29}
$$

from which we deduce that $h_n \to h$ in sup-norm, from the convergence of $f_n \to f$ and $g_n \to g$ in sup-norm.

Now, we can show the pointwise convergence of $\hat{\eta}_n$. For any $\mathbf{x} \in \mathcal{X}$, one has:

$$
\begin{aligned}
&|\hat{\eta}_n(\mathbf{x}) - \eta(\mathbf{x})| \\
&= \left|\int h_n(\mathbf{x},\mathbf{y})\,\mathrm{d}\hat{\nu}_n(\mathbf{y}) - \int h(\mathbf{x},\mathbf{y})\,\mathrm{d}\nu(\mathbf{y})\right| \\
&\leq \left|\int h_n(\mathbf{x},\mathbf{y})\,\mathrm{d}\hat{\nu}_n(\mathbf{y}) - \int h(\mathbf{x},\mathbf{y})\,\mathrm{d}\hat{\nu}_n(\mathbf{y})\right| + \left|\int h(\mathbf{x},\mathbf{y})\,\mathrm{d}\hat{\nu}_n(\mathbf{y}) - \int h(\mathbf{x},\mathbf{y})\,\mathrm{d}\nu(\mathbf{y})\right| \\
&\leq \int \|h_n - h\|_\infty\,\mathrm{d}\hat{\nu}_n(\mathbf{y}) + \left|\int h(\mathbf{x},\mathbf{y})\,\mathrm{d}\hat{\nu}_n(\mathbf{y}) - \int h(\mathbf{x},\mathbf{y})\,\mathrm{d}\nu(\mathbf{y})\right| \\
&= \|h_n - h\|_\infty + \left|\int h(\mathbf{x},\mathbf{y})\,\mathrm{d}\hat{\nu}_n(\mathbf{y}) - \int h(\mathbf{x},\mathbf{y})\,\mathrm{d}\nu(\mathbf{y})\right|
\end{aligned}
\tag{30}
$$

Therefore, it almost surely holds that $\hat{\eta}_n(\mathbf{x}) \to \eta(\mathbf{x})$. Indeed, $\|h_n - h\|_\infty \to 0$ since we have shown that $h_n \to h$ in sup-norm. Then, $h$ is continuous on the compact $\mathcal{X}\times\mathcal{Y}$, so it is bounded, so since $\mu_n \rightharpoonup \mu$, we get $\int h\,\mathrm{d}\hat{\nu}_n \to \int h\,\mathrm{d}\nu$. Next, we show similarly that, almost surely, $\xi_n \to \xi$ pointwise.

$\square$

# C Metrics

We start with introducing general metrics in § C.1, some of which will be used in the metrics introduced in the context of experiments on single-cell data in § C.2.

## C.1 General metrics

In the following, we discuss a way how to classify predictions in a generative model. We start with the setting where each mapped sample is to be assigned to a category based on labelled data in the target distribution. We then continue with the case where there are also labels for samples in the source distribution, and this way define a classifier $f_{\text{class}}$ mapping from the source distribution to labels in the target distribution with the possibility to compute accuracy. Building upon this, we assign the classifier $f_{\text{class}}$ an uncertainty score for each prediction. Finally, we define a calibration score assessing the quality of a given uncertainty score.

**Turning a generative model into a classifier** In the following, consider a finite set of samples in the target domain $\mathbf{y}_1, \ldots, \mathbf{y}_M \in \mathcal{Y}$. Assume $\{\mathbf{y}_m\}_{m=1}^M$ allows for a partition $\{\mathbf{y}_m\}_{m=1}^M = \sqcup_{k \in K} \mathcal{T}_k$. Hence, each sample belongs to exactly one class, which we interchangeably refer to as the sample being labelled. Let $T : \mathcal{X} \to \mathcal{Y}$ be a map (deterministic or stochastic), and let $f_{\text{1-NN}} : \mathcal{Y} \to \{\mathcal{T}_k\}_{k=1}^K$ be the 1-nearest neighbor classifier. We obtain a map $g$ from $\mathcal{X}$ to $\{\mathcal{T}_k\}_{k=1}^K$ by the concatenation of $f_{\text{1-NN}}$ and $T$. This map $g$ proves useful in settings when mapped cells are to be categorized, e.g. to assign mapped cells to a cell type.

**A metric to assess the accuracy of a generative model** In the following, assume that the set of samples in the source domain $\mathbf{x}_1, \ldots \mathbf{x}_N$ allows for a partition $\{\mathbf{x}_n\}_{n=1}^N = \sqcup_{k \in K} \mathcal{S}_k$. We want to construct a classifier $f_{\text{class}}$ assigning each category in the source distribution $\{\mathcal{S}_k\}_{k=1}^K$ probabilistically to a category in the target distribution $\{\mathcal{T}_k\}_{k=1}^K$. Define $f_{\text{class}} : \{\mathcal{S}_k\}_{k=1}^K \to \mathbb{N}^K$ via $(f_{\text{class}}(\mathcal{S}_k))_j = \sum_{\mathbf{x}_n \in \mathcal{S}_k} 1_{\{g(\mathbf{x}_n) = \mathcal{T}_j\}}$ where $g : \mathcal{X} \to \{\mathcal{T}_k\}_{k=1}^K$ was defined above.

Assume that there exists a known one-to-one (injective would be sufficient) match between elements in $\{\mathcal{S}_k\}_{k=1}^K$ and elements in $\{\mathcal{T}_k\}_{k=1}^K$. Then we can define a confusion matrix $\mathcal{A}$ with entries $\mathcal{A}_{ij} := \sum_{\mathbf{x}_n \in \mathcal{T}_i} 1_{\{g(\mathbf{x}_n) = \mathcal{S}_j\}}$. In the context of entropic OT the confusion matrix is element-wise defined as

$$\mathcal{A}_{ij} := \sum_{\mathbf{x}_n \in \mathcal{T}_i} 1_{\{f_{\text{1-NN}}(T(\mathbf{x}_n)) = \mathcal{S}_j\}} \tag{31}$$

This way we obtain an accuracy score of the classifier $f_{\text{class}}$ mapping a partition of one set of samples to a partition of another set of samples.

**Calibration score** To assess the meaningfulness of an uncertainty score, we introduce the following calibration score. Assume we have a classifier which yields predictions along with uncertainty estimations. Let $\mathbf{u} \in \mathbb{R}^K$ be a vector containing an uncertainty estimation for each element in $\{\mathcal{S}_k\}_{k=1}^K$. Moreover, let $\mathbf{a} \in \mathbb{R}^K$ be a vector containing the accuracy for each element in $\{\mathcal{S}_k\}_{k=1}^K$. We then define our calibration score to be the Spearman rank correlation coefficient between $u$ and $\mathbf{1}_K - a$, where $\mathbf{1}_K$ denotes the $K-$dimensional vector containing 1 in every entry. In effect, the calibration score is close to 1 if the model assigns high uncertainty to wrong predictions and low uncertainty to true predictions, while the calibration score is close to $-1$ if the model assigns high uncertainty to correct predictions and low uncertainty to wrong predictions.

In the following, we consider a stochastic map $T$. Let $\mathbf{y}_1, ..., \mathbf{y}_L \sim \pi_\theta(\cdot|\mathbf{x})$ obtained from $T$. To obtain a calibration score for $f_{\text{class}}$ we estimate a statistic $V(\pi_\theta(\cdot|\mathbf{x}))$ from the samples $\mathbf{y}_1, ..., \mathbf{y}_L$, reflecting an estimation of uncertainty. Then, we let the uncertainty of the prediction of $f_{\text{class}}$ for category $\mathcal{S}_i$ be the mean uncertainty statistic, i.e. $\sum_{\mathbf{x} \in \mathcal{S}_i} \frac{V(\pi_\theta(\cdot|\mathbf{x}))}{|\mathcal{S}_i|}$. In effect, for each prediction $f_{\text{class}}(\mathcal{S}_i)$ we get the uncertainty score

$$u_i = \sum_{\mathbf{x} \in \mathcal{S}_i} \frac{V(\pi_\theta(\cdot|\mathbf{x}))}{|\mathcal{S}_i|}. \tag{32}$$

**Assessing the uncertainty with the *cos-var* metric**  Gayoso et al. [25] introduce a statistic to assess the uncertainty of deep generative RNA velocity methods from samples of the posterior distribution, which we adapt to the OT paradigm to obtain

$$\text{cos-var}(\pi_\theta(\cdot|\mathbf{x})) = \text{Var}_{Y\sim\pi_\theta(\cdot|\mathbf{x})}[\text{cos-sim}(Y, \mathbb{E}_{Y\sim\pi_\theta(\cdot|\mathbf{x})}[Y])], \tag{33}$$

where cos-sim denotes the cosine similarity. We refer to this metric as cos-var, as it computes the variance of the cosine similarity of samples following the conditional distribution and the conditional mean. We use 30 samples from the conditional distribution to compute this metric.

## C.2  Single-cell specific metrics

**TSI score for developmental systems**  In the following, we describe TSI metric used in Fig. 2. The terminal state identification score (TSI) was introduced in Weiler et al. [92] and assesses to what extent the learnt vector field of a developmental system points towards the correct directions. In effect, we assume that terminal states of the biological developmental process are known. CellRank [42] identifies macrostates using a Schur decomposition based on the transition matrix computed from the learnt vector field (using CellRank's *VelocityKernel*). From these macrostates, CellRank infers a set of putative terminal states, loosely speaking by looking for sinks in the transition matrix. To define the TSI score, we consider a system containing $m$ terminal states and the function $f$ that assigns each number of macrostates $n$ the corresponding number of identified terminal states. In the case of a strategy that identifies terminal states optimally, $f_{\text{opt}}$ describes the step function

$$f_{\text{opt}}(n) = \begin{cases} n, & n < m \\ m, & n \geq m. \end{cases}$$

The TSI score of the learnt vector field $\theta$ is the area under the curve $f_\theta$ relative to the area under the curve $f_{\text{opt}}$, *i.e.*,

$$TSI(\theta) = \frac{\sum_{n=1}^{N_{\max}} f_\theta(n)}{\sum_{n=1}^{N_{\max}} f_{\text{opt}}} = \frac{2}{m(1 + N_{\max} - m)} \sum_{n=1}^{N_{\max}} f_\theta(n).$$

Thus, the TSI score lies between $0$ and $1$, with $1$ being optimal. We compute the LSI score with `https://cellrank.readthedocs.io/en/latest/api/_autosummary/estimators/cellrank.estimators.GPCCA.html#cellrank.estimators.GPCCA.tsi`

In the specific dataset considered here, the terminal cell states are Alpha, Beta, Delta, and Epsilon cells [2]. Hence, the TSI score measures to what extent the underlying vector field captures the dynamics such that cells develop into either of these cell states, while these cell states being a sink. In the CellRank API, we set n_macrostates to 15, i.e., the algorithm looks for 15 macrostates.

**Cell type scores/errors**  As in most single-cell tasks there is no ground truth of matches between cells, we rely on labels of clusters of the data, i.e. on cell types. We then assess the accuracy of a generative model by considering the accuracy of the corresponding classifier $f_{\text{class}}$ as described above. The correct matches between classes have to be considered task-specifically.

In the developmental pancreas example considered in Figures 2 and 9, we have $\{\mathcal{S}_k\}_{k=1}^K = \{\mathcal{T}_k\}_{k=1}^K = \{$Alpha, Beta, Delta, Epsilon, Fev+ Alpha, Fev+ Beta, Fev+ Delta, Fev+ Delta, Fev+ Pyy, Ngn3 High late, Ngn3 High early, Ngn3 low EP$\}$. Here, we do not look for a one-to-one match, as from a developmental biology perspective, it is reasonable (and even expected) that e.g. a Fev+ Alpha cell evolves into an Alpha cell. To measure to what extent the dynamics follow the manifold, we measure how many cells in one of the terminal cell states Alpha, Beta, Delta, Epsilon are derived directly from Ngn3 low EP cells. These transitions are biologically unlikely, as cells are known to evolve via the Ngn3 high cluster and either of Fev+ Alpha, Fev+ Beta, Fev+ Delta or Fev+ Epsilon. Hence, the direct transition from Ngn3 low EP cells to one of the terminal cell states is very unlikely. Thus, we denote this fraction as the cell transition error (*CT error*).

In the following, we discuss the choice of the labels $\{\mathcal{S}_k\}_{k=1}^K$ and $\{\mathcal{T}_k\}_{k=1}^K$ for perturbation prediction in Fig.3, Fig.12. Each drug was applied to cells belonging to three different cell types/cell lines, namely A549, K562, and MCF7. Hence, we can define $\{\mathcal{S}_k\}_{k=1}^K = \{\mathcal{T}_k\}_{k=1}^K =$

$\{A549, K562, MCF7\}$ as for each perturbed cell we know the cell type at the time of injecting the drug.

**FOSCTTM score**   In the following, we consider a setting where the true match between *samples* is known. The FOSCTTM score ("Fraction of Samples Closer than True Match") measures the fraction of cells which are closer to the true match than the predicted cell. Hence, a random match has a FOSCTTM score of $0.5$, while a perfect match has a FOSCTTM score of $0.0$. In the following we only consider discrete distributions. To define the FOSCTTM score for a map $T : \mathcal{X} \to \mathcal{Y}$, let $\mathbf{x}_1, \ldots \mathbf{x}_K \in \mathcal{X}$ be samples from the source distribution and $\mathbf{y}_1, \ldots \mathbf{y}_K \in \mathcal{Y}$ be samples from the target distribution, such that $\mathbf{x}_k$ and $\mathbf{y}_k$ form a true match. Moreover, let $\hat{\mathbf{y}}_k = T(\mathbf{x}_k)$. Let

$$p_j = \frac{\sum_{k \in K} \mathbf{1}_{\|\mathbf{y}_k - \hat{\mathbf{y}}_j\|_2^2 \leq \|\mathbf{y}_j - \hat{\mathbf{y}}_j\|_2^2}}{|K|} \tag{34}$$

and

$$q_j = \frac{\sum_{k \in K} \mathbf{1}_{\|\mathbf{y}_j - \hat{\mathbf{y}}_k\|_2^2 \leq \|\mathbf{y}_j - \hat{\mathbf{y}}_k\|_2^2}}{|K|} \tag{35}$$

Then, the FOSCTTM score between the predicted target $\{\hat{\mathbf{y}}_k\}_{k \in K}$ and the target $\{\mathbf{y}_k\}_{k \in K}$ is obtained as

$$\text{FOSCTMM}(\{\hat{\mathbf{y}}_k\}_{k \in K}, \{\mathbf{y}_k\}_{k \in K}) = \sum_{k \in K} \frac{p_j + q_j}{2}. \tag{36}$$

## D   Datasets

### D.1   Gushchin et al. [30]'s benchmark datasets

We follow the notebook in `https://github.com/ngushchin/EntropicOTBenchmark/blob/main/notebooks/mixtures_benchmark_visualization_eot.ipynb`, which allows to download the data when instantiating the samplers by setting `download=True` in the function `get_guassian_mixture_benchmark_sampler()`.

### D.2   Pancreas single-cell dataset

The dataset of the developing mouse pancreas was published in Bastidas-Ponce et al. [2] and can be downloaded following the guidelines on `https://www.ncbi.nlm.nih.gov/geo/query/acc.cgi?acc=GSE132188`. The full dataset contains measurements of embryonic days 12.5, 13.5, 14.5, and 15.5, while we only consider time points 14.5 and 15.5.

We filter the dataset such that we only keep cells belonging to the cell types of the endocrine branch to ensure that learnt transitions are biologically plausible. Moreover, cells annotated as *Ngn3 high cycling* were removed due to its unknown stage in the developmental process [2]. The removal is justified by the small number of cells belonging to this cell type and its outlying position in gene expression space. Experiments were performed on 30-dimensional PCA space of log-transformed gene expression counts.

### D.3   Drug perturbation single-cell dataset

The dataset was published in [78]. We download the dataset following the instructions detailed on `https://github.com/bunnech/cellot/tree/main`.

For all analyses we computed PCA embeddings (50 dimensions) on the filtered dataset including the control cells and the corresponding drug only. This ensures the capturing of relevant distributional shifts and hence prevents the model from near-constant predictions as the effect of numerous drugs is weak.

The drugs in figure 3 were chosen as in Hetzel et al. [33].

### D.4   Human bone marrow single-cell dataset for modality translation

This dataset contains paired measurements of single-cell RNA-seq readouts and single-nucleus ATAC-seq measurements [51]. This means that we have a ground truth one-to-one matching

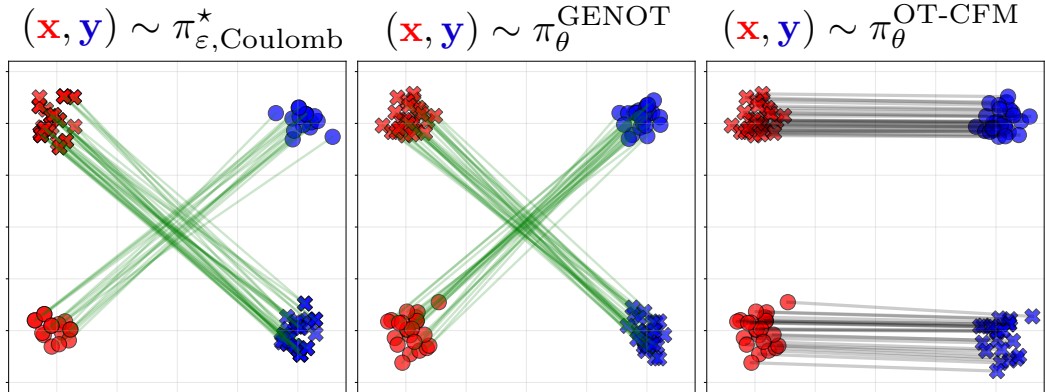

Figure 5: Fitting the EOT coupling between two Gaussian mixtures for the Coulomb cost [4] $c(\mathbf{x}, \mathbf{y}) = 1/\|\mathbf{x} - \mathbf{y}\|_2$, and $\varepsilon = 0.01$, using GENOT and OT-CFM [85]. For both methods, we hence use mini-batch couplings computed with this cost. We connect paired samples with a line. The EOT coupling pairs source samples and target samples diagonally (left). GENOT (middle) generates samples correctly, while OT-CFM (right) fails to preserve the signal from mini-batch couplings. The data-setting is inspired by Bortoli et al. [5, Fig. 1]

for each cell. We use the processed data provided in moscot [39], which can be downloaded following the instructions on `https://moscot.readthedocs.io/en/latest/genapi/moscot.datasets.bone_marrow.html#moscot.datasets.bone_marrow`. This version of the dataset additionally contains a shared embedding for both the RNA and the ATAC data, which we use in the fused term. This embedding was created using a variational autoencoder (scVI [49]) by integrating the RNA counts of the gene expression dataset and gene activity [79] derived from the ATAC data, a commonly used approximation for gene expression estimation from ATAC data [34].

In RNA space we use the PCA embedding (the dimension of which is detailed in the corresponding experiments), while the embedding used in ATAC space is the given LSI (latent semantic indexing) embedding, followed by a feature-wise L2-normalization as proposed in Demetci et al. [17]. The legend for the cell type labels is given in Fig. 25.

# E   Additional information and results for experiments

If not stated otherwise, the GENOT model configuration follows the setup described in appendix G.

## E.1   (U)-GENOT-L

**Entropic OT benchmark by Gushchin et al. [30]**   We assess the performance of GENOT-L following the benchmark in Gushchin et al. [30]. While the results reported in Gushchin et al. [30] are the best ones across different configurations of hyperparameters for certain models, we choose *a single hyperparameter configuration* across all dimensions and entropy regularisation parameters. In particular, GENOT is trained for 100,000 iterations and the learning rate is set to $10^{-5}$. The dimension of the layers in the vector field is set to 1024, and the entropy regularization parameter is set according to the task (i.e. 0.1, 1.0 or 10.0.). The batch size is set to 2048, and its influence is further studied in figure 6. The number of layers per block is set to 3, and the embedding dimension of the time, condition, and noise is set to 1,024. All remaining parameters are chosen as described in appendix G.

Following Gushchin et al. [30], Tab. 2 reports the Bures-Wasserstein Unexplained Variance Percentage ($\mathrm{BW}_2^2$-UVP) between the marginal distributions $p_2 \sharp \pi_\theta$ and $\nu$ defined as

$$\mathrm{BW}_2^2\text{-UVP}\left(p_2 \sharp \pi_\theta, \nu\right) = \frac{100\%}{\frac{1}{2}\,\mathrm{Var}\left(\nu\right)}\,\mathrm{BW}_2^2\left(p_2 \sharp \pi_\theta, \nu\right) \qquad (37)$$

where $\mathrm{BW}_2^2\left(p_2\sharp\pi_\theta, \pi_2^*\right)$ is the Bures-Wasserstein metric between the Gaussian approximations of the distributions $p_2\sharp\pi_\theta$ and $\nu$:

$$\mathrm{BW}_2^2\left(p_2\sharp\pi_\theta, \nu\right) = W_2^2\left(\mathcal{N}\left(\mathrm{m}_\nu, \Sigma_\nu\right), \mathcal{N}\left(\mathrm{m}_{p_2\sharp\pi_\theta}, \Sigma_{p_2\sharp\pi_\theta}\right)\right). \tag{38}$$

Here, $W_2^2$ denotes the Wasserstein-2 distance, and for any measure $\gamma$, $\mathrm{m}_\gamma$ denotes its mean and $\Sigma_\gamma$ its covariance.Tab. 3 displays the conditional Bures-Wasserstein Unexplained Variance Percentage (cBW-UVP), which is the integration of the Bures-Wasserstein distances of the estimated and the true conditional distributions:

$$\mathrm{cBW}_2^2\text{-UVP}\left(\pi_\theta, \pi_\varepsilon^\star\right) = \frac{100\%}{\frac{1}{2}\mathrm{Var}\left(\nu\right)} \int_\mathcal{X} \mathrm{BW}_2^2\left(\pi_\theta(\cdot|\mathbf{x}), \pi_\varepsilon^\star(\cdot|\mathbf{x})\right) \mathrm{d}\mu(\mathbf{x}). \tag{39}$$

---

**Algorithm 2** U-GENOT with stratified sampling of the discrete conditionals. Skip teal steps for GENOT.

1: **Require parameters:** Batch size $n$, number of (per $\mathbf{x}$) conditional sample $k$, entropic regularization $\varepsilon$, unbalancedness parameter $\tau$, discrete solver $\mathrm{Solver}_{\varepsilon,\tau}$.
2: **Require network:** Time-dependent conditional velocity field $v_{t,\theta}(\cdot|\cdot) : \mathbb{R}^q \times \mathbb{R}^p \to \mathbb{R}^q$, re-weighting functions $\eta_\theta : \mathbb{R}^p \to \mathbb{R}$, $\xi_\theta : \mathbb{R}^q \to \mathbb{R}$.
3: **for** $t = 1, \ldots, T_{\mathrm{iter}}$ **do**
4:    $X_1, \ldots, X_n \sim \mu$ and $Y_1, \ldots, Y_n \sim \nu$.
5:    $\mathbf{P}_{\varepsilon,\tau} \leftarrow \mathrm{Solver}_{\varepsilon,\tau}\left((X_i)_{i=1}^n, (Y_i)_{i=1}^n\right) \in \mathbb{R}_+^{n\times n}$.
6:    $\mathbf{a} \leftarrow \mathbf{P}_{\varepsilon,\tau}1_n$ and $\mathbf{b} \leftarrow \mathbf{P}_{\varepsilon,\tau}^\top 1_n$.
7:    Sample $i_1, \ldots, i_n \sim \mathbf{a}$ and set $(X_1, \ldots, X_n) \leftarrow (X_{i_1}, \ldots, X_{i_n})$.
8:    **for** $i = 1, \ldots, n$ **do**
9:        Sample $i_1, ..., i_k \sim \mathbf{P}_{\varepsilon,\tau}[i, :]$.
10:        Sample noise vectors $Z_{i,1}, \ldots, Z_{i,k} \sim \rho$ and time-steps $t_{i,1}, \ldots, t_{i,k} \sim \mathcal{U}([0,1])$.
11:    **end for**
12:    $\mathcal{L}(\theta) \leftarrow \frac{1}{n}\sum_{i=1}^n \frac{1}{k}\sum_{l=1}^k \|v_{t,\theta}([Z_{i,l}, Y_{i_l}]_t|X_i) - (Y_{i_l} - Z_{i,l})\|_2^2 + \frac{1}{n}\sum_{i=1}^n (\eta_\theta(X_i) - n\mathbf{a}_k)^2 + (\xi_\theta(Y_i) - n\mathbf{b}_k)^2$.
13:    $\theta \leftarrow \mathrm{Update}(\theta, \nabla\mathcal{L}(\theta))$.
14: **end for**

---

Hence, $\mathrm{BW}_2^2\text{-UVP}\left(p_2\sharp\pi_\theta, \nu\right)$ measures the fitting property of the push-forward distribution while $\mathrm{cBW}_2^2\text{-UVP}\left(\pi_\theta, \pi_\varepsilon^\star\right)$ measures the fitting property of the joint distribution. Hence, $\mathrm{BW}_2^2\text{-UVP}\left(p_2\sharp\pi_\theta, \nu\right)$ measures the fitting property of the push-forward distribution while $\mathrm{cBW}_2^2\text{-UVP}\left(\pi_\theta, \pi_\varepsilon^\star\right)$ measures the fitting property of the joint distribution.

The values for the competing methods are taken from the original manuscript Gushchin et al. [30]. Competing methods are Seguy et al. [73] (LSOT), Daniels et al. [15] (SCONES), Korotin et al. [40] (NOT), Mokrov et al. [55] (EgNOT), Gushchin et al. [29] (ENOT), Vargas et al. [87] (MLE-SB), De Bortoli et al. [16] (DiffSB), Chen et al. [10] (FB-SDE-A, FB-SDE-J).

To make the results of the benchmark more interpretable, we rank the methods for each experimental setup. If multiple methods yield the same result, we assign the methods the mean of the ranks. Tab. 1 reports the average rank across the twelve different experimental setups.

**Ablation study of GENOT**   A natural way to extend the GENOT algorithm is to sample $k > 1$ instances from the conditional distribution. Alg. 2 describes this procedure in lines 8 to 11, resulting only in an additional sum in the loss in line 12. As for each condition $X_i$ we then have multiple samples $\{Z_{i_l}\}_{l=1}^k$ and $\{Y_{i_l}\}_{l=1}^k$(in the unbalanced case $k$ depends on $i$), we can then couple the noise samples with the samples from the conditional distribution using a discrete Sinkhorn solver (note that the noise and the conditional distribution always live in the same space). This corresponds to using the approach introduced in Pooladian et al. [64], Tong et al. [85] for each single conditional generator $T_\theta(\cdot|\mathbf{x})$. We study this natural extension of the GENOT model in Fig 7. While in a few cases this extensions helps, in the majority of experiments using $k > 1$ decreases the performance.

**GENOT-L for trajectory inference**   Discrete optimal transport is an established method for trajectory inference in single-cell genomics [72]. Due to the ever increasing size of these datasets

| Model | rank $\mathrm{BW}_2^2$-UVP | rank $\mathrm{cBW}_2^2$-UVP $\downarrow$ |
|---|---|---|
| LSOT | 11.33 | 11.33 |
| SCONES | 8.67 | 9.33 |
| DiffSB | 8.92 | 9.50 |
| FB-SDE-J | 6.04 | 9.33 |
| FB-SDE-A | 8.08 | 8.66 |
| EgNOT | 3.04 | 7.08 |
| DSBM | 8.83 | 6.0 |
| NOT | 5.42 | 3.66 |
| ENOT | 6.92 | 3.58 |
| CFM-SB | 5.54 | 3.42 |
| MLE-SB | 3.33 | 3.25 |
| GENOT-L | **1.88** | **2.83** |

Table 1: Average ranks of each method across experiments, on estimating the ground-truth EOT coupling $\pi_\varepsilon^\star$ between each pair of measures $\mu, \nu$, in dimension $d \in \{2, 16, 64, 128\}$ and $\varepsilon \in \{0.1, 1, 10\}$ of Gushchin et al. [30]'s benchmark. As proposed by the authors of the benchmark, we measure performances with both $\mathrm{BW}_2^2$-UVP (37) and $\mathrm{cBW}_2^2$-UVP (39) metrics. See Table 2 and Table 3 for detailed results.

| Model | $\epsilon = 0.1$ | | | | $\epsilon = 1$ | | | | $\epsilon = 10$ | | | |
|---|---|---|---|---|---|---|---|---|---|---|---|---|
| | 2 | 16 | 64 | 128 | 2 | 16 | 64 | 128 | 2 | 16 | 64 | 128 |
| LSOT | - (11.5) | - (11.5) | - (11.5) | - (11.5) | - (12) | - (12) | - (12) | - (12) | - (10.5) | - (10.5) | - (10.5) | - (10.5) |
| SCONES | - (11.5) | - (11.5) | - (11.5) | - (11.5) | 1.06 (10) | 4.24 (10) | 6.67 (9) | 11.54 (9) | 1.11 (7) | 2.98 (6) | 1.33 (3) | 7.89 (4) |
| NOT | 0.016 (3) | 0.63 (6) | 1.53 (6) | 2.62 (6) | 0.08 (3) | 1.13 (9) | 1.62 (8) | 2.62 (7) | 0.225 (5) | 2.603 (5) | 1.872 (4) | 6.12 (3) |
| EgNOT | 0.09 (6) | 0.31 (5) | 0.88 (4) | 0.22 (1) | 0.46 (7) | 0.3 (4) | 0.85 (4) | 0.12 (1) | 0.077 (1) | 0.02 (1.5) | 0.15 (1) | 0.23 (1) |
| ENOT | 0.2 (7) | 2.9 (9) | 1.8 (7) | 1.4 (4) | 0.22 (6) | 0.4 (6) | 7.8 (10) | 29 (10) | 1.2 (8) | 2 (4) | 18.9 (6) | 28 (6) |
| MLE-SB | 0.01 (2) | 0.14 (3) | 0.97 (5) | 2.08 (5) | 0.005 (1) | 0.09 (2) | 0.56 (2) | 1.46 (5) | 0.01 (2) | 1.02 (3) | 6.65 (5) | 23.4 (5) |
| DiffSB | 2.88 (9) | 2.81 (8) | 153.22 (10) | 232.67 (10) | 0.87 (9) | 0.99 (8) | 1.12 (5) | 1.56 (6) | - (10.5) | - (10.5) | - (10.5) | - (10.5) |
| FB-SDE-A | 2.37 (8) | 2.55 (7) | 68.19 (9) | 27.11 (9) | 0.6 (8) | 0.63 (7) | 0.65 (3) | 0.71 (4) | - (10.5) | - (10.5) | - (10.5) | - (10.5) |
| FB-SDE-J | 0.03 (4.5) | 0.05 (2) | 0.25 (2) | 2.96 (7) | 0.07 (2) | 0.13 (3) | 1.52 (7) | 0.48 (3) | - (10.5) | - (10.5) | - (10.5) | - (10.5) |
| CFM-SB | 0.03 (4.5) | 0.30 (4) | 0.55 (3) | 0.81 (3) | 0.09 (5) | 0.36 (5) | 1.24 (6) | 2.76 (8) | 0.31 (6) | 89.84 (8) | 37.50 (7) | 97.44 (7) |
| DSBM | 13.85 (10) | 34.22 (10) | 16.54 (8) | 20.90 (8) | 2.12 (11) | 11.38 (11) | 19.27 (11) | 39.19 (11) | 0.07 (3) | 34.75 (7) | 227.80 (8) | 632.58 (8) |
| GENOT | 0.003 (1) | 0.015 (1) | 0.13 (1) | 0.33 (2) | 0.085 (4) | 0.06 (1) | 0.08 (1) | 0.18 (2) | 0.10 (4) | 0.02 (1.5) | 0.30 (2) | 0.47 (2) |

Table 2: $\mathrm{BW}_2^2$-UVP (equation (37)) between $\pi_2$ and $p_2 \sharp \pi_\theta$ between the distributions defined in Gushchin et al. [30], with numbers in parantheses denoting the rank within the experimental configuration defined by the dimension of the space and the entropy regularization parameter $\varepsilon$.

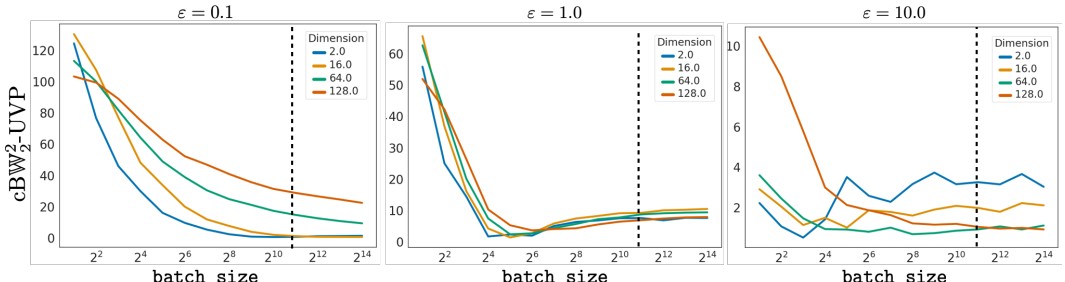

Figure 6: $\mathrm{cBW}_2^2$-UVP (37) between $\pi_\varepsilon^\star$ and $\pi_\theta$ between the distributions defined in Gushchin et al. [30] depending on the batch size. The dotted line corresponds to batch size 2048 and hence displays the values reported in Tab. 3.

[31], neural OT solvers are of particular interest and deterministic Monge map estimators have been successfully applied to millions of cells [32].

Obtaining samples from the conditional distribution allows for an assessment of the uncertainty of the trajectory of a cell. We use the metric $\mathrm{Var}_{Y \sim \pi_\theta(\cdot | \mathbf{x})}[\cos\text{-sim}(Y, \mathbb{E}_{Y \sim \pi_\theta(\cdot | \mathbf{x})}[Y])]$ suggested in Gayoso et al. [25] for generative RNA velocity models (appendix C.1). Therefore, we use 30 samples from the conditional distribution.

| Model | $\epsilon = 0.1$ | | | | $\epsilon = 1$ | | | | $\epsilon = 10$ | | | |
|---|---|---|---|---|---|---|---|---|---|---|---|---|
| | 2 | 16 | 64 | 128 | 2 | 16 | 64 | 128 | 2 | 16 | 64 | 128 |
| LSOT | - (11.5) | - (11.5) | (11.5) | - (11.5) | - (12) | - (12) | - (12) | - (12) | - (10.5) | - (10.5) | - (10.5) | - (10.5) |
| SCONES | - (11.5) | - (11.5) | (11.5) | - (11.5) | 34.88 (10) | 71.34 (10) | 59.12 (9) | 136.44 (9) | 32.9 (8) | 50.84 (7) | 60.44 (7) | 52.11 (6) |
| NOT | 1.94 (3) | 13.67 (3) | 11.74 (2) | 11.4 (2) | 4.77 (3) | 23.27 (6) | 41.75 (7) | 26.56 (4) | 2.86 (3) | 4.57 (5) | 3.41 (3) | 6.56 (3) |
| EgNOT | 129.8 (10) | 75.2 (9) | 60.4 (7) | 43.2 (7) | 80.4 (11) | 74.4 (11) | 63.8 (10) | 53.2 (6) | 4.14 (6) | 2.64 (4) | 2.36 (2) | 1.31 (2) |
| ENOT | 3.64 (4) | 22 (5) | 13.6 (3) | 12.6 (3) | 1.04 (1) | 9.4 (4) | 21.6 (4) | 48 (5) | 1.4 (1) | 2.4 (3) | 19.6 (5) | 30 (5) |
| MLE-SB | 4.57 (5) | 16.12 (4) | 16.1 (5) | 17.81 (4) | 4.13 (2) | 9.08 (2) | 18.05 (3) | 15.226 (3) | 1.61 (2) | 1.27 (1) | 3.9 (4) | 12.9 (4) |
| DiffSB | 73.54 (8) | 59.7 (8) | 1386.4 (10) | 1683.6 (10) | 33.76 (9) | 70.86 (9) | 53.42 (8) | 156.46 (10) | - (10.5) | - (10.5) | - (10.5) | - (10.5) |
| FB-SDE-A | 86.4 (9) | 53.2 (7) | 1156.82 (9) | 1566.44 (9) | 30.62 (8) | 63.48 (7) | 34.84 (5) | 131.72 (8) | - (10.5) | - (10.5) | - (10.5) | - (10.5) |
| FB-SDE-J | 51.34 (7) | 89.16 (10) | 119.32 (8) | 173.96 (8) | 29.34 (7) | 69.2 (8) | 155.14 (11) | 177.52 (11) | - (10.5) | - (10.5) | - (10.5) | - (10.5) |
| CFM-SB | 0.45 (1) | 4.09 (2) | 4.94 (1) | 7.17 (1) | 6.31 (5) | 5.99 (1) | 4.76 (1) | 5.26 (1) | 10.18 (7) | 90.85 (8) | 43.45 (6) | 103.59 (7) |
| DSBM | 28.84 (6) | 49.33 (6) | 37.39 (6) | 41.52 (6) | 5.78 (4) | 17.12 (5) | 36.81 (6) | 60.51 (7) | 2.95 (4) | 35.89 (6) | 229.96 (8) | 642.51 (8) |
| Independent | 166.0 | 152.0 | 126.0 | 110.0 | 86.0 | 80.0 | 72.0 | 60.0 | 4.2 | 2.52 | 2.26 | 2.4 |
| GENOT | 0.46 (2) | 0.91 (1) | 14.48 (4) | 28.52 (5) | 7.43 (6) | 9.10 (3) | 8.61 (2) | 6.76 (2) | 3.24 (5) | 1.97 (2) | 0.91 (1) | 1.03 (1) |

Table 3: $\text{cBW}_2^2$-UVP (37) between $\pi_\varepsilon^\star$ and $\pi_\theta$ between the distributions defined in Gushchin et al. [30], with numbers in parantheses denoting the rank within the experimental configuration defined by the dimension of the space and the entropy regularization parameter $\varepsilon$.

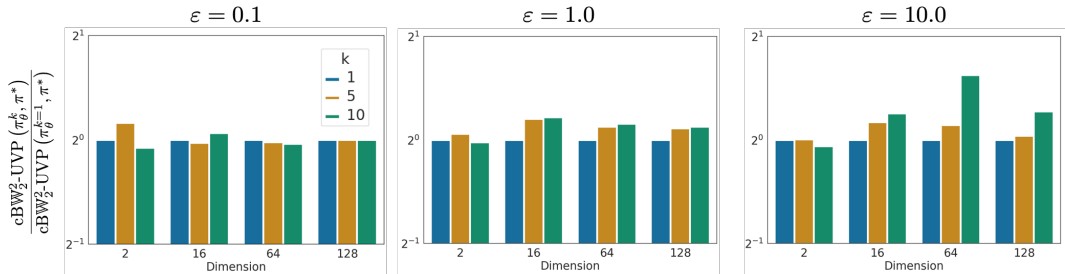

Figure 7: Evaluation of the influence of the number of data points sampled from the conditional distribution in each step, and optimally coupling noise and samples from the target distribution, see Alg. 2. $k = 1$ corresponds to the model performances reported in table 3.

Fig. 8 visualizes the pancreatic endocrinogenesis dataset (D.2) according to source (embryonic day 14.5) and target distribution (15.5) as well as according to cell type. To quantitatively confirm the visual results from Fig. 2, we aggregate the uncertainty to cell type level. Indeed, we observe a higher uncertainty in cells which have not committed to a certain lineage (Ngn3 low, Ngn3 High early, Ngn3 High late), while more mature cell types indicate lower variances.

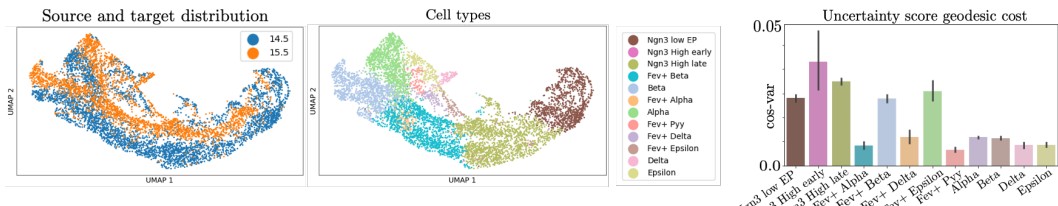

Figure 8: UMAP of the mouse pancreas development dataset colored by sample (left) and cell type (middle). We transport samples from embryonic day 14.5 to embryonic day 15.5. Uncertainty score as displayed in Fig 2 aggregated to cell type level.

A cell is expected to have an uncertain trajectory when it can still develop into different cell lineages. In contrast, cells are expected to have a less uncertain trajectory when their descending population is homogeneous or they belong to a terminal cell state, and hence have committed to a certain lineage.

GENOT-L produces meaningful uncertainty assessments as can be seen from Fig. 2 and Fig. 8. Indeed, the Ngn3 low EP population and the Ngn3 high EP populations have a higher uncertainty than later cell types. The uncertainty is particularly high for Ngn3 high EP late cells, a state where cells commit to a fate towards alpha, beta, delta, or epsilon cells [2].

We now focus on the conditional distributions of cells from the Ngn3 low population, i.e. the least mature cell population in the dataset. Fig. 9 shows more examples of selecting one Ngn3 low cell and generating 30 samples from its conditional distribution as displayed in Fig 2. It is important to

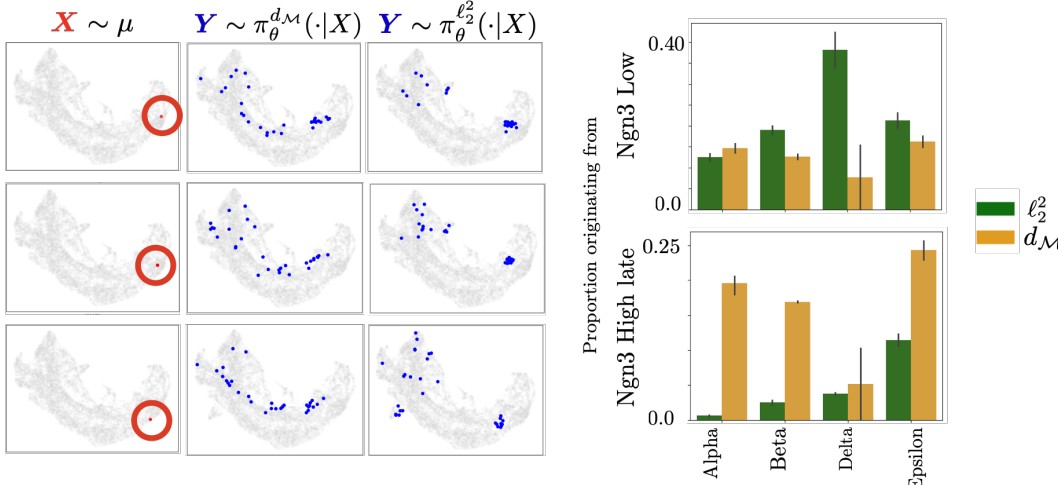

Figure 9: **Left**: Samples from the Ngn3 Low EP population in the leftmost column, and next to it 30 samples from their conditional distribution samples from GENOT with the geodesic cost (middle) and the squared Euclidean cost (right). **Right**: Proportions of alpha, beta, delta, and epsilon cells derived from the Ngn3 Low population (top) and the Ngn3 High population (bottom). Mean and standard deviation across three different runs are reported.

note that the UMAP has to be recomputed to include the generated samples and hence the UMAPs change slightly. The three instances in Fig.9 support the hypothesis that the conditional distribution follows the data manifold when using the geodesic cost (here we assume that the UMAP represents the manifold well). In contrast, choosing the squared Euclidean distance results in samples of the conditional distribution being either very close on the UMAP or very far away, while the intermediate stage (i.e. Ngn3 High late) is skipped. Biologically, this result is not plausible as all cells are known to evolve from Ngn3 Low EP cells to Ngn3 High EP cells, before they finally commit to a lineage (towards alpha, beta, delta, or epsilon cells). It is important to note that both GENOT-L models are trained with the same entropy regularization parameter $\varepsilon = 0.01$.

Based on the examples, we hence hypothesize that when using the squared Euclidean cost, generated cells in the terminal cell states (alpha, beta, delta, epsilon) are much more likely to be derived from Ngn3 low EP cells than Ngn3 High EP cells (in the following we focus on Ngn3 High Late cells and neglect Ngn3 High EP early cells as their number is very low, see Fig.8). As outlined above, this is biologically not plausible. The impression conveyed by the UMAP plots are confirmed numerically on the right hand side of Fig. 9. For each generated cell, we compute its nearest neighbor in the target distribution, and this way assign it a label (cell type), see App. C.2. We then obtain a cell type to cell type transition matrix, which we column-normalise (with columns containing the cell types of the generated cells), such that for each cell type, we obtain the distribution of ancestor cell types. Figure 9 shows that for the Euclidean cost, around 30% of all delta cells are derived from Ngn3 Low EP cells, while only 5% are derived from Ngn3 High EP cells. In contrast, the proportion of progenitors in Ngn3 low EP cells and Ngn3 high EP cells when using the geodesic cost is much more comparable. Similar results can be observed for alpha, beta, and epsilon cells.

**1d simulated data** Figure 10 visualizes the influence of $\tau$ on a simple setting between mixtures of Gaussians. While $\tau = 1.0$ corresponds to the fully balanced case, setting $\tau = 0.97$ results in a complete discardment of one mode in the target distribution. The ground truth is computed with a discrete entropy-regularized OT solver [14], with kernel density estimation to create the plot. This figure is meant to visualize the task, while in the following, we assess the results quantitatively.

**Benchmark of U-GENOT-L between Gaussian distributions** Therefore, we leverage the closed-form solutions of the unbalanced EOT coupling between Gaussian distributions for the squared Euclidean cost [36]. We follow the data generation process described by Janati et al. [36], and evaluate the results for different entropy-regularisation parameters $\varepsilon$ and different unbalancedness parameters $\lambda = \frac{\varepsilon\tau}{1-\tau}$, and across different dimensions.

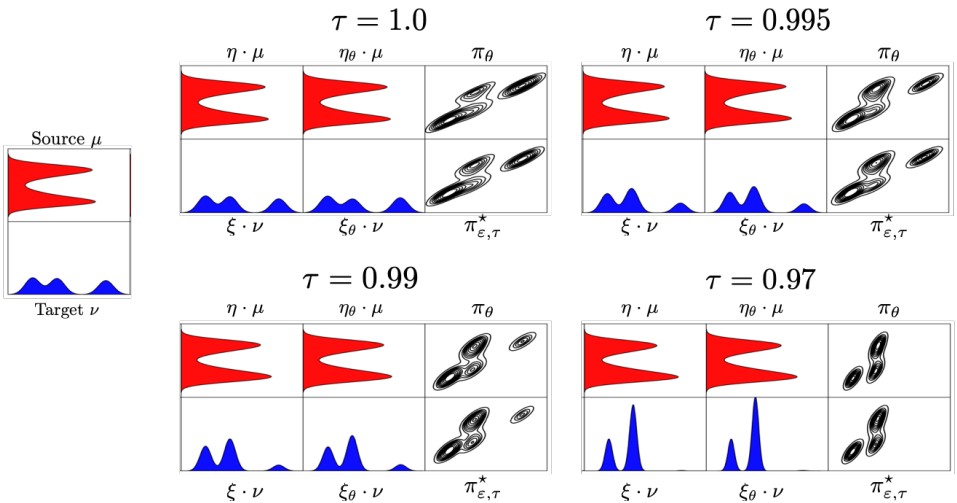

Figure 10: Unbalanced entropic neural optimal transport plan with $\varepsilon = 0.05$ and varying unbalancedness parameter $\tau = \tau_1 = \tau_2$.

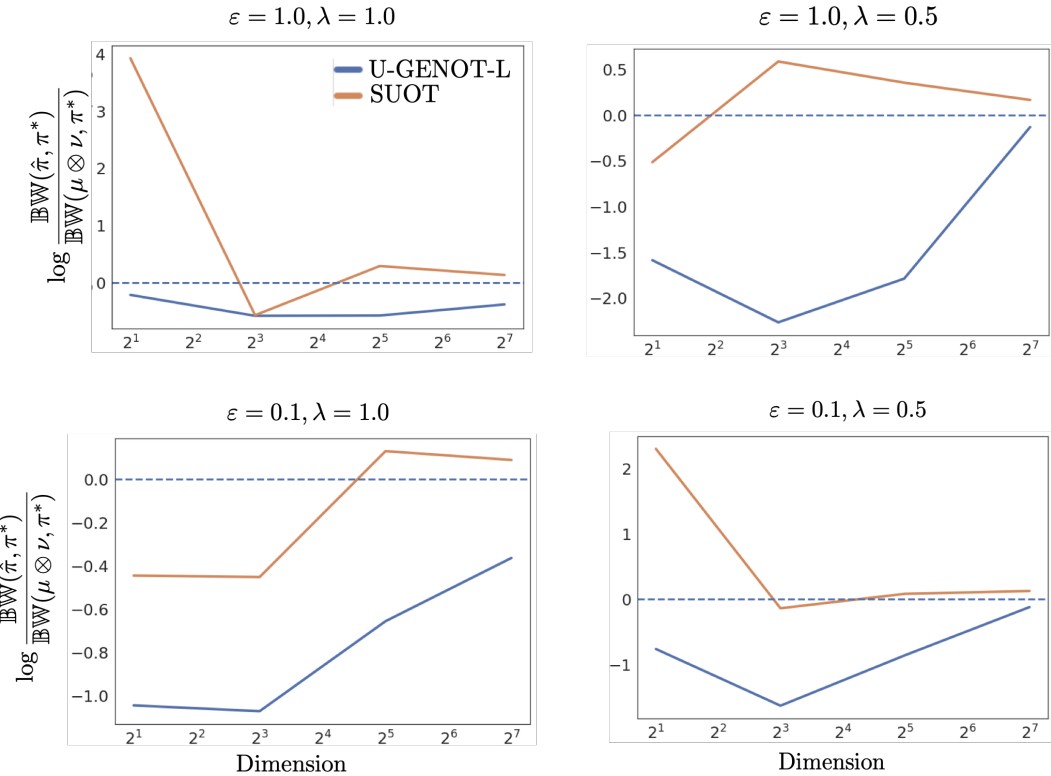

Figure 11: Benchmark of U-GENOT-L on the task of learning an unbalanced entropic OT plan between Gaussian distributions across different dimensions, entropy regularisation parameters $\varepsilon$, and unbalancedness parameters $\lambda$, following Janati et al. [36]. The dotted line represents the outer coupling, while SUOT denotes the method proposed in Yang and Uhler [93].

Yang and Uhler [93] propose a method (scalable unbalanced optimal transport, SUOT) to learn unbalanced EOT plans, but their implementation only supports unbalanced deterministic Monge maps. We visualize the results in Fig.11. Besides the simple switch to learning unbalanced EOT plans, U-GENOT-L was run with the same setup as in the balanced linear EOT benchmark.

**Perturbation modeling with GENOT-L and U-GENOT-L** For each drug, we project the single-cell RNA-seq readout of the unperturbed and perturbed cells to a 50-dimensional PCA embedding. Subsequently, we split the data randomly to obtain a train and test set with a ratio of 60%/40%. This preprocessing step holds for both the calibration score experiments and the experiments conducted with U-GENOT-L to assess the influence of unbalancedness to the accuracy score.

The uncertainty score for the calibration study is computed based on 30 samples from the conditional distribution, see appendix C.2.

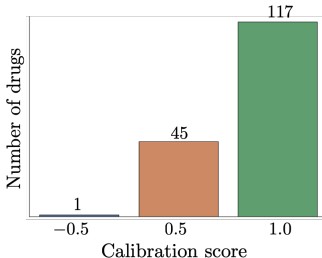

Figure 12: Calibration score for the predictions of GENOT-L for modeling cellular responses to 163 cancer drugs.

In Figure 13 and Figure 14, we visualize the influence of unbalancedness for perturbation modeling with Dacinostat and Tanespimycin, respectively. These experiments were conducted on the full dataset for visualization reasons. While the fitting property seems to be little affected by incorporating unbalancedness (top rows), the cell type clusters are better separated for U-GENOT-L transport plans than for GENOT-L transport plans.

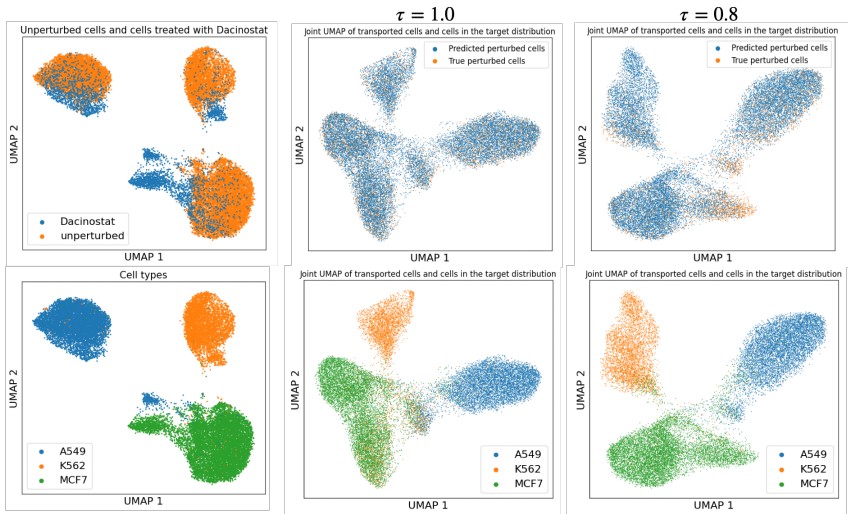

Figure 13: Visual assessment of the influence of unbalancedness in modeling cellular predictions to the cancer drug Dacinostat. In the left column, the source and target distribution are jointly plotted with cells colored by whether they belong to the source (unperturbed) or the target (perturbed) distribution (top), and which cell type they belong to (bottom). In the center column, we plot a UMAP embedding of target and predicted target distribution. The top plot colors cells according to whether a cell belongs to the target distribution or the predicted target distribution. The bottom plot is colored by cell type. The cell type of the predicted target distribution is the cell type of the pre-image of the predicted cell. The right column visualizes the same results, but this time obtained from U-GENOT-L with unbalancedness parameters $\tau = \tau_1 = \tau_2 = 0.8$.

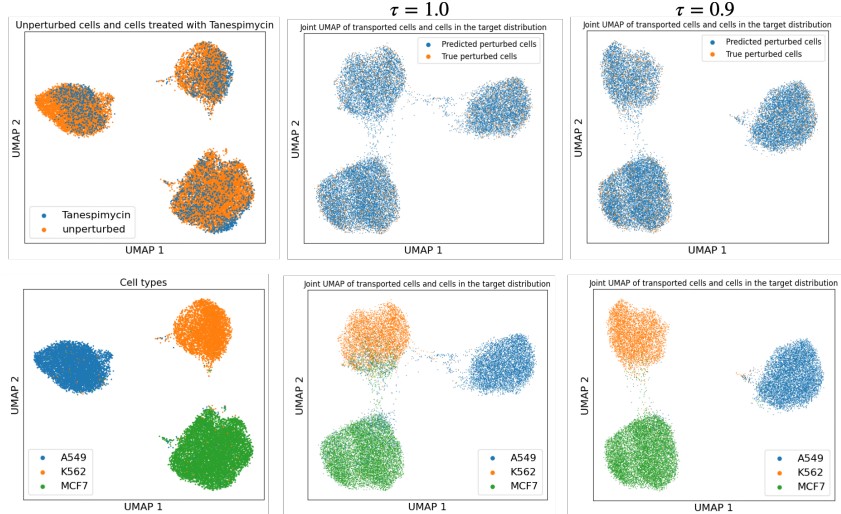

Figure 14: Visual assessment of the influence of unbalancedness in modeling cellular predictions to the cancer drug Tanespimycin. In the left column, the source and target distribution are jointly plotted with cells colored by whether they belong to the source (unperturbed) or the target (perturbed) distribution (top), and which cell type they belong to (bottom). In the center column, we plot a UMAP embedding of target and predicted target distribution. The top plot colors cells according to whether a cell belongs to the target distribution or the predicted target distribution. The bottom plot is colored by cell type. The cell type of the predicted target distribution is the cell type of the pre-image of the predicted cell. The right column visualizes the same results, but this time obtained from U-GENOT-L with unbalancedness parameters $\tau = \tau_1 = \tau_2 = 0.9$.

## E.2 (U)-GENOT-Q & (U)-GENOT-F

**GENOT-Q for mapping a Swiss role to a spiral** Fig. 15 visualizes the dependence of the conditional distribution on the entropy regularization parameter $\varepsilon$ with the same setup as in Figure 3. In particular, we display what happens when we consider the outer coupling. Fig. 16 visualizes the density in the target space, evolving from a 2-dimensional Gaussian distribution at $t = 0$ to conditional distributions at $t = 1$. Analogously to Fig.15, we visualize the influence of the entropy regularization parameter $\varepsilon$.

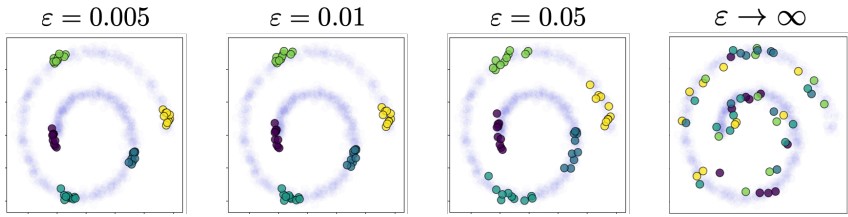

Figure 15: Samples from conditional distributions $\pi_\theta(\cdot|\mathbf{x})$ for GENOT-Q models trained with different entropy regularization parameters $\varepsilon$. The setup is the same as in Figure 3, in effect we transport a three-dimensional Swiss roll to a two-dimensional spiral, which is colored in blue (with high transparency). The right plot shows the result when choosing the outer coupling as opposed to an EOT coupling. The source distribution as well as the data points which are conditioned on are visualized in Figure 3.

**How to handle non-uniqueness of quadratic EOT couplings** In the following, we show how to empirically overcome the non-uniqueness of solutions to (QEOT). The left hand side of Fig. 17 visualizes the solutions to an entropic GW coupling between two Gaussian distributions learnt by GENOT-Q without any modification. It becomes evident that the learned solution is not close to a valid solution, but rather resembles a mixture of solutions.

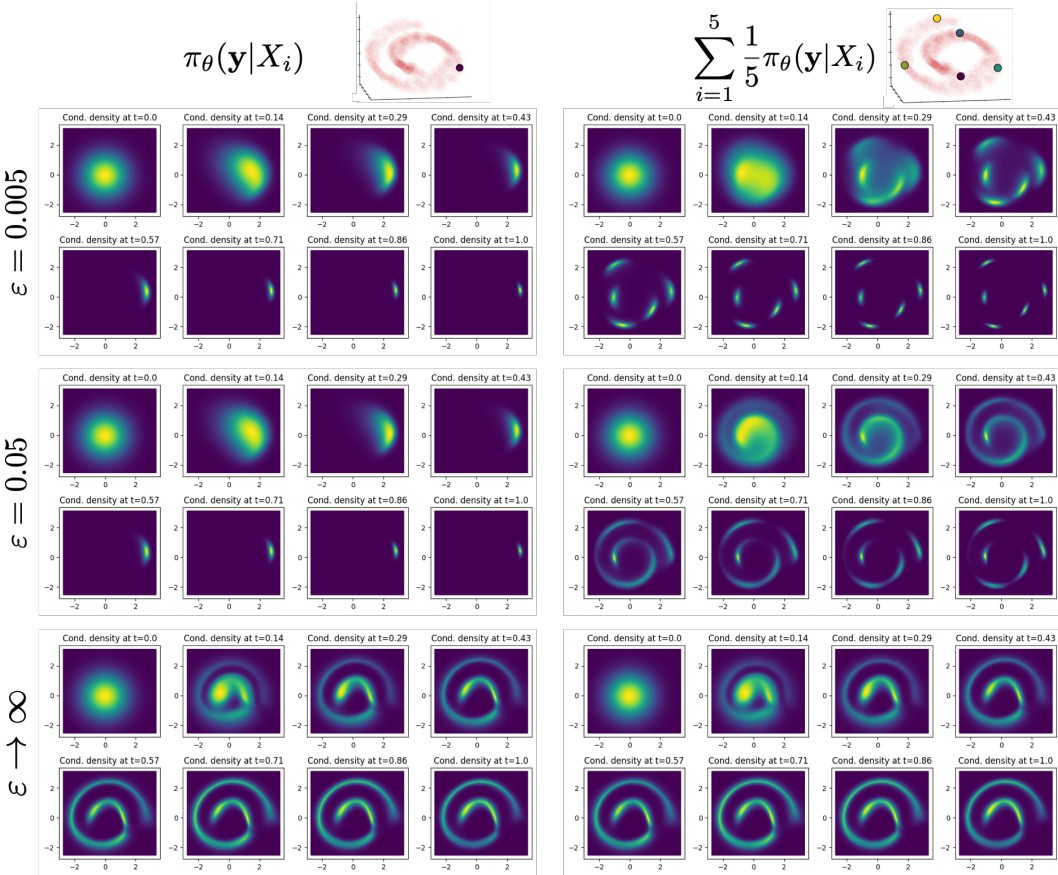

Figure 16: Evolution of the density from $t = 0$ corresponding to the latent distribution (i.e. multivariate standard Gaussian distribution) to $t = 1$ corresponding to the density of the conditional distribution $\pi_\theta(\mathbf{y}|X_i)$. The setup is the same as in Figure 3, in effect we transport a three-dimensional Swiss roll to a two- dimensional spiral The left panel displays the densities of *one* conditional distribution, while the right panel visualizes a mixture of densities over time. The top part of the figure corresponds to GENOT-Q models with entropy regularization parameter $\varepsilon = 0.005$, while the middle ones corresponds to $\varepsilon = 0.005$. The bottom ones correspond to the outer coupling. The markers on the 3-dimensional plot on the very top correspond to data points in the source distribution.

We discuss three ways to overcome the limitations of non-uniqueness of quadratic OT couplings:

- Incorporation of a fused term: Whenever the data allows, using a fused term empirically helps to stabilize the solution.

- Changing the cost function: The set of solutions to (QEOT) depends on the cost function. Thus, finding a cost with a small set of solutions overcomes the limitation to a certain extent. Empirically, the geodesic cost helps to do so for single-cell data.

- Initialisation of the discrete solver: We initialize the discrete GW solver at iteration $i + 1$ using the solution to the linear EOT problem between $p_2\hat{\pi}_\theta^i$ and $\nu$, with $\hat{\pi}_\theta^i$ denoting the learnt coupling at step $i$.

Visually, the initialisation scheme helps for learning the GW-EOT coupling between Gaussian distributions (17).

We now quantify to what extent we can reduce the diversity of the solutions by considering the variance of the conditional distribution of a single data point. Therefore, we consider two datasets:

1. 30 dimensional Gaussian distributions (20 dimensions in the quadratic term and 10 dimensions in the additional fused term): For both the source ($\mu$) and the target ($\nu$) distribution we

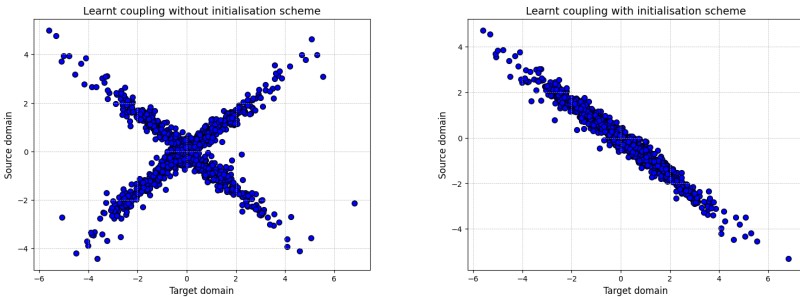

Figure 17: **Left:** Learnt transport plan of GENOT-Q *without initialization* between two one-dimensional Gaussian distributions. **Right:** Learnt transport plan of GENOT-Q *with* initialization. The initialization scheme prevents GENOT from learning mixtures of valid transport plans.

> sample mean vectors from $\mathcal{U}([-1, 1])$ and construct the covariance matrices by sampling each entry of their square roots from $\mathcal{U}([0, 3])$

2. 30 dimensional single-cell data (as in 4): 20-dimensional PCA of the normalized ATAC data, 20-dimensional PCA projection of the gene expression data, and 10-dimensional PCA of the VAE embedding (shared space).

To assess the stability of the solution of discrete EOT, we fix one sample from the source distribution, and compute the variance of $\hat{\pi}(\cdot|X_0[10:])$, i.e. of the incomparable space only, which is obtained from a discrete EOT solver with different samples in source and target. We do this for $\varepsilon = 0.01$ and $\varepsilon = 0.001$ with the squared Euclidean cost.

We consider the following couplings:

1. outer coupling, supposed to serve as a baseline.

2. Gromov-Wasserstein (GW): We use $\tilde{\mathcal{X}}$ and $\tilde{\mathcal{Y}}$, i.e., we use the last 20 dimensions of both the source and the target distribution.

3. Fused Gromov-Wasserstein (FGW): We use $\Omega \times \tilde{X}$ and $\Omega \times \tilde{Y}$ with $\alpha = \frac{1}{11}$.

4. Gromov-Wasserstein with geodesic cost (Geodesic): We use $\tilde{\mathcal{X}}$ and $\tilde{\mathcal{Y}}$ with the geodesic cost.

5. Gromov-Wasserstein with initialization as described above.

Fig.18 shows that both the fused term and the initialization scheme empirically help to preserve the variance of the conditional distributions. Moreover, the geodesic cost helps for small $\varepsilon$ to reduce the conditional variance, suggesting a less diverse set of solutions to (QEOT).

**GENOT-Q und U-GENOT-Q for learning EOT plans between Gaussians** To quantitatively assess the performance of GENOT-Q, we evaluate its performance by learning quadratic EOT couplings between Gaussian distributions [43]. We follow the authors' data generation process and computations for the closed-form solution and the computation of the discrete GW solver. Note that we can only evaluate the cost of the learnt transport plan as the solution of the EOT plan is not unique. Thus, we use the initialisation scheme suggested above. The discrete solution calculated based on the discrete coupling (1000 samples), while we report results from GENOT-GW on both training data (30k samples) and test data (5k samples). GENOT-GW is trained with batch size 1024 and for 5k iterations, with the same network architecture as used for the benchmark for balanced, linear EOT for the squared Euclidean cost (§ F.1). As a baseline, we report the cost induced by the outer coupling. To keep the results on a similar scale across different dimensions, we report the L1-distance between the predicted cost and the true cost, divided by the true cost.

We continue with the evaluation in the unbalanced setting in Fig. 20. Here, we compare only the costs induced by the distortion, i.e. not the one induced by the KL penalisation, as for the latter one requires access to the densities. As demonstrated, GENOT can estimate the densities, but it comes at the cost of an estimation error. As the purpose of this benchmark is to show the accuracy of the learnt

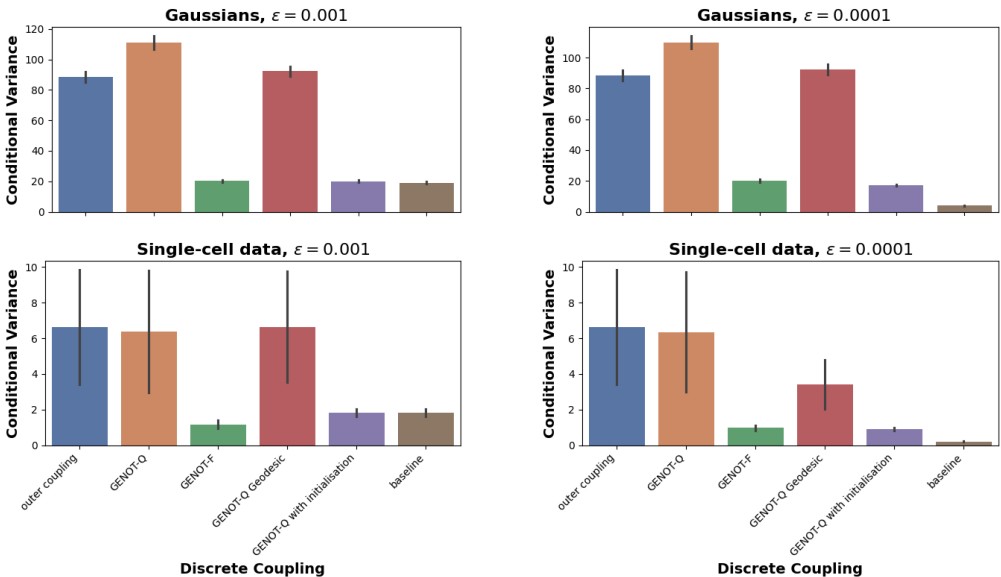

Figure 18: Variance of conditional distributions of different couplings for $\varepsilon = 0.001$ (left) and $\varepsilon = 0.0001$ (right) for both Gaussian data (top) and single-cell data (HSCP also considered in e.g. Fig. 4. Error bars denote standard error across dimensions of the 20-dimensional space.) (bottom)

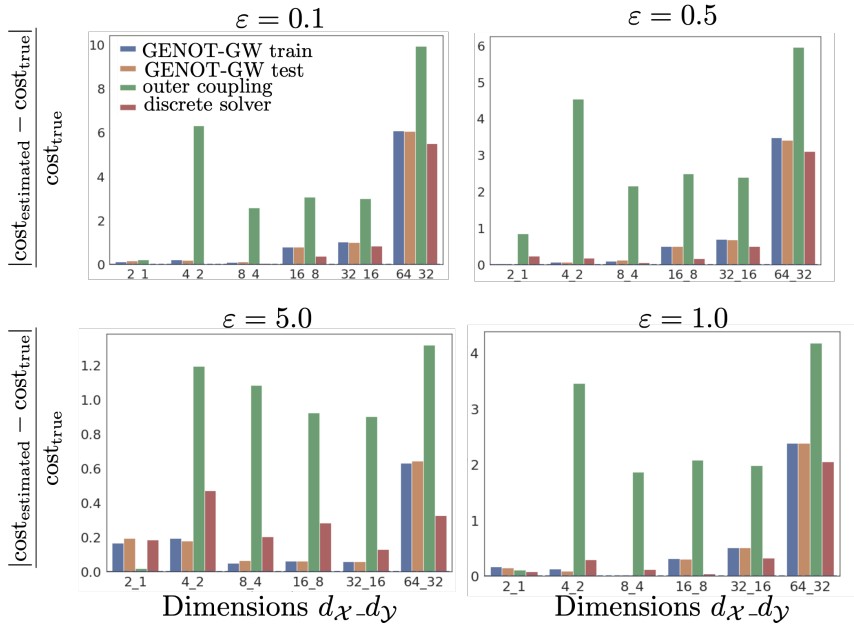

Figure 19: Learning Gromov-Wasserstein entropic optimal transport couplings between Gaussian distributions where the ground truth plan is known for [43]. The quality of the learnt coupling is assessed with the induced cost.

couplings (and not the estimation of the density), we compare based on the distortion cost. As done by the authors in Le et al. [43], we set the total mass of both distributions to 1.0.

**Modality translation with GENOT-Q**   For all experiments, we perform a random 60/40 split for training and test data. All results are reported on the test dataset. The cost matrices of all models

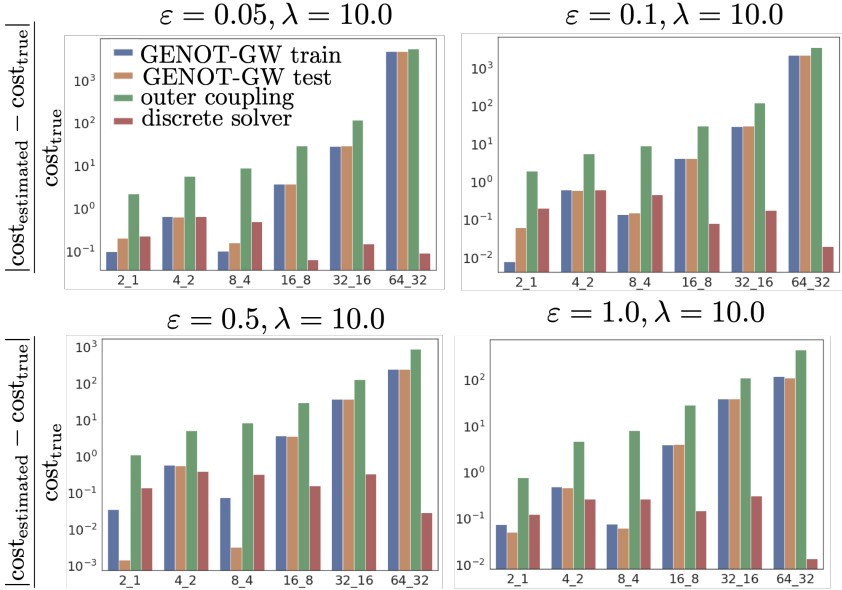

Figure 20: Learning unbalanced Gromov-Wasserstein entropic optimal transport couplings between Gaussian distributions where the ground truth plan is known for [43]. The quality of the learnt coupling is assessed with the induced cost.

were scaled by its mean and the entropy regularization parameter $\varepsilon$ was set to $0.0001$. Moreover, the models were trained for 5,000 iterations.

**Modality translation with GENOT-F** For all experiments, we perform a random 60-40 split for training and test data. All results are reported on the test dataset. The cost matrices of all models were scaled by its mean and the entropy regularization parameter $\varepsilon$ was set to $0.001$. Moreover, the models were trained for 20,000 iterations.

Figure 21 reports results of the GENOT-F model with interpolation parameters $\alpha = 0.3$ (left) and $\alpha = 0.7$ (right). While the Sinkhorn divergences are not comparable with results of the GENOT-Q model due to the respective target distributions living in different spaces, we can compare GENOT-Q with GENOT-F with respect to the FOSCTTM score. Figure 21 shows that GENOT-F strikingly outperforms GENOT-Q, hence the incorporation of the fused term is crucial for a good performance. At the same time, it is important to mention that the GW terms add valuable information to the problem setting, which can be derived from the results for GENOT-F with $\alpha = 0.3$. Here, the higher influence of the fused term causes the model to perform overall worse.

Moreover, we can visualize the optimality and fitting term in a UMAP embedding [53]. To demonstrate the robustness of our model, we train a GENOT-F model with $\varepsilon = 0.01$, $\alpha = 0.5$ and the Euclidean distance on 60% of the dataset (38 dimensions for the ATAC LSI embedding, 50 dimensions for the RNA PCA embedding, and 28 dimensions for the VAE embedding in the fused term) and evaluate the learnt transport plan visually. Figure 22 shows joint UMAP embeddings of predicted target and target distributions, the full legend of cell types can be found in Figure 25. Qualitatively, a good mix between data points of the predicted target and the target distribution suggests a good fitting term. Optimality of the mapping can be visually assessed by considering to what extent cell types are mixed (low optimality) or separated from other cluster (high optimality).

We observed that taking the conditional mean improves results on the FOSCTTM score, but can impair the fitting property. Indeed, the mixing rate between data points belonging to the target and data points belonging to the predicted target seems to be slightly worse when considered in the joint embedding as well as when considering only the fused space and only considering the quadratic space (22) .

**Modality translation with U-GENOT-F** To simulate a setting where there is not a match for certain cells in the target space (RNA), we remove the cells labelled as Proerythroblasts, Erythroblasts,

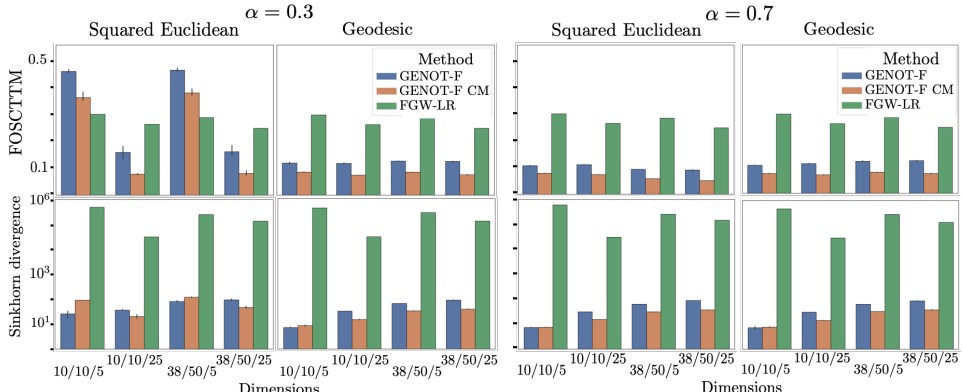

Figure 21: Mean and 95% confidence interval (across three runs) of the FOSCTTM score (top) and the Sinkhorn divergence (bottom) of GENOT-F and discrete FGW with linear regression for out-of-sample estimation. Experiments are categorized by the numbers $d_1/d_2/d_3$, where $d_1$ is the dimension of the space corresponding to the GW of the source distribution, $d_2$ is the dimension of the space corresponding to the GW of the target distribution, and $d_3$ is the dimension of the shared space. Results are reported for the interpolation parameter $\alpha = 0.3$ (left half) and $\alpha = 0.7$ (right half).

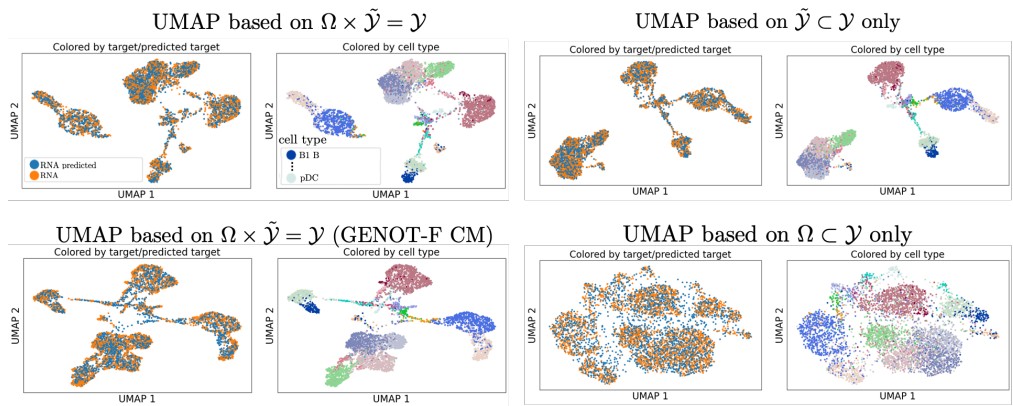

Figure 22: UMAP embedding of target and predicted target based on subspaces of the target domain. Top left: full target space. Top right: incomparable space only. Bottom left: full target space (with predicted target computed by the mean of the conditional distributions). Bottom right: share space only. Cells are colored based on whether they belong to the target distribution or the predicted target distribution in the left plot of each subplot. Cells are colored based on their cell type in the right plot of each subplot. For cells which belong to the predicted target distribution, the cell type is defined as the cell type of the preimage. Shown are results on the test data set, corresponding to 40% of the full dataset.

and Normoblasts as these cells form a lineage, developing into mature Reticulocytes (not present in the dataset). Thus, they are similar in their cellular profile while being clearly distinguishable from the remaining cells. Note that this simulates an extreme setting of class imbalances, as these cells make up for more than 20% of the dataset.

While we keep the right marginals constant, as we have a true match for each cell in the target distribution, we introduce unbalancedness in the source marginals. It is important to note that the influence of the unbalancedness parameters are affected by the number of samples, as well as the entropy regularization parameter $\epsilon$. To demonstrate the robustness of GENOT-F with respect to hyperparameters, we still choose $\alpha = 0.7$, but this time set $\varepsilon = 5 \cdot 10^{-3}$. We use 50-dimensional PCA-space for the Gromov term in the RNA space, 38-dimensional LSI-space for the Gromov term in the ATAC space, and a 30-dimensional VAE-embedding for the shared space.

Source: ATAC space

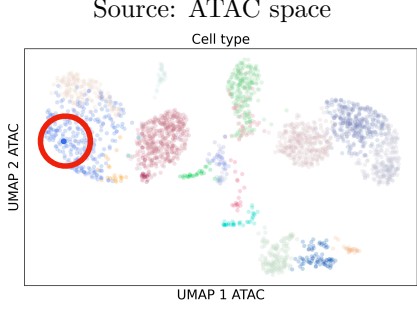

Target: RNA space

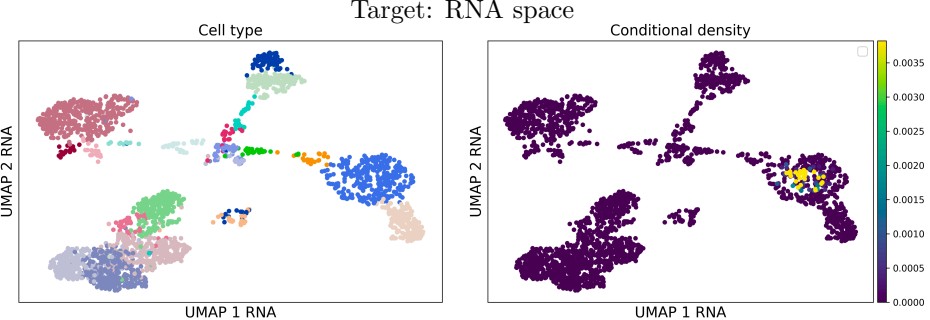

Figure 23: Top: A single Erythroblast cell is highlighted while all other cells in the source distribution are depicted with increased transparency. Bottom: Target distribution colored by cell type (left) and colored by the conditional density (conditioned on the highlighted cell above) evaluated on all cells in the target distribution. All cells with a relatively high density belong to the Eythroblast cluster, hence the mapping learnt by GENOT-F is meaningful.

| model ($\tau_1$) | Normoblast | Erythroblast | Proerythroblast | other | FOSCTTM |
|---|---|---|---|---|---|
| Discrete UFGW (0.8) | 0.788 | 0.820 | **0.842** | **0.945** | 0.258 |
| U-GENOT-F (0.8) | **0.622** | **0.733** | 0.894 | 1.077 | **0.131** |
| Discrete UFGW (0.3) | 0.591 | 0.586 | 0.734 | 0.761 | 0.311 |
| U-GENOT-F (0.3) | **0.295** | **0.430** | **0.554** | **1.186** | **0.162** |

Table 4: Mean value of the rescaling function per cluster for U-GENOT-F and discrete unbalanced FGW together with the FOSCTTM scores across three runs. Tab. 5 reports the variances for the GENOT-Q models.

The computation of the growth rates for the discrete setting is described in appendix F.2. We perform a random 60-40 split to divide the data into training and test set. The FOSCTTM score only considers those cells which have a true match, i.e. cells in the source distribution belonging to the Normoblast, Erythroblast, and Proerythroblast cell types are not taken into account as their true match was removed from the target distribution.

We assess the performance w.r.t. the FOSCTTM score to ensure that the model still learns meaningful results, and consider the average reweighting function $\hat{\eta}$ per cell type (appendix E.2). We consider two values ($\tau_1 = 0.8$ and $\tau_1 = 0.3$) of the left unbalancedness parameter, while $\tau_2 = 1.0$ as for every cell in the target distribution there exists the true match in the source distribution. Tab. 4 shows that U-GENOT-F learns more meaningful reweighting functions than discrete UFGW as the average rescaling function on the left-out cell types is closer to $0$, while the mean value of the rescaling function on all remaining cell types ("other") is closer to 1. At the same time, U-GENOT-F yields lower FOSCTMM scores and hence learns more optimal couplings.

Tab. 5 shows the variance across three runs, demonstrating the stability of both the learnt rescaling function as well as the performance with respect to the FOSCTTM score.

| model ($\tau_1$) | Normoblast | Erythroblast | Proerythroblast | other | FOSCTTM |
|---|---|---|---|---|---|
| U-GENOT-F (0.8) | $3 \cdot 10^{-6}$ | $2 \cdot 10^{-5}$ | $1 \cdot 10^{-4}$ | $5 \cdot 10^{-5}$ | $3 \cdot 10^{-5}$ |
| U-GENOT-F (0.3) | $2 \cdot 10^{-6}$ | $9 \cdot 10^{-6}$ | $5 \cdot 10^{-4}$ | $8 \cdot 10^{-4}$ | $9 \cdot 10^{-5}$ |

Table 5: Comparison of reweighting functions learnt by U-GENOT-F and discrete unbalanced FGW.

Fig. 24 shows the mean and the 95% confidence interval of the learnt growth rates per cell type. First, it is interesting to see that Normoblasts have the lowest mean of rescaling function evaluations (for both discrete UFGW and U-GENOT-F), which is due to them being most mature among the left out cell types and hence being furthest away in gene expression space / ATAC space from the common origin of all cells, the HSC cluster. Moreover, it is obvious that the 95% confidence interval of the reweighting function (across cells in one cell type) is much smaller for U-GENOT-F than for discrete UFGW. This is desirable as cells within one cell type are very similar in their ATAC profile.

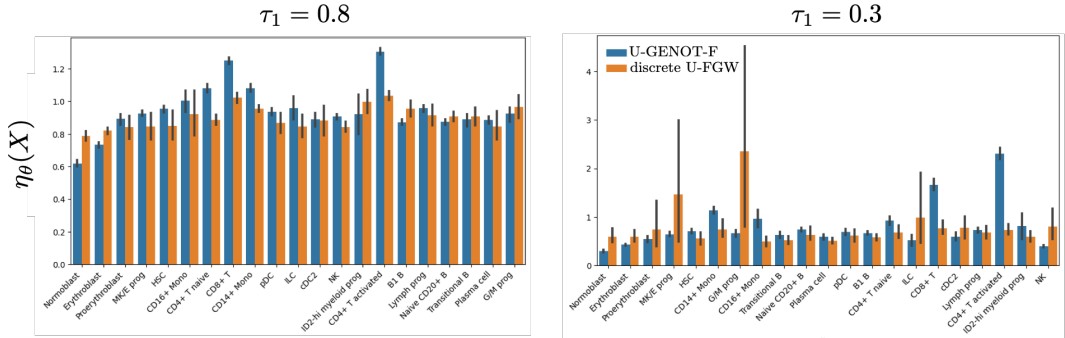

Figure 24: Comparison of learnt growth rates of discrete UFGW and U-GENOT-F aggregated to cell type level for unbalancedness parameters $\tau_1 = 0.8, \tau_2 = 1$ (left) and $\tau_1 = 0.3, \tau_2 = 1$ (right).

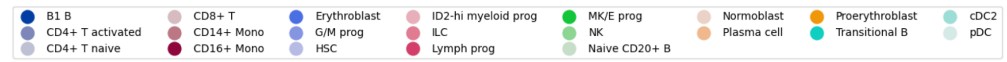

Figure 25: Complete legend of cell types for Fig. 4, Fig. 22 and Fig.23.

# F Competing methods

## F.1 EOT benchmark in the balanced, linear case

As results for competing methods were taken from Gushchin et al. [30], we refer to the original publication for details on the competing methods. For SB-CFM we follow the setup in the authors' tutorial in `https://github.com/atong01/conditional-flow-matching/blob/main/examples/single_cell/single-cell_example.ipynb`.For DSBM, we follow the setup in `https://github.com/yuyang-shi/dsbm-pytorch/blob/main/DSBM-Gaussian.py`, but increase the number of inner iterations from 10 to 50, the number of outer iterations from 40 to 80, and the btach size from 128 to 2048 to account for higher dimensionality.

## F.2 Regression for out-of-sample data points

Out-of-sample prediction for GW has been considered in [1]. Yet, their methods rely on an orthogonal projection, which only works if both the sample size and the feature dimensions are the same in both spaces. Hence, we rely on a barycentric projection for in-sample data points. For out-of-sample data points we project a data point onto the training set and apply the barycentric projection to the linear combination of points in the in-sample distribution. Let $\mathbf{X} \in \mathbb{R}^{n \times d}$ be the matrix containing $n$ in-sample data points.

Then, for a data point in the source distribution $\mathbf{x} \in \mathbb{R}^d$, let

$$\hat{\beta}_{\mathbf{x}} = \arg \min_{\beta \in \mathbb{R}^n} \|\hat{\mathbf{x}} - \mathbf{X}^T \beta\|_2^2 \tag{40}$$

where the sum is taken over the $n$ in-sample data points. Moreover, let $p_i = \sum_{j=1}^m \Pi_{ij}$. Then, the barycentric projection of a point in the source distribution is given as

$$\hat{\mathbf{y}} = \sum_{i=1}^n \frac{\hat{\beta}_i}{p_i} \sum_{j=1}^m \Pi_{ij} \mathbf{y}_j \in \mathcal{Y}. \tag{41}$$

Similarly, we can apply this procedure to estimate rescaling factors in the unbalanced setting. To ensure non-negativity of the rescaling function, we perform regression with non-negative weights:

$$\hat{\alpha} = \arg \min_{\alpha \in \mathbb{R}_{\geq 0}^n} \|\hat{\mathbf{x}} - \mathbf{X}^T \alpha\|_2^2 \tag{42}$$

To estimate the rescaling function for a data point $\hat{x}$, the estimated left rescaling function is given as

$$\hat{\eta} = \sum_{i=1}^n \hat{\alpha}_i \eta_i \in \mathbb{R} \tag{43}$$

where $\{\eta_i\}_{i=1}^n$ is the set of reweighting function evaluations of in-sample data points.

# G    Implementation

The GENOT framework is implemented in `JAX` [6]. We use the discrete OT solvers provided by `OTT-JAX` [14].

## G.1    Parameterization of the vector field

The vector field is parameterized with a feed-forward neural network which takes as input the time (which is cyclically encoded with frequency 128), the condition (i.e. the samples from the source distribution) and the latent noise. Each of these input vectors are independently embedded by one block of layers before the embeddings are concatenated and applied to another block of layers, followed by one output layer. If not stated otherwise, one block of layers consists of 8 layers of width 256 with *silu* activation functions.

## G.2    Parameterization of the rescaling functions

Rescaling functions are parameterized as feed-forward neural networks with 5 layers of width 128, followed by a final *softplus* activation function to ensure non-negativity.

## G.3    Training details

We report default values for the different parameters of the GENOT Alg.1. If not stated otherwise in the corresponding experiments section, we use these parameters. As in Alg.1, parameters related to U-GENOT are provided in teal.

- **Batch size**: $n = 1024$.
- **Entropic regularization strength**: $\varepsilon = 10^{-2}$. By default, we do not scale the cost matrices passed to discrete OT solvers.
- **Unbalancedness parameter**: $\tau = (1,1)$. This means that by default, we impose the hard marginal constraints.
- **Number of training iterations**: $T_{\text{iter}} = 10,000$.
- **Optimizer**: AdamW with **learning rate** $\text{lr} = 10^{-4}$, and **weight decay** $\lambda = 10^{-10}$. This is used for both, the vector field $v_{t,\theta}$, and the re-weighting functions $\eta_\theta, \xi_\theta$.

When using the graph distance, we construct a k-nearest neighbor graph with $batch\_size$ number of edges. Then we apply a negative exponential kernel to obtain connectivties from the distances in the knn-graph. For the approximation of the heat kernel, we use the default parameters provided by the implementation in OTT-JAX [14].

## G.4    Code availability

The GENOT model along with the code to reproduce the experiments can be found at `https://github.com/MUCDK/genot`, while a more modular implementation can be found in OTT-JAX [14]. Additionally, we implement applications in moscot [39].

# H    Impact statement

This work introduces novel approaches in neural optimal transport and presents applications in single-cell biology. Leveraging single-cell genomics data holds promise for advancing personalized medicine but requires cautious handling due to its potential inclusion of sensitive information. Moreover, as neural optimal transport techniques find applicability across diverse domains, the societal ramifications of this research extend beyond single-cell genomics. Our intention is to provide access to the GENOT code, enabling its utilization by both general machine learning practitioners and, in a subsequent release, ensuring its accessibility to advance and accelerate research specifically tailored for single-cell biologists.

