# OpenReview forum: "GENOT: Entropic (Gromov) Wasserstein Flow Matching with Applications to Single-Cell Genomics"
_NeurIPS.cc/2024/Conference — NeurIPS 2024 poster_

### Official Review · Reviewer_GBWj · 2024-07-10

**Soundness:** 3
**Presentation:** 3
**Contribution:** 3
**Rating:** 7
**Confidence:** 3

**Summary:**

The paper proposes a multi-variable OT-based framework for solving Single-Cell related problems. The main idea behind the approach is to use the solutions for different variants of discrete entropy-regularize OT problems to learn a continuous parametric plan (measure) given by its conditional distributions, which approximates the true solution to the corresponding continuous problem. The authors propose utilizing conditional Flow Matching to parameterize the learned plan. The methodology is validated on various synthetic and real (single-cell) use cases. The obtained results testify the competitiveness of the framework.

**Strengths:**

Despite the fact that the paper is full of specific technical details on single-cell stuff, the manuscript is clear and pleasant for reading.

The method is clearly introduced. It seems that the authors adequately and fairly exhibit both the advantages of the method - its universality (coverage of different variants of OT formulations), simplicity (both for understanding and I guess for implementation); and limitations of the methodology - good limitation section + discussed reliance on discrete OT.

Also, I would like to commend a good experimental section - a lot of interesting use cases are considered and interesting results (sometimes even negative) are presented. These experiment settings are worth considering regardless of the proposed method.

**Weaknesses:**

* It seems the method could not (or hardly could)  be adapted to high-dimensional (e.g., image-data) setups due to reliance on discrete EOT. At least, I expect that the learned plan in such setups will significantly deviate from the true EOT plan. Still, I understand that the proposed framework is targeted on single cell data, which is not that high-dimensional.
* Reliance on Discrete OT is something I personally treat as a weakness. It introduces biases which hardly could be analyzed. Still, in practice, the minibatch OT seems to work well.

**Questions:**

* The main of my questions is about the utilization of (conditional) flow matching for learning the parametric plan out of discrete minibatches. Why flow matching? As I understand, the proposed framework could be used on par with other generative techniques like GANs/normalizing flows/diffusion models. Why do you restrict yourself by flow matching?
* I found it interesting that GENOT achieves (on average) the best results on EOT benchmark [3] among the competitors. Could the authors provide the code for the benchmark experiment?
* Unbalanced Entropic OT work (to be mentioned): [1]. Also, a recent preprint you may find to be interesting: [2].
* Line 89: Do I understand correctly that $\mu$ and $\nu$ are not necessarily probability distributions, but could be positive measures with the same mass?
* Why does in (UQEOT) one utilize $\varepsilon \text{KL}^{\otimes}$, while in (QEOT) - just  $\varepsilon \text{KL}$ (without $\otimes$)?
* Line 289: What is the “conditional mean” regime of the GENOT model?
* Why does the unfused methods work badly on the single cell modalities translation problem with $\ell_2^2$ intra-costs (Figure 4, FOSCTTM metric is almost everywhere near 0.5)?

Misprints:
- (QEOT) formula: $Q_{c_X, c_Y}$ -> $D_{c_X, c_Y}$
- line 101: complexity if -> complexity is
- line 351: Fig. 3 -> Fig. 4

[1] Pariset et. al., Unbalanced Diffusion Schrödinger Bridge, ICML’23 workshop.

[2] Gazdieva et. al., Light Unbalanced Optimal Transport.

[3] Gushchin et. al., Building the Bridge of Schrödinger: A Continuous Entropic Optimal Transport Benchmark

------------------------

POST REBUTTAL

I thank the authors for the answers provided. I am satisfied with them. I have decided to raise my grade.

**Limitations:**

Ok

---

> ### Author Response · Authors · 2024-08-02
> **Additional Answer to Reviewer GBWj**
>
> > ***What is the “conditional mean” regime of the GENOT model?***
>
> ➤  This refers to mapping a point $\mathbf{x}$ to the target domain via the conditional mean, i.e., by averaging multiple samples from the conditional distribution $\pi^\star_\varepsilon(\cdot|\mathbf{x})$ instead of taking a unique one $\mathbf{y}\sim\pi^\star_\varepsilon(\cdot|\mathbf{x})$. This is detailed in lines 134-138. While we would like to highlight that taking the conditional mean has to be taken with a grain of salt (as in most cases, this is not the quantity we are interested in when computing EOT plans), it can be beneficial in some single-cell applications with respect to certain metrics. For instance, in Figure 4 (translating modalities), we see that the FOSCTTM of GENOT-Q CM is lower (hence better) than that of GENOT-Q. Importantly, numerous single-cell applications explicitly model the barycentric projection (i.e., conditional mean), e.g., for translating modalities of cells ([Demetci+2022]), mapping cells to their spatial organization [Nitzan+2019], and aligning slides of spatial transcriptomics [Zeira+2023].
>
>
> > ***Why does the unfused methods work badly on the single cell modalities translation problem with l2  intra-costs (Figure 4, FOSCTTM metric is almost everywhere near 0.5)?***
>
> ➤  This is an interesting question. It seems that pairwise Euclidean distances are not sufficiently characteristic for cells. The suitability of the cost might depend a lot on the dataset. These observations are also observed in the original publication for using Gromov-Wasserstein for aligning modalities of cells (SCOT, [Demetci+2022], Fig. 6). Unfortunately, we are not aware of any method on how to choose the “best” cost a priori, but our experiments (as well as the experiments shown in Fig.1, pdf) show that the choice of the cost can have an impact on the performance.  Note that the poor performance of using the squared Euclidean cost motivated our "extension" of SCOT, i.e. constructing a fused term which empirically helps.
>
>
> > ***Misprints.***
>
> Thanks a lot for catching these typos. We adapted the relevant sections in the paper.

---

> ### Author Rebuttal · Authors · 2024-08-07
>
> > ***The method is clearly introduced [...] These experiment settings are worth considering regardless of the proposed method.***
>
> ➤ We thank the reviewer for appreciating our contributions.
>
> > ***It seems the method could not (or hardly could) [...] which is not that high-dimensional.***
>
> ➤ We appreciate that the reviewer understands that our method is motivated by and targeted at single-cell data. As mentioned in the general response, we found this question very interesting and applied GENOT to image data. Again, we would like to highlight that we do not intend to include the results in the paper and leave further investigation and optimization of GENOT applied to image data to future work, as i) we believe it does not fit the story line, ii) reviewers appreciated the large number of experiments, but also reviewer GAFl also critized that such a large number of experiments is in the appendix, hence adding computer vision examples would hamper readability even more. In addition to what we wrote in the general response, we would like to add the following: We follow the same hyperparameter and architectural setup as the authors in [Eyring+24]. We only added the FiLM condition module and modified the number of iterations to 350k. Moreover, we consider the balanced scenario, because - as mentioned - we did not perform any hyperparameter optimization.
>
> > ***Reliance on Discrete OT is something I personally treat as a weakness. It introduces biases which hardly could be analyzed. Still, in practice, the minibatch OT seems to work well.***
>
> ➤ We respect that the reviewer considers the reliance on discrete OT as a weakness. We would also like to highlight that by no means we assume there is no better way to solve some of the tasks which does not rely on discrete OT. Instead, we would like to encourage the community to build upon our method, as we hope to clearly motivate the challenges which are faced in single-cell genomics while providing in some of these cases a first method which can be benchmarked against. There are clear limitations of our method, which we explicitly discuss in the appendix A or can be directly seen from the results. Yet, as stated by the reviewer, we also see that in some cases the performance of GENOT is good. We thank the reviewer for appreciating that the results of the experiments justify the usage of discrete OT.
>
> > ***The main of my questions is about the utilization of (conditional) flow matching for [...]. Why do you restrict yourself by flow matching?***
>
> ➤ We do agree that our proposed method could be built with any generative model. The choice of flow matching is mainly practical, but it also has some methodological advantages. First and most importantly, training is fast (simulation-free as opposed to normalizing flows) and stable due to its simple minimization objective (as opposed to GANs). Moreover, we experienced diffusion models to be harder to train on single-cell data (worse performance in terms of distributional matching). On the methodological side, flow matching allows for estimating densities (as opposed to GANs), which can be useful in single-cell genomics as demonstrated in Figure 21.
>
> > ***I found it interesting that GENOT achieves (on average) the best results on EOT benchmark [3] among the competitors. Could the authors provide the code for the benchmark experiment?***
>
> ➤ We provide the code in an **anonymous** Google Drive: [https://drive.google.com/drive/folders/17DUZr_bnTY4gv4nd8szk1h0wJ5NdQjlH?usp=sharing](https://drive.google.com/drive/folders/17DUZr_bnTY4gv4nd8szk1h0wJ5NdQjlH?usp=sharing). The folder `genot_rebuttal` includes a self-contained notebook `benchmark_example_64_1.ipynb` to reproduce an example of the results for the benchmark pair in dimension $d=64$ and $\varepsilon=1$. We also included a file `requirements.txt` containing the package requirements.
>
>
> > ***Unbalanced Entropic OT work (to be mentioned): [1]. Also, a recent preprint you may find to be interesting: [2].***
>
> ➤ We apologize for having missed this work, and now explicitly mention the work by replacing the sentence in line 56-59: "[Eyring+2024], [20], [Lübeck+2023], [Yang+2019] proposed a way to incorporate unbalancedness into deterministic linear OT maps, while unbalanced formulations for entropic OT in both the linear and the quadratic case have not been explored.” by [Eyring+2024], [20], [Lübeck+2023], [Yang+2019] proposed a way to incorporate unbalancedness into deterministic linear OT maps, while [Yang+2019] extended their method to the entropic setting (but do not provide code), while recent preprints [Pariset+2023], [Gazdieva+2024] also suggested methods to approximate unbalanced linear EOT solutions. To the best of our knowledge, unbalanced neural formulations in quadratic OT have not yet been explored."
>
> > ***Line 89: Do I understand correctly that \mu and \nu are not necessarily probability distributions, but could be positive measures with the same mass?***
>
> ➤ In the balanced OT setting, $\mu$ and $\nu$ are taken as probability measures, i.e., their mass is 1. This extends to all measures with arbitrary but equal masses, as they can always be normalized, *with the same renormalization constant*, without losing generality. The unbalanced setting is used for positive $\mu$ and $\nu$ measures with different masses.
>
> > ***Why does in (UQEOT) one utilize,  while in (QEOT) - just  (without )?***
>
> ➤ This is a very good question. For QEOT, we do not need to use $\mathrm{KL}^\otimes$ because, when $\mu$ and $\nu$ have the same total mass, as shown in [Séjourné+2024, Prop 8.], $\mathrm{KL}^\otimes(\pi|\mu\otimes\nu) = 2 \mathrm{KL}(\pi|\mu\otimes\nu)$. Therefore, using $\mathrm{KL}$ is equivalent to using $\mathrm{KL}^\otimes$, with the adjustment of replacing $\varepsilon$ by $\varepsilon/2$.
>
> For the remaining comments, we would like to kindly ask the reviewer to read the "Additional Answer to Reviewer GBWj" added.

---

> ### Comment · Reviewer_GBWj · 2024-08-08
> **Thanks for the answers**
>
> I thank the authors for the provided answers. I have decided to raise the score (up to 7)

---

> > ### Author Response · Authors · 2024-08-10
> > **Many thanks for reading our rebuttal.**
> >
> > We are very grateful for your score increase, and we are happy to answer any further question on our work.
> >
> > The Authors

---

### Official Review · Reviewer_NoBU · 2024-07-12

**Soundness:** 3
**Presentation:** 3
**Contribution:** 3
**Rating:** 6
**Confidence:** 4

**Summary:**

The paper proposes a framework for realigning cells in single-cell genomics which lies in the field of neural Optimal Transport (OT) solvers. The method utilizes a generative flow-based model for computing entropic OT couplings and tackles several practical challenges, e.g., it allows for using arbitrary cost functions, learning stochastic transport plans, relaxing mass conservation constraint, and tackling challenges of the (Fused) Gromov-Wasserstein problem. Specifically, the method consists of training of the conditional flow matching model on top of estimated discrete entropic OT plans. The approach is tested in different tasks from the single-cell biology field.

**Strengths:**

The paper proposes a flexible framework which allows for addressing different challenges of application of neural OT solvers in the single cell genomics field. It is shown that the approach provides good results in several tasks from the single cell genomics according to the metrics under consideration. The paper is well-written and structured.

**Weaknesses:**

My major concern is that the provided approach is not guaranteed to learn the ground-truth plans. Most of the experimental details and theoretical results related to this topic are moved to Appendix which hampers the understanding of this question.

Indeed, GENOT is based on the idea of distilling discrete entropic OT (EOT) solutions using the flow matching (which is close, but as the authors explain - different, to the idea from the work [1]). It raises a question: how well does such a kind of distillation of discrete plans approximate the ground-truth continuous ones (assuming that the approach works with batches of finite size)? Some explanations regarding the mini-batch biases are provided in lines 267-280. However, as far as I see it, the provided discussion (and references) do not cover the statistical properties of the discrete *unbalanced* linear or *quadratic plans*. The situation with quadratic setup is the most tricky here, since it is known that GW problem admits multiple solutions and, thus, the optimization problem may differ at each step of training depending on the calculated discrete entropic GW plans.

From the empirical side, the authors address the ‘biasedness’ issue by testing their approach using a benchmark with analytically known EOT [2], U-EOT [3], and GW/Unbalanced GW [4] plans (all in Appendix E). According to these experiments, I tend to agree that the proposed approach provides meaningful results for the balanced (or unbalanced) linear case. However, the situation with the GW case is not clarified – according to the experiment description (Appendix E.3), the authors introduce some heuristics to “preserve the orientation of GW solution” and claim that in the case of biological tasks it is not needed. (Still, as a non-expert in biology, I can not assess the validity of this claim.)

**In summary**, there are some evident issues with the implementation of the proposed approach in the Gromov-Wasserstein case. I appreciate that the authors mentioned this issue in the limitations section of their work (Appendix A). However, it is not obvious for me to what extent this is a serious drawback hampering the applicability of the approach to biological tasks. Besides, I think that as an important aspect of the implemented approach, this issue should be mentioned in the main body of the paper.

**Questions:**

- In the related work section you cited several works on unbalanced EOT [5,6,7]. Why did you include only one of them [7]  in comparison in Gaussians experiment?

- In Table 1-3 you use abbreviations DBSM and CFM-SM which are not introduced in the text. Please introduce them in the relevant section.

- Could you explain in more details why the issues of your implementation in the GW case will not arise in biological tasks? I see your explanation in lines 1345-1346, however, it seems that the issue might arise not only in the case of highly symmetric distributions

**Limitations:**

The authors adequately addressed the limitations and potential negative societal impact of their work in a specific section of their work.

---

> ### Author Response · Authors · 2024-08-05
> **Additional Answer to Reviewer NoBU 1/2**
>
> > ***The situation with quadratic setup is the most tricky here, since it is known that GW problem admits multiple solutions and, thus, the optimization problem may differ at each step of training depending on the calculated discrete entropic GW plans.***
>
> ➤ We agree that the quadratic setup is the most tricky one, and we thank the reviewer for explicitly mention this issue again. As mentioned in the general response, we set out to investigate this further (Alg. 1, pdf), and results can be found in Fig. 2, pdf.
>
> We consider two datasets:
>
> 30-dim. Gaussians (20 dim. quadratic + 10 dim. fused term): For both the source ($\mathcal{\mu}$) and the target ($\nu$) distribution we sample mean vectors from $\mathcal{U}([-1, 1])$ and construct the covariance matrices by sampling each entry of their square roots from $\mathcal{U}([0, 3])$.
> 30-dim. single-cell data (the one used for modality translation, but no train/test split): 20-dimensional PCA of the normalized ATAC data, 20-dimensional PCA projection of the gene expression data, and 10-dimensional PCA of VAE space (shared space).
> To assess the stability of the solution of discrete EOT, we fix one sample $X_0 \sim \mu$ from the source distribution, and compute the variance of $\hat{\pi}(\cdot|X_0)[10:]$, i.e. of the incomparable space $\tilde{\mathcal{Y}}$ only, which is obtained from a discrete EOT solver with different samples in source and target. We do this for $\varepsilon=0.01$ and $\varepsilon=0.0001$ with the Sq. Euclidean cost. We consider:
>
> - outer coupling: this serves as orientation.
> - Gromov-Wasserstein (GW): We use $\tilde{\mathcal{X}}$ and $\tilde{\mathcal{Y}}$, i.e., we use the last 20 dimensions of both the source and the target distribution.
> - Fused Gromov-Wasserstein (FGW): We use $\Omega \times \tilde{\mathcal{X}}$ and $\Omega \times \tilde{\mathcal{Y}}$, $\alpha=\frac{1}{11}$.
> - Gromov-Wasserstein with geodesic cost (Geodesic): We use $\tilde{\mathcal{X}}$ and $\tilde{\mathcal{Y}}$ with the geodesic cost.
> - Gromov-Wasserstein with initialisation (with init): We use $\tilde{\mathcal{X}}$ and $\tilde{\mathcal{Y}}$. We use the initialisation scheme as discussed in paper (Appendix A, E).
>
> > ***However, the situation with the GW case is not clarified – according to the experiment description (Appendix E.3), the authors introduce some heuristics to “preserve the orientation of GW solution”***
>
> ➤ We visualise what we mean by "preservation of the orientation" of the GW solution in Fig. 1 in the provided pdf, where we transport a 1-dimensional Gaussian to a 1-dimensional Gaussian. The LHS of Fig. 1 shows the result obtained without initialisation, and the learnt coupling is clearly a mixture of solutions, while on the RHS of Fig. 1, we visualise that GENOT-Q learns a coupling which appears to be no mixture of solutions. While this is only an example in 1 dimension, the results in Fig. 2, pdf, highlight that using the proposed initialisation scheme stabilises the orientation of the GW solution also in high dimensions, for both the Gaussian and the single-cell experiments.
>
>
> > ***[...] in the case of biological tasks it is not needed. (Still, as a non-expert in biology, I can not assess the validity of this claim.)***
>
> ➤ We thank the reviewer for pointing this out, apologize for this bold statement, and would like to correct ourselves and weaken the statement to: "In the case of single-cell data, choosing the right cost might help to overcome the need for the initialisation scheme to a certain degree." as can be seen from Fig. 2, pdf, where the geodesic cost seems to help make the solution more unique with $\varepsilon=0.0001$, as well as the results in Fig. 4, manuscript, LHS. Moreover, we will add the sentence "Additionally, most single-cell applications are a FGW problem (as opposed to a GW problem), and empirically, the fused term helps to make the set of solutions of the EOT coupling less diverse, as can be seen from Fig. 5, FIg. 19, and [provided pdf, Fig. 2].". Indeed, the only pure GW problem in common single-cell applications we are aware of is SCOT [Demetci+22]. As we realised the performance of both SCOT and GENOT-Q is poor, we introduced a novel way to include a fused term into the task of modality translation by constructing a joint space by creating a joint conditional VAE embedding from gene activity (source space) and gene expression (target space).
>
> > ***However, it is not obvious for me to what extent this is a serious drawback hampering the applicability of the approach to biological tasks.***
>
> ➤ We hope to have clarified this with the additional experiments and the answers above.

---

> ### Author Response · Authors · 2024-08-05
> **Additional Answer to Reviewer NoBU 2/2**
>
> > ***Besides, I think that as an important aspect of the implemented approach, this issue should be mentioned in the main body of the paper.***
>
> ➤ We fully agree with the reviewer, and will definitely include the discussion mentioned above in the main body of the paper, highlighting this limitation. Moreover, we will include the results on the stability of quadratic EOT couplings as outlined in the provided pdf document.
>
> > ***In the related work section you cited several works on unbalanced EOT [5,6,7]. Why did you include only one of them [7] in comparison in Gaussians experiment?***
>
> ➤ We assume the reviewer is alluding to lines 56-59: [Eyring+2024], [Lübeck+2023], [Yang+2019] proposed a way to incorporate unbalancedness into deterministic linear OT maps, while unbalanced formulations for entropic OT in both the linear and the quadratic case have not been explored.” In fact, [Eyring+2024], [Lübeck+2023] are unbalanced Monge Maps (i.e. deterministic) estimators, and hence do not solve EOT. [Yang+2019] suggest a solver for both deterministic and entropic OT couplings, but their implementation only allows for the deterministic OT coupling. We now clarify this, and also include references to other preprints (thanks to reviewer GBWj) by adapting the statement in lines 56-59 to: "[Eyring+2024], [Lübeck+2023], [Yang+2019] proposed a way to incorporate unbalancedness into deterministic linear OT maps, while [Yang+2019] extended their method to the entropic setting (but do not provide code), while recent preprints [Pariset+2023], [Gazdieva+2024] also suggested methods to approximate unbalanced linear EOT solutions. To the best of our knowledge, unbalanced neural formulations in quadratic OT have not yet been explored."
>
> > ***In Table 1-3 you use abbreviations DBSM and CFM-SM which are not introduced in the text. Please introduce them in the relevant section.***
>
> ➤ We thank the reviewer for pointing us to this. We will add all references to Table 1.
>
> > ***Could you explain in more details why the issues of your implementation in the GW case will not arise in biological tasks? I see your explanation in lines 1345-1346, however, it seems that the issue might arise not only in the case of highly symmetric distributions***
>
> ➤ As mentioned above, this has two main reasons: First, empirically, the geodesic cost seems to overcome this limitation to a certain extent (see superior performance in Fig. 4, paper, and lower conditional variance in Fig.2, pdf). Second, most single-cell applications can be extended to a fused problem. Yet, we apologize for our formulation that this issue only arises in highly symmetric contributions, and will correct the statement to "Empirically, both changing the cost and including a fused term can help to account for this limitation. If none of these approaches work, GENOT can be run with its quadratic initialisation scheme, which empirically helps to make the set of EOT solutions less diverse, at the cost of loosing simulation-free training."

---

> ### Author Rebuttal · Authors · 2024-08-07
>
> > ***It is shown that the approach provides good results in several tasks from the single cell genomics according to the metrics under consideration. The paper is well-written and structured.***
>
> ➤ We thank the reviewer for the positive feedback.
>
> > ***My major concern is that the provided approach is not guaranteed to learn the ground-truth plans.***
>
> ➤ We agree with the reviewer that distilling mini-batch coupling introduces a bias, which means that we can, in theory, only learn the true coupling in the asymptotic setting of infinite batch size. We discussed this aspect in detail in lines 267-280. On the other hand, it's important to remember that we are not aiming to learn a deterministic Monge map or a Benamou-Brenier velocity field. Instead, we operate in the entropic regime with $\varepsilon \gg 0$. In this regime, the estimation rates are parametric in $\varepsilon$, helping to avoid the curse of dimensionality and significantly reducing the bias introduced compared to estimating deterministic maps (see the discussion related to statistical estimation below.)
>
> Moreover, solving a continuous OT problem in practice is very challenging, and most neural OT techniques introduce either a bias or a significant computational challenge into the learning process. In this paper, we have developed a method that is easy to train, even though it introduces a bias. This approach proved effective, as GENOT outperforms 12 baselines on a recent benchmark [Gushchin+2023], where we estimate a ground truth linear EOT known in closed form (see Tables 1-2).
>
> Finally, we want to emphasize that, to the best our knowledge, there is only one other (unpublished) neural OT solver targeting quadratic OT, which comes with a number of limitations, especially for single-cell genomics. As stated in lines 62-64 of the manuscript: *"To our knowledge, the only neural formulation for GW proposed so far by [57] learns deterministic and balanced maps for inner product costs, using a min-max-min optimization procedure, which severely limits its application in single-cell genomics."* In general, solving a quadratic OT problem is extremely challenging. The only case where we can satisfactorily approximate GW couplings is in the discrete setting, using [Peyré+2016]'s scheme. Therefore, with GENOT, we aimed to develop a method that builds on these discrete GW couplings.
>
> All in all, we would also like to highlight that by no means we assume there is no better way to solve some of the tasks which does not rely on discrete OT. Instead, we would like to encourage the community to build upon our method, as we hope to clearly motivate the challenges which are faced in single-cell genomics while providing in some of these cases a first method which can be benchmarked against.
>
> > ***Most of the experimental details and theoretical results related to this topic are moved to the Appendix, which hampers the understanding of this question.***
>
> ➤ We agree with the reviewer that it is not ideal that experimental details and theoretical results are moved to the appendix. As the development of GENOT arose from the need for flexible neural OT estimators in single-cell genomics, we decided to highlight the applied point of view. We admit that the readability of the paper would benefit from having e.g. more experiments or propositions in the main body. Yet, as we hope this paper will motivate single-cell biologists to build upon these methods for analyzing their data, we prioritised the application, but we still hope to include a few more experimental details and/or theoretical results in the final version of the manuscript.
>
> > ***How well does such a kind of distillation of discrete plans approximate the ground-truth continuous ones? [...] Some explanations regarding the mini-batch biases are provided in lines 267-280. However, as far as I see it, the provided discussion (and references) do not cover the statistical properties of the discrete unbalanced linear or quadratic plans.***
>
> ➤ This is a very good question. Most papers dealing with the estimation of EOT objects provide estimation rates for (Gromov)-Wasserstein distances. As stated in lines 267-281:
> - In the linear EOT setting, both balanced [Genevay+2019] and unbalanced [Séjourné+2021] settings achieve a parametric rate in $\varepsilon$ dodging the curse of dimensionality.
> - In the quadratic EOT setting, for the balanced case, [Zhang+2023] demonstrate that similar parametric rate in $\varepsilon$ can also be achieved. They prove this result by linearization of the quadratic EOT problem. Specifically, this rate depends on the minimum dimension of the source and target domains, indicating that estimation is easier when one of the domains is low-dimensional.
>
> There is only one very recent paper [89] that provides a rate estimate for the ground truth coupling $\pi^\star_\varepsilon$ by the empirical coupling $\hat{\pi}^n_\varepsilon$ in the balanced linear EOT setting. This paper shows the same type of parametric rate in $\varepsilon$, mitigating the curse of dimensionality.
>
> Extending this result to (i) the unbalanced EOT setting and (ii) the quadratic EOT setting is a very exciting perspective. For (i), since we have shown that an unbalanced EOT coupling is essentially a balanced EOT coupling between re-scaled measures (see Prop B.1), this result seems intuitively extendable. For (ii), the extension could be achieved by linearizing the quadratic EOT problem, similar to the approach used by [Zhang+2023] for costs.
>
> For the remaining comments, we would like to kindly ask the reviewer to read the "Additional Answer to Reviewer NoBU 1/2" and "Additional Answer to Reviewer NoBU 2/2" added.

---

> > ### Comment · Reviewer_NoBU · 2024-08-13
> >
> > I thank the authors for providing the answers to my concerns and raise my score to 6.

---

> > > ### Author Response · Authors · 2024-08-13
> > > **Thanks for the response to our rebuttal**
> > >
> > > We are glad we could address all concerns and are more than happy to address any further questions you may have about our work.
> > >
> > > The Authors

---

### Official Review · Reviewer_GAfL · 2024-07-12

**Soundness:** 3
**Presentation:** 3
**Contribution:** 3
**Rating:** 6
**Confidence:** 3

**Summary:**

The authors present Generative Entropic Neural Optimal Transport (GENOT), a flexible method for learning entropic couplings for linear and quadratic entropic optimal transport (EOT), unbalanced OT, and OT across incomparable spaces. At its core, their method uses conditional flow matching (CFM) to solve for the OT couplings. Through extensive empirical experiments, the authors show the use-case of their method for learning couplings of cell trajectories, predicting single-cell response to perturbations, and translating between single-cell data modalities.

**Strengths:**

The authors present promising and comprehensive set of empirical results that support the applicability of their method for OT related problems in single-cell biology. They demonstrate a novel use case of learning OT couplings between different data modalities in singel-cell biology, i.e. from RNA-seq to ATAC-seq. Their proposed method is also supported with sound theoretical background and justification. In general, the paper is well motivated and addresses important and challenging problems in single-cell biology.

**Weaknesses:**

Overall, this work has all the elements that constructs a complete and thorough application-based contribution to the field of neural optimal transport and single-cell biology. However, there remain several items that I feel hinder the understanding and impact of this work:

- It seems to me that a key novelty of this work is the use of quadratic OT for learning couplings between incomparable spaces -- i.e. from ATAC-seq to RNA-seq for the single-cell case considered in this work. The authors consider a large quantity of experiments for both linear and quadratic OT over various settings, with many of the experiment being reported in the appendix. This makes some of the results and discussion hard to follow, and overall diluting the presentation of the key items of this paper. Given the large quantity of experiments and dense content of this paper, maybe it would be beneficial to shift more focus on these aforementioned experiments and move other to the appendix. This may help the reader better understand some of the key contributions of this work.
- A large amount of items that are placed in the appendix tare referenced in the main text and seem to be important. This in general disrupts the overall fluidity of the paper. For instance, propositions in section 3 are listed in the appendix. Moreover, some experiments discussed in the main text only reference figures that are in the appendix. For example, experiments in section 5.1 "GENOT-L on simulated data" and "U-GENOT-L predicts single-cell responses to perturbations". In general, there seem to be a large proportion of experimental results that are placed in the appendix and referenced in section 5, which seem to be given equal wait to the experiment included in the main text. This makes it difficult to decipher the significance/importance of the respective results. As mentioned in my previous comment, possibly focussing the paper slightly may be beneficial to the overall presentation of this work.

**Questions:**

- For the experiment on single-cell response to perturbations, it appears there are no comparisons to baseline methods? Is the reason for this because existing baseline methods are deterministic? Is there intuition on why the stochastic approach is helpful (or necessary) for this task?

**Limitations:**

The authors discuss limitations and broader impacts in the appendix.

---

> ### Author Rebuttal · Authors · 2024-08-07
>
> > ***The authors present promising and comprehensive set of empirical results [...] and addresses important and challenging problems in single-cell biology.***
>
> ➤ We thank the reviewer for the positive feedback.
>
> > ***It seems to me that a key novelty of this work is the use of quadratic OT for learning couplings between incomparable spaces [...] This may help the reader better understand some of the key contributions of this work..***
>
> ➤ We are grateful to the reviewer for providing this constructive criticism. We acknowledge that the paper includes a large number of experiments, which could hinder the reading flow. We decided to focus on the motivation of this work, which is the need for widely applicable and flexible neural OT estimators in single-cell genomics. In particular, we tried to motivate each of the 4 necessities **N1**, **N2**, **N3**, and **N4** (see lines 67-70 of the submission) as much incrementally as possible. Yet, we will try to include more experiments of the fused Gromov-Wasserstein case in the final version of the manuscript, which is indeed the major novelty.
>
> Additionally, it is important to note that GENOT satisfying **N2**—approximating EOT coupling for **any cost function**—is a significant novelty. As detailed in lines 156-161 and illustrated in Figure 5, previous simulation-free linear EOT solvers, such as [Shi+2023], [Liu+2023], [Pooladian+2023], and [Tong+2024;a,b], are limited to the squared Euclidean cost $c(\mathbf{x},\mathbf{y}) = |\mathbf{x}-\mathbf{y}|_2^2$. This flexibility is crucial. We have demonstrated that in both linear (Figure 2) and quadratic EOT settings (Figure 4), using data-driven cost functions that approximate the geodesic distance on the data manifold yields more meaningful results from a biological perspective.
>
>
> > ***A large amount of items that are placed in the appendix are referenced in the main text and seem to be important. [...] As mentioned in my previous comment, possibly focussing the paper slightly may be beneficial to the overall presentation of this work.***
>
> ➤ We agree with the reviewer that the large number of references in the main text to the appendix might hinder readability. As mentioned above, we intended to highlight the need for flexible neural OT estimators in single-cell genomics, and thus prioritized the motivation from a single-cell point of view. We admit that the readability of the paper would benefit from having e.g. more experiments or propositions in the main body. Yet, as we hope this paper will motivate single-cell biologists to build upon these methods for analyzing their data, we prioritized the applied point of view.
>
> > ***For the experiment on single-cell response to perturbations, it appears there are no comparisons to baseline methods? Is the reason for this because existing baseline methods are deterministic?***
>
> ➤ This experiment serves as a motivating example for the need of neural unbalanced EOT estimators. The reason for not having included baselines in the perturbation prediction task is the lack of unbalanced entropic OT estimators. In fact, [Yang+2019] proposes a neural OT estimator in both a stochastic and a deterministic version, but their implementation only allows for the deterministic version. We decided to still include it in the benchmark of learning an unbalanced EOT plan between Gaussians (Figure 11) to demonstrate that learning an unbalanced EOT coupling is not trivial and cannot be replaced by estimators learning an unbalanced deterministic Monge Map.
>
> > ***Is there intuition on why the stochastic approach is helpful (or necessary) for this task?***
>
> ➤ As motivated in the introduction (line 45: “[...] cells evolve stochastically […]”), cells evolve stochastically rather than deterministically, and hence obtaining stochastic predictions is relevant to model trajectories of cells. For example, stochastic evolution is relevant in developmental single-cell data (measured across different time points) where a homogeneous progenitor population (in the most extreme case, a single fertilized egg cell) stochastically proliferates into more mature cells. This is the motivation for the use case we consider in Figure 2.
>
> Another source of stochasticity can be introduced from technical/experimental errors and biases, as we state in line 325: “imbalances might occur due to biases in the experimental setup or due to cell death”. The topic of uncertainty estimation is prevalent in single-cell genomics. For example, [Laehnemann+2020], state in “Eleven grand challenges in single-cell data science”: “Optimally, sc-seq analysis tools would accurately quantify all uncertainties arising from experimental errors and biases.” In particular, the perturbation data we consider e.g. in Figure 3, LHS, measures cells before and after perturbation. While in such experimental setups, it is common to perturb the same number of cells with each drug, the final data rarely has the same number of cells for each condition due to experimental errors. This is what motivates the calibration score in Figure 13, where we showed that cells with a high variance are mapped incorrectly (see Appendix C.2 for the description of the metric), resulting in a good calibration score.
>
>  [Laehnemann+2020] Laehnemann et al., Eleven grand challenges in single-cell data science, 2020

---

> > ### Comment · Reviewer_GAfL · 2024-08-08
> >
> > I thank the authors for their detailed response and answers to my questions. I will keep my positive score.

---

> > > ### Author Response · Authors · 2024-08-10
> > > **Thanks for the response to our rebuttal**
> > >
> > > We appreciate the positive feedback and are more than happy to address any further questions.
> > >
> > > The Authors

---

### Author Rebuttal · Authors · 2024-08-04

We would like to thank all reviewers for their encouraging feedback, constructive criticism, and thoughtful comments, as well as for pointing out typos.

In response to questions about the uniqueness of quadratic OT solutions raised by reviewer NoBU, we included an analysis of the stability of discrete OT solutions (see pdf document). We perform experiments on both simulated Gaussian data and single-cell data (from Fig. 4). In both cases the data has 20 dimensions (+10 dim. for fused exp.). The empirical analysis (Alg. 1, pdf) suggests:
1. Including a fused term in GW makes the solution more stable in both datasets (Fig. 2, pdf). Intuitively, the addition of this fused penalty on extra features $\mathbf{u},\mathbf{v}$ enables the "selection" of an optimal GW coupling. Essentially, we "trade off a bit of GW optimality" by choosing a coupling that minimizes the distortion of structural information $|c_\mathcal{X}(\mathbf{x}, \mathbf{x}') - c_\mathcal{Y}(\mathbf{y}, \mathbf{y}')|^2$ **while also minimizing** the cost on features $c(\mathbf{u}, \mathbf{v})$. Empirically, this mitigates the problem of the non-uniqueness of (pure) GW coupling to a large extent, making our procedure more stable.
2. The geodesic cost makes the solution more stable for suffiently small $\varepsilon$ in single-cell experiments, but barely in Gaussian experiments (Fig. 2, pdf)
3. The initialization scheme (App. A/E) for quadratic OT solvers makes the solution more stable (Fig. 2, pdf)

We intend to include these results in the final version of the manuscript.
For experimental details and a more detailed discussion, we refer the reader to the responses to reviewer NoBU.

While we thank reviewer GBWj (as well as reviewers GAfL and NoBU) for appreciating that GENOT is built for single-cell data, reviewer GBWj expressed skepticism about the applicability of GENOT to high-dimensional data like image data. We agree with reviewer GBWj that GENOT is built for single-cell data specifically, but we found the question interesting and thus conducted experiments on image data. Importantly, we would like to highlight that we do **not intend to include computer vision experiments in the final manuscript**, as in accordance with all reviewers, GENOT is built for single-cell data. We set out to apply GENOT-L to the common task of image translation on the CelebA dataset, translating females to males. We leveraged the flexibility of using any cost function and used CLIP [Radford+21] embeddings for both the discrete matching and the conditioning (together with FiLM [Perez+17]).

While FID scores and examples of translated images can be found in the provided pdf, we made the following observations:
1. In terms of FID, GENOT-L performs comparably, but slightly worse than other flow-matching based methods (Table 1, pdf)
2. Visually, the generated images look realistic, but the coupling information is not always well preserved (Fig. 3, 4, pdf), as predicted by reviewer GBWj.



[Liu+14] Liu et al. “Deep Learning Face Attributes in the Wild.”, 2014

[Perez+17] Perez et al. “FiLM: Visual Reasoning with a General Conditioning Layer.”, 2017

[Eyring+24] Eyring et al., "Unbalancedness in Neural Monge Maps Improves Unpaired Domain Translation", 2024

[Radford+21] Radford et al., “Learning Transferable Visual Models From Natural Language Supervision” 2021

---

### Decision · Program_Chairs · 2024-09-25

**Decision:**

Accept (poster)

**Comment:**

This paper applies optimal transport/conditional flow matching to learn cell development dynamics from single cell transcriptomics.

All the reviewers liked the paper and are in favour of acceptance. The reviewers appreciated the clarity of the paper which is an accomplishment given that it covers a lot of pretty complicated ground.